# ReservoirTTA: Prolonged Test-time Adaptation for Evolving and Recurring Domains

**Guillaume Vray**[1*]    **Devavrat Tomar**[1*]    **Xufeng Gao**[1]
**Jean-Philippe Thiran**[1,2]    **Evan Shelhamer**[3,4]    **Behzad Bozorgtabar**[1,2,5]
[1]EPFL    [2]CHUV    [3]UBC    [4]Vector Institute    [5]Aarhus University
[1,2]{firstname.lastname}@epfl.ch    [3,4]shelhamer@cs.ubc.ca

## Abstract

This paper introduces **ReservoirTTA**, a novel plug–in framework designed for prolonged test–time adaptation (TTA) in scenarios where the test domain continuously shifts over time, including cases where domains recur or evolve gradually. At its core, ReservoirTTA maintains a reservoir of domain-specialized models—an adaptive test-time model ensemble—that both detects new domains via online clustering over style features of incoming samples and routes each sample to the appropriate specialized model, and thereby enables domain-specific adaptation. This multi-model strategy overcomes key limitations of single model adaptation, such as catastrophic forgetting, inter-domain interference, and error accumulation, ensuring robust and stable performance on sustained non-stationary test distributions. Our theoretical analysis reveals key components that bound parameter variance and prevent model collapse, while our plug–in TTA module mitigates catastrophic forgetting of previously encountered domains. Extensive experiments on scene-level corruption benchmarks (ImageNet-C, CIFAR-10/100-C), object-level style shifts (DomainNet-126, PACS), and semantic segmentation (Cityscapes→ACDC) — covering recurring and continuously evolving domain shifts — show that ReservoirTTA substantially improves adaptation accuracy and maintains stable performance across prolonged, recurring shifts, outperforming state-of-the-art methods. Our code is publicly available at https://github.com/LTS5/ReservoirTTA.

## 1  Introduction

Deep networks have achieved state-of-the-art performance across many tasks, but their reliability degrades when test-time data deviates from the training distribution. Real-world deployment scenarios such as autonomous driving or surveillance often involve dynamic shifts caused by changing weather, sensor degradation, or environmental variation. These settings call for robust *test-time adaptation (TTA)* methods [26, 47, 21, 40, 37] that enable pre-trained models to adapt on-the-fly, ideally over prolonged periods, without catastrophic forgetting or model collapse. Most existing TTA methods, e.g., efficient TTA (ETA) [27], assume each domain appears only once in the test stream. In real-world long-term deployments, however, domain conditions often recur. As Figure 1 (left) shows, visual distributions may shift and later reappear. Empirically, this recurring behavior destabilizes ETA [27], which lacks explicit regularization, whereas anti-forgetting TTA (EATA) [27] maintains greater long-term stability by constraining parameter drift. Even so, when domains re-emerge, these regularized methods remain vulnerable to *catastrophic forgetting*, as illustrated in Figure 1 (right).

To address this challenge, we clarify two related yet distinct axes of variation—*style* and *domain*—and revisit how their boundaries are detected. Prior work typically treats domains as discrete, source-annotated groups (e.g., different sensors or collection conditions). Styles and domains may not map

---

*Equal contribution

39th Conference on Neural Information Processing Systems (NeurIPS 2025).

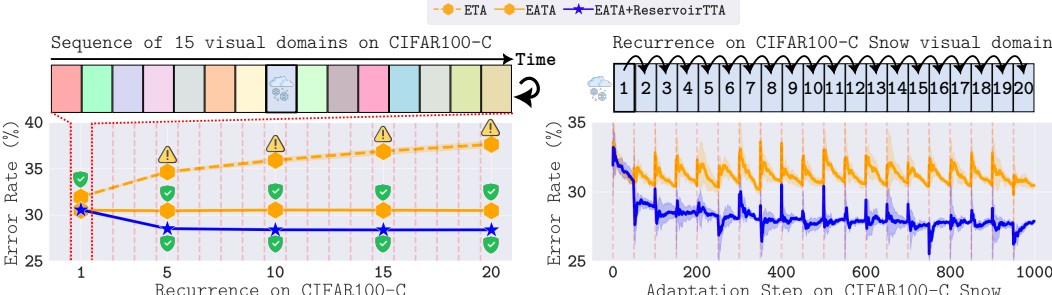

Figure 1: **Recurring test-time adaptation scenarios. Left:** Visual domains can recur over time; ETA [27], lacking regularization, steadily degrades under these repeated shifts. **Right:** A zoom-in on the snow corruption across 20 recurrences shows that EATA [27] remains overall stable but still exhibits error spikes on returning to the same corruption across recurrences. **ReservoirTTA** detects returning domains and reuses specialized models to preserve learned knowledge, delivering improved robustness and faster (re-)adaptation over successive recurrences.

one-to-one: a single domain can exhibit multiple styles due to intra-domain variability, and distinct domains can share similar style signatures (for example, *zoom blur* and *defocus blur* yield nearly identical style statistics). Nevertheless, in our framework, a *domain* is represented by a group of similar styles, where *style* is computed as the set of low-level appearance attributes captured via channel-wise feature statistics from early layers of a pre-trained VGG network [35]. This allows us to leverage the style embeddings to detect new domains or recognize the recurring ones via online clustering without over-fragmentation—an issue observed in the recent methods lacking robust style representations (see Figure 17) [47, 3]. Based on the style of the current test samples, we route them to the corresponding *domain model* in our reservoir, where specialized adaptation is performed.

In summary, we introduce **ReservoirTTA**, which maintains a pool of domain-specialized models, adapts each one independently with its corresponding TTA updates, and combines their parameters in a weighted ensemble for final predictions. Our contributions are as follows:

- **Multi-model TTA with domain-aware specialization.** ReservoirTTA explicitly decouples adaptation across domains using a pool of models. This plug-in design supports diverse architectures and lightweight adapters, including normalization statistics, prompts, and LoRA [13] modules.

- **Style-driven clustering for online domain discovery.** We propose an online clustering algorithm based on deep style features. By quantifying style at test time, our method can detect and reuse previously adapted models, enabling continual and efficient re-adaptation.

- **Theoretical insights into long-term stability.** We provide theoretical bounds showing how parameter regularization curbs collapse in single-domain TTA—clarifying the stability of methods such as EATA [27]—and motivate the modular design of ReservoirTTA for maintaining stability under recurring domain shifts.

- **Extensive empirical evaluation.** We test ReservoirTTA on long-term/recurring TTA: classification on **ImageNet-C**, **CIFAR-10/100-C**, and segmentation on **Cityscapes**→**ACDC**. We also evaluate object-level style shifts (**DomainNet-126**, **PACS**), consistently surpassing PeTTA [12], RDumb [31], and CoTTA [42] across datasets/backbones.

## 2  Related Work

Prior works on prolonged, recurring test-time adaptation fall into two main strands. Continual TTA methods [42, 45, 36] continually update a single model to track evolving domains but suffer from drift and forgetting when domains recur, while robust/persistent TTA methods [31, 12, 27, 24] employ techniques such as variance constraints or periodic resets to preserve stability yet lack efficient re-adaptation to previously seen shifts. For a concise survey of representative algorithms and their limitations in recurring and evolving settings, see Appendix A.

# 3 Recurring Continual TTA and Theoretical Analysis

## 3.1 Background

**Notation and Setting.** We consider a deep network $f_\theta : X \to Y$, pre-trained on an inaccessible source dataset. At test time, the model parameters $\theta$ are updated using an unsupervised TTA objective $\mathcal{L}_{\text{TTA}}(\mathbf{x}, \theta)$ on incoming test images $\mathbf{x}$. In practice, these updates typically affect only a subset of parameters. Test data is received sequentially in batches, and we assume that a batch of test images $\mathbf{B}_t := \{\mathbf{x}_t^1, \ldots, \mathbf{x}_t^b\}$ at time $t$ is drawn from a test domain distribution $\mathcal{D}_t$, where $\mathcal{D}_t \in \{\mathcal{D}_1, \ldots, \mathcal{D}_K\}$. These distributions are unknown during training and evolve dynamically after deployment. We distinguish three primary scenarios for prolonged domain evolution at test time:

- **Recurring Continual Structure Change (CSC):** Domains change in a predictable order but may reappear (e.g., day–night cycles), denoted as $\mathcal{D}_1 \to \mathcal{D}_2 \to \mathcal{D}_3 \looparrowright \mathcal{D}_1$.
- **Recurring Continual Dynamic Change (CDC):** Domains shift unpredictably, though some conditions recur (e.g., abrupt weather changes), represented as $\mathcal{D}_1 \dashrightarrow \mathcal{D}_4 \dashrightarrow \mathcal{D}_5 \looparrowright \mathcal{D}_4$.
- **Continuously Changing Corruptions (CCC):** Domains evolve gradually via incremental changes (e.g., weather or degradation), so that $\mathcal{D}_1 \to \mathcal{D}_1' \to \mathcal{D}_1'' \to \mathcal{D}_2$, where $\mathcal{D}_i'$ and $\mathcal{D}_i''$ denote successive variations of $\mathcal{D}_i$ before transitioning to $\mathcal{D}_{i+1}$.

For notation, we use $\mathcal{D}_i \to \mathcal{D}_j$ for structured shifts in CSC and CCC, $\mathcal{D}_i \dashrightarrow \mathcal{D}_j$ for unstructured shifts in CDC, and $\mathcal{D}_i \looparrowright \mathcal{D}_j$ to denote recurring domains.

## 3.2 Test-Time Adaptation Trajectory

**Stability Regions for Individual Domains.** We begin by analyzing how standard single-model TTA updates via stochastic gradient descent (SGD) can cause parameter variance to grow linearly over time, increasing the risk of drifting outside the stability region.

**Assumption 1.** *At test-time, the model is updated using an unsupervised TTA objective on the target domain, $\mathcal{L}_{\text{TTA}}(\theta, \mathbf{x})$, which serves as a surrogate for the true task loss $\mathcal{L}_{\text{Task}}(\theta) = \mathbb{E}_{(\mathbf{x}, \mathbf{y})}[\mathcal{L}_{\text{sup}}(\mathbf{x}, \mathbf{y}, \theta)]$, where $\mathcal{L}_{\text{sup}}$ measures model performance using ground truth labels. The optimal parameters for the given task are given as: $\theta_{\text{Task}}^* = \arg\min_\theta \mathcal{L}_{\text{Task}}(\theta)$.*

**Definition 1** (Stability Region). *For a task with optimal parameters $\theta_{\text{Task}}^*$, the stability region is defined as the set of parameters $\{\theta : |\theta - \theta_{\text{Task}}^*| \leq \beta\}$, where $\beta$ is the stability radius beyond which model performance collapses when updated using the TTA objective $\mathcal{L}_{\text{TTA}}(\theta, \mathbf{x})$.*

**Lemma 1** (Parameter Variance Growth). *Under standard SGD-based adaptation, the variance of the updated parameters grows linearly with adaptation steps:*

$$Var[\theta_t] = \eta^2 \sum_{i=0}^{t-1} Var_{\mathbf{x}_i \sim \mathbf{x}}[\nabla \mathcal{L}_{\text{TTA}}(\theta_i, \mathbf{x}_i)] \approx t \cdot \eta^2 \cdot \bar{V}, \tag{1}$$

*where $\eta$ is the learning rate and $\bar{V}$ is the average gradient variance.*

**Theorem 1** (Bound on Divergence Probability). *Let $\theta_t$ denote the model parameters at time $t$, and let $\theta_{\text{Task}}^*$ be the task-specific optimum. Suppose that $\mathbb{E}[\theta_t] \to \theta_{\text{Task}}^*$ and $\|\mathbb{E}[\theta_t] - \theta_{\text{Task}}^*\| < \|\theta_0 - \theta_{\text{Task}}^*\|$. Then for any threshold $\beta > \|\theta_0 - \theta_{\text{Task}}^*\|$, the probability of divergence from the stability region is bounded by:*

$$\Pr\left[\|\theta_t - \theta_{\text{Task}}^*\| > \beta\right] \leq \frac{Var[\theta_t]}{\left(\beta - \|\theta_0 - \theta_{\text{Task}}^*\|\right)^2}, \tag{2}$$

*where $\theta_0$ represents the initial model parameters.*

The above analysis reveals why conventional TTA approaches often suffer from model collapse during prolonged adaptation: *as the number of update steps increases, so does the variance of model parameters*, eventually causing them to drift beyond the stability radius. The proof of Theorem 1 is provided in Appendix B.1. Next, we examine strategies to mitigate this variance growth.

**Variance Reduction Strategies.** To address the parameter variance growth problem, we explore two key strategies utilized in [27, 24]:

**Proposition 1** (Sample Filtering). *Using an active sample selection function $S(x)$ that filters out unreliable samples with high entropy reduces the effective gradient variance.*

In practice, not all test samples contribute equally to model adaptation. One remedy would be to employ an active sample selection function $S(x)$ that filters out unreliable samples—those with high entropy or that are redundant (see [27, 24]). Thus, the effective gradient update becomes:

$$\nabla \tilde{\mathcal{L}}_{\text{TTA}}(\theta, \mathbf{x}) = S(\mathbf{x}) \, \nabla \mathcal{L}_{\text{TTA}}(\theta, \mathbf{x}), \tag{3}$$

thereby reducing effective gradient variance to $V_{\text{eff}} < \bar{V}$ while preserving linear dependence on $t$.

**Proposition 2** (Weight Ensembling). *Interpolating the updated parameters with the source model parameters constrains the adaptation trajectory and bounds the overall variance as:*

$$\text{Var}[\theta_t] \approx \eta^2 \bar{V} \cdot \frac{\alpha^2(1 - \alpha^{2t})}{1 - \alpha^2} < \eta^2 \bar{V} \cdot \frac{\alpha^2}{1 - \alpha^2}, \tag{4}$$

*where $\alpha \in (0, 1]$ controls the contribution of source model parameters.*

Another common strategy to bound the variance is to update parameters by interpolating with the source model $\theta_0$ [24], thereby anchoring the adaptation steps:

$$\hat{\theta}_t = \theta_{t-1} - \eta \nabla \mathcal{L}_{\text{TTA}}(\theta_{t-1}, \mathbf{x}_{t-1}), \quad \theta_t = \alpha \hat{\theta}_t + (1 - \alpha)\theta_0. \tag{5}$$

This "weight ensembling" bounds the overall variance, as each gradient update is geometrically damped by a factor of $\alpha^{2(t-i)}$. A similar strategy [27] applies Fisher regularization [18] with respect to the source parameters $\theta_0$ in the TTA loss:

$$\mathcal{L}_{\text{TTA-fis}}(\theta, \mathbf{x}) = \mathcal{L}_{\text{TTA}}(\theta, \mathbf{x}) + \lambda \cdot (\theta - \theta_0)^T \Omega (\theta - \theta_0), \tag{6}$$

where, $\Omega = \text{diag}([\omega_1, \ldots, \omega_n])$ is the Fisher coefficient matrix that weights each parameters $\theta = [\theta_1, \ldots, \theta_n]$ based on its Fisher Information, and $\lambda$ sets the regularization strength. This formulation keeps the updated parameters close to the source and is equivalent to "weight ensembling."

A detailed proof by induction and the complete variance analysis are provided in Appendix B.1.

**Parameter Drift in Recurring Continual TTA.** Even with these variance control strategies (e.g., sample filtering and weight ensembling), a single adapting model remains vulnerable to parameter drift when the shift between domain-optimal parameters exceeds the stability radius, i.e., $\|\theta^*_{\text{Task}_i} - \theta^*_{\text{Task}_{i+1}}\| > \beta_{i+1}$, for some $i$. As illustrated in Figure 5 of Appendix B.2, such shifts cause the model's parameter trajectory $\{\theta_t\}$ to deviate from the optimal region for a given domain, leading to catastrophic forgetting and negative transfer.

**ReservoirTTA: Decoupled Adaptation Across Domains.** To mitigate catastrophic forgetting and inter-domain interference in recurring continual TTA, we propose ReservoirTTA, a novel framework that partitions adaptation across domains by maintaining up to $K$ domain-specialized models. Each reservoir component is updated exclusively when its corresponding domain is active, thereby isolating the test-time objective $\mathbb{E}_{\mathbf{x} \sim \mathcal{D}_i}[\mathcal{L}_{\text{TTA}}(\theta, \mathbf{x})]$ for $i \in \{1, \ldots, K\}$. While existing strategies—such as sample filtering [27], weight ensembling and Fisher regularization [24] help control the variance of updates and prevent model collapse, they do not fully address catastrophic forgetting when domains reoccur. In contrast, our plug-in ReservoirTTA module disentangles domain-specific adaptation and further mitigates unnecessary re-adaptation when previously encountered domains return.

## 4 Methodology

In this section, we introduce our framework, **ReservoirTTA**, for prolonged TTA in environments with recurring and evolving domains. As illustrated in Figure 2, our framework comprises four stages: (1) **Style Characterization and Domain Identification**—leveraging style-based online clustering to determine the domain characteristics of the current test batch; (2) **Model Reservoir Initialization**—allocating a new domain-specific model when a novel domain is detected in the domain identification step; (3) **Model Reservoir Adaptation**—selectively updating only the model associated with the current domain using state-of-the-art TTA techniques; and (4) **Model Prediction**—Predictions are then obtained via the ensemble's parameters from all domain-specific models (see pseudocode in Appendix C).

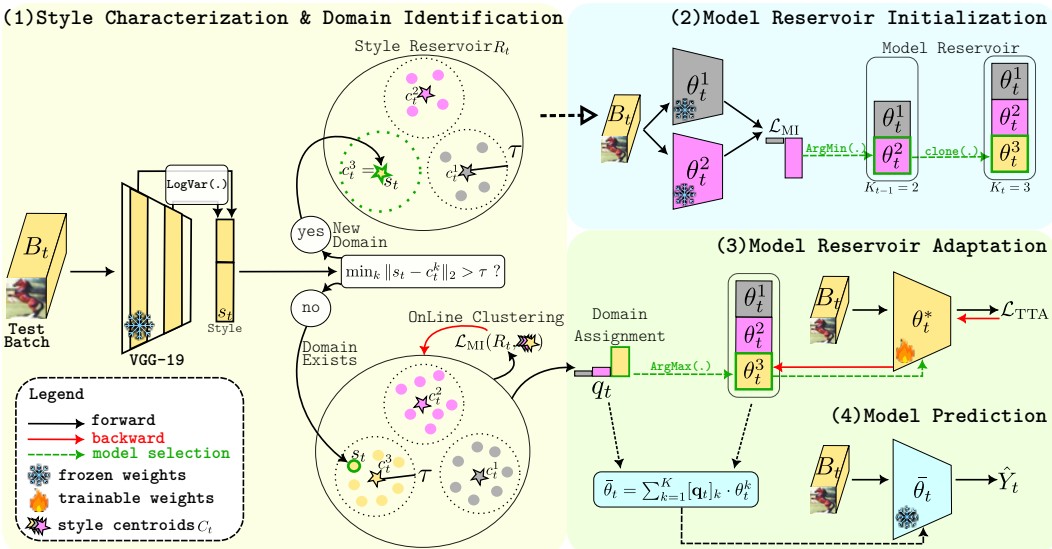

Figure 2: **Overview of ReservoirTTA.** ReservoirTTA operates in four stages: (1) **Style Characterization and Domain Identification** extracts early convolutional features and assigns incoming test batches to a style cluster via an online clustering mechanism; (2) **Model Reservoir Initialization** adds a new model for a detected domain, initializing it with parameters that maximize prediction mutual information; (3) **Model Reservoir Adaptation** selectively adapts the most relevant model using TTA methods; and (4) **Model Prediction** is then obtained via the ensemble's parameters.

## 4.1 Style Characterization and Domain Identification

**Style Characterization.** To distinguish domains at test time, following prior style-characterization work [9, 15, 41, 35], we encode image style as batch-wise log-variances from the first $L$ layers of a frozen, ImageNet-trained VGG-19 $g_{\text{style}}$. Given a batch $\mathbf{B}_t \sim \mathcal{D}_t$ and feature maps $\{\mathbf{z}_1, \ldots, \mathbf{z}_L\}$ with $\mathbf{z}_l \in \mathbb{R}^{b \times h_l \times w_l \times c_l}$, the style vector for layer $l$ is computed as $\mathbf{s}_l(\mathbf{B}_t) = \texttt{logvar}(\mathbf{z}_l)$ where $\texttt{logvar}$ computes the natural logarithm of the variance over the batch, height, and width. The overall style descriptor is formed by concatenating these vectors: $\mathbf{s}_t = \left[\mathbf{s}_1(\mathbf{B}_t), \ldots, \mathbf{s}_L(\mathbf{B}_t)\right] \in \mathbb{R}^d$. This architecture-agnostic statistic captures robust texture cues while remaining independent of the source model; Appendix D analyzes VGG configurations and shows that VGG-based style vectors yield more stable, higher-quality style clustering than source-model and ViT-based alternatives.

**Domain Identification via Online Clustering.** At test time, the number of visual domains is unknown. We introduce an online clustering algorithm inspired by DP-Means [19] to dynamically identify new domains using style vectors $\mathbf{s}_t$. At each timestep $t$, we maintain centroids $\mathbf{C}_t = [\mathbf{c}_t^1, \ldots, \mathbf{c}_t^{K_t}] \in \mathbb{R}^{d \times K_t}$, where $K_t$ is the number of domains identified up to time $t$. $\mathbf{C}_t$ is initialized with a single centroid as the mean style feature from the source domain ($K_0 = 1$). A new cluster is created if the current style vector is sufficiently distant from all existing centroids:

$$\min_{k \in \{1, \ldots, K_t\}} \|\mathbf{s}_t - \mathbf{c}_t^k\|_2 > \tau, \tag{7}$$

where $\tau$ is set to the $q$-quantile of pairwise distances among style vectors from the source domain, ensuring that new clusters are formed only under substantial domain shifts.

To adapt the centroids over time, we optimize a mutual information loss $\mathcal{L}_{\text{MI}}$ between past style vectors and the current centroids. Storing all past styles is infeasible, so we maintain a fixed-size *Style Reservoir* $\mathbf{R}_t := [\mathbf{s}_{t_1}, \ldots, \mathbf{s}_{t_M}]$, updated via Reservoir Sampling [39, 4] to approximate uniform sampling over all previously seen $\mathbf{s}_t$. At each step, $\mathbf{s}_t$ is added to $\mathbf{R}_{t-1}$ if $t \leq M$; otherwise, it replaces a randomly selected vector with probability $\frac{M}{t}$, ensuring unbiased coverage.

We define the soft-assignment for each style vector in the style reservoir to the centroids as follows:

$$\mathbf{Q}(\mathbf{R}_t, \mathbf{C}) = \texttt{softmax}\left( \begin{bmatrix} -\|\mathbf{s}_{t_1} - \mathbf{c}^1\|, \ldots, -\|\mathbf{s}_{t_1} - \mathbf{c}^K\| \\ \vdots \\ -\|\mathbf{s}_{t_M} - \mathbf{c}^1\|, \ldots, -\|\mathbf{s}_{t_M} - \mathbf{c}^K\| \end{bmatrix} / \sqrt{d} \right). \tag{8}$$

The centroids $\mathbf{C}_t$ are updated by gradient descent on the mutual information loss:

$$\mathbf{C}_t \leftarrow \mathbf{C}_{t-1} - \eta \nabla_{\mathbf{C}} \mathcal{L}_{\text{MI}}(\mathbf{Q}(\mathbf{R}_t, \mathbf{C}))\Big|_{\mathbf{C}=\mathbf{C}_{t-1}}, \tag{9}$$

where

$$\mathcal{L}_{\text{MI}}(\mathbf{Q}(\mathbf{R}_t, \mathbf{C})) = \mathcal{L}_{\text{ent}}(\mathbf{Q}(\mathbf{R}_t, \mathbf{C})) + \mathcal{L}_{\text{cm}}(\mathbf{Q}(\mathbf{R}_t, \mathbf{C})). \tag{10}$$

Here, $\mathcal{L}_{\text{ent}}(\mathbf{Q}) = -\frac{1}{M} \sum_{i=1}^{M} \sum_{j=1}^{K} q_{ij} \log q_{ij}$ encourages confident assignments of style vectors in $\mathbf{R}_t$ to the nearest style centroids, while $\mathcal{L}_{\text{cm}}(\mathbf{Q}) = \sum_{j=1}^{K} \bar{q}_j \log \bar{q}_j$ (with $\bar{q}_j = \frac{1}{M} \sum_{i=1}^{M} q_{ij}$) acts to prevent centroid collapse by promoting assignment diversity. Appendix E presents the sensitivity of all introduced hyperparameters, alternative online clustering strategies, and alternative distance metrics, including cosine similarity and Kullback–Leibler (KL) divergence, as well as the style reservoir update.

## 4.2 Model Reservoir: Initialization, Adaptation, and Prediction

Building on the theoretical analysis of parameter drift (Section 3.2), we maintain a *Model Reservoir* comprising $K_t$ domain-specialized models $\{\theta_t^1, \ldots, \theta_t^{K_t}\}$, with one model per discovered domain up to time $t$. To ensure computational efficiency, we store only the trainable parameters of each model, rather than full model instances. The total number of domains is bounded by a fixed constant $K^{\text{max}}$ to prevent memory exhaustion. At initialization, the reservoir contains a single model $\{\theta_0^1\}$ corresponding to the source domain. When a new domain is detected, we instantiate a new model by cloning the parameters of an existing reservoir model that yields the most confident and diverse predictions on the current test batch $\mathbf{B}_t$. This is formalized using mutual information as follows:

$$\theta_0^{K_t} = \arg \min_{\theta \in \{\theta_t^1, \ldots, \theta_t^{K_t-1}\}} \mathcal{L}_{\text{MI}}(f_\theta(\mathbf{B}_t)), \tag{11}$$

where $f_\theta(\mathbf{B}_t) \in \mathbb{R}^{b \times |Y|}$ is the softmax output of the model. This criterion favors models whose predictions are simultaneously confident and diverse, reducing the risk of collapse when adapting to a novel domain (see alternative initialization in Appendix F).

**Model Adaptation.** Given a test batch $\mathbf{B}_t$, we compute a soft assignment vector $\mathbf{q}_t \in \mathbb{R}^{K_t}$ by comparing the current style vector $\mathbf{s}_t$ to domain centroids $\{\mathbf{c}_t^1, \ldots, \mathbf{c}_t^{K_t}\}$ using scaled negative squared Euclidean distances followed by a `softmax`:

$$\mathbf{q}_t = \texttt{softmax}\left( \left[ -\|\mathbf{s}_t - \mathbf{c}_t^1\|, \ldots, -\|\mathbf{s}_t - \mathbf{c}_t^K\| \right] / \sqrt{d} \right). \tag{12}$$

The most relevant model is then selected as $k_* = \arg\max_{1 \leq j \leq K} [\mathbf{q}_t]_j$. The selected model $\theta_t^{k_*}$ is adapted using a test-time adaptation objective $\mathcal{L}_{\text{TTA}}$:

$$\theta_{t+1}^{k_*} \leftarrow \theta_t^{k_*} - \eta \nabla_\theta \mathcal{L}_{\text{TTA}}(\mathbf{B}_t, \theta_t^{k_*}), \tag{13}$$

**Model Prediction.** For inference, the trainable parameters from all reservoir models are ensembled according to their soft assignment weights, yielding the ensemble parameters:

$$\bar{\theta}_t = \sum_{k=1}^{K} [\mathbf{q}_t]_k \cdot \theta_t^k. \tag{14}$$

The final prediction is then computed as $\hat{\mathbf{Y}}_t = f_{\bar{\theta}_t}(\mathbf{B}_t)$. Notably, $\bar{\theta}_t$ is used solely for prediction and is not updated, ensuring that all specialized models contribute to the output without being overwritten.

# 5 Experiments

**Datasets and Evaluation Metrics.** We evaluate on standard scene-level corruption benchmarks—CIFAR-10→CIFAR-10-C, CIFAR-100→CIFAR-100-C, and ImageNet→ImageNet-C—using three CNN backbones with only batch/group norms updated; for ImageNet-C, we also test ViT-B/16 (see Appendix H). To demonstrate shift-agnosticism, we further assess object-level style-shift benchmarks, DomainNet-126 [30, 33] and PACS [22], using a ResNet-50 following [2]. For segmentation, we use Segformer-B5 as in CoTTA [42]. Classification is tested under CCC [31], CSC, and CDC settings over 20 rounds (averaging error rates, %; a subset is shown for clarity). For segmentation, we follow the Cityscapes→ACDC protocol [42], where ACDC presents four weather conditions (Fog, Night, Rain, Snow) sequentially. We report the mean IoU (%) averaged over 10 repetitions.

**Baselines.** We evaluate our method against several state-of-the-art TTA baselines (for more details, see the Appendix G). For **single-target TTA**, we compare with TENT [40]. For **continual TTA**, we consider CoTTA [42] (using the affine-parameter variant, CoTTA*), RoTTA [45], ETA [27], and SAR [28]. To assess long-term stability, we include **persistent TTA** methods such as RDumb [31], PeTTA [12], EATA [27] (with Fisher-based regularization), and ROID [24] (reported as ROID*—a version omitting the augmentation consistency loss). For ViT-based models, we also compare with domain-disentanglement approaches like CoLA [3] and DPCore [47]. In segmentation experiments, we evaluate segmentation variants of TENT, CoTTA*, and BECoTTA [20]. For fair comparison, all methods update only the backbone's affine parameters—except when we compare ReservoirTTA and BECoTTA (using LoRA) and DPcore (using visual prompts).

**Methods Tested for ReservoirTTA Plug-in.** Based on our theoretical analysis, we integrate ReservoirTTA only into TTA methods that employ variance reduction via sample filtering along with variance control through weight ensembling and Fisher regularization. Accordingly, we apply ReservoirTTA to EATA and ROID*, which incorporate both components, as well as to ETA and SAR, which use sample filtering alone, and to TENT even though it lacks both. For segmentation tasks, we plug ReservoirTTA into CoTTA* and BECoTTA to demonstrate its generality. Although methods such as RoTTA and PeTTA are compatible in principle, they require a separate memory bank for each reservoir model which makes their practical integration computationally prohibitive.

**Implementation Details.** All methods are re-implemented in `PyTorch` [29] within a unified TTA repository [24] for fair comparison, using pre-trained source models from `RobustBench` [11]. See Appendix H for further implementation details.

## 5.1 Main Results

**Classification.** Table 1 highlights the limitations of existing TTA methods when repeatedly adapted under recurring domain shifts and prolonged testing with three CNN-based backbones. **(1) Single-target TTA**: Methods like TENT initially adapt well but suffer from severe error accumulation over multiple cycles—for example, on CIFAR-10-C, error rises from 19.3% at visit 1 to 87.8% at visit 20, with similar trends on CIFAR-100-C and ImageNet-C. Plugging in ReservoirTTA significantly improves performance, though error accumulation still occurs, highlighting the need for both variance control modules from our theoretical analysis. **(2) Continual TTA**: Approaches like CoTTA* and RoTTA mitigate short-term instability yet still accumulate errors with repeated adaptation. ReservoirTTA decouples domain-specific adaptation to prevent parameter drift—reducing, for instance, ETA's CIFAR-10-C error from 30.9% to 16.4% at the 20th visit (Recurring CSC) and similarly improving ImageNet-C performance. **(3) Persistent TTA**: Although methods such as EATA and ROID* are designed to prevent catastrophic forgetting, they exhibit limited re-adaptation (Figure 1), with minimal improvements over repeated visits. ReservoirTTA overcomes this by enabling controlled adaptation without excessive re-learning—reducing EATA's ImageNet-C error from 55.9% to 51.0% in Recurring CSC and ROID's from 55.5% to 52.1%.

As shown in Table 2, using a ViT-B-16 backbone on ImageNet-C further underscores ReservoirTTA's advantages in recurrent continual TTA. Domain-disentangled methods like DPCore improve early adaptation but lack long-term stability. For instance, in Recurring CSC, CoLA reduces ETA's 20th-visit error to 33.8%, whereas ReservoirTTA lowers it further to 31.9%. In CCC, CoLA stabilizes ETA at 40.2%, whereas ReservoirTTA achieves 38.5%. DPCore also struggles: in CCC, its error rises from 42.2% to 43.1%, whereas combining ReservoirTTA with the same prompt-tuning approach

Table 1: **Average classification error (%)** on corruption benchmarks under recurring CSC and CDC. Results shown at visits 1 and 20, with their difference ($\Delta$), for CIFAR-10-C, CIFAR-100-C, and ImageNet-C using WideResNet-28, ResNeXt-29, and ResNet-50, respectively. Averages over five runs. Best in **bold**, second best underlined.

| | Recurring CSC | | | | | | | | | Recurring CDC | | | | | | | | |
| | CIFAR-10-C | | | CIFAR-100-C | | | ImageNet-C | | | CIFAR-10-C | | | CIFAR-100-C | | | ImageNet-C | | |
| | *Recurring visit* | | | *Recurring visit* | | | *Recurring visit* | | | *Recurring visit* | | | *Recurring visit* | | | *Recurring visit* | | |
| **Method** | 1 | 20 | $\Delta$ | 1 | 20 | $\Delta$ | 1 | 20 | $\Delta$ | 1 | 20 | $\Delta$ | 1 | 20 | $\Delta$ | 1 | 20 | $\Delta$ |
|---|---|---|---|---|---|---|---|---|---|---|---|---|---|---|---|---|---|---|
| Source | 43.5 | 43.5 | +0.0 | 46.5 | 46.5 | +0.0 | 82.0 | 82.0 | +0.0 | 43.5 | 43.5 | +0.0 | 46.5 | 46.5 | +0.0 | 82.0 | 82.0 | +0.0 |
| **Single-Target TTA** | | | | | | | | | | | | | | | | | | |
| TENT ( ICLR 21 ) | 19.3 | 87.8 | +68.5 | 61.4 | 99.0 | +37.6 | 62.6 | 99.5 | +36.9 | 20.5 | 87.0 | +66.5 | 60.2 | 98.9 | +38.7 | 62.0 | 99.5 | +37.5 |
| *+ReservoirTTA* | 18.3 | 17.6 | -0.7 | 38.1 | 44.0 | +5.9 | 62.6 | 58.2 | -4.4 | 18.2 | 17.4 | -0.8 | 33.9 | 39.7 | +5.8 | 62.4 | 57.5 | -4.9 |
| **Continual TTA** | | | | | | | | | | | | | | | | | | |
| CoTTA* ( CVPR 22 ) | 18.8 | 22.4 | +3.6 | 35.1 | 65.5 | +30.4 | 67.6 | 62.7 | -4.9 | 18.8 | 22.3 | +3.5 | 35.1 | 65.1 | +30.0 | 67.7 | 61.5 | -6.2 |
| RoTTA ( CVPR 23 ) | 19.4 | 18.4 | -1.0 | 34.8 | 59.1 | +24.3 | 67.3 | 99.4 | +32.1 | 21.9 | 20.4 | -1.5 | 36.8 | 73.8 | +37.0 | 71.6 | 99.5 | +27.9 |
| ETA ( ICML 22 ) | 17.8 | 30.9 | +13.1 | 32.0 | 37.6 | +5.6 | 60.0 | 59.4 | -0.6 | 17.9 | 33.5 | +15.6 | 32.4 | 37.6 | +5.2 | 59.3 | 60.1 | +0.8 |
| *+ReservoirTTA* | 17.5 | 16.4 | -1.1 | 31.6 | 30.0 | -1.6 | 59.8 | 53.1 | -6.7 | 17.4 | 16.3 | -1.1 | 30.9 | 29.7 | -1.2 | 58.6 | 52.2 | -6.4 |
| SAR ( ICLR 23 ) | 20.4 | 20.4 | +28.5 | 31.9 | 60.4 | +28.5 | 61.9 | 67.1 | +5.2 | 20.4 | 20.4 | +0.0 | 31.6 | 57.8 | +26.2 | 61.5 | 66.2 | +4.7 |
| *+ReservoirTTA* | 20.4 | 20.4 | +0.0 | 31.9 | 30.5 | -1.4 | 62.2 | 53.1 | -9.1 | 20.4 | 20.4 | +0.0 | 31.7 | 29.8 | -1.9 | 62.6 | 53.6 | -9.0 |
| **Persistent TTA** | | | | | | | | | | | | | | | | | | |
| RDumb ( NeurIPS 23 ) | 17.8 | 18.4 | +0.6 | 32.0 | 32.9 | +0.9 | 59.8 | 56.8 | -3.0 | 17.9 | 18.1 | +0.2 | 32.4 | 32.6 | +0.2 | 59.6 | 59.5 | -0.1 |
| PeTTA ( NeurIPS 24 ) | 23.0 | 17.2 | -5.8 | 39.4 | 32.9 | -6.5 | 67.5 | 60.1 | -7.4 | 27.2 | 20.8 | -6.4 | 42.1 | 35.3 | -6.8 | 71.6 | 69.5 | -2.1 |
| EATA ( ICML 22 ) | 17.5 | 17.8 | +0.3 | 30.5 | 30.5 | +0.0 | 57.5 | 55.9 | -1.6 | 17.7 | 17.9 | +0.2 | 31.0 | 31.1 | +0.1 | 58.5 | 57.0 | -1.5 |
| *+ReservoirTTA* | 17.5 | 16.4 | -1.1 | 30.6 | 28.4 | -2.2 | 58.0 | 51.0 | -7.0 | 17.5 | 16.4 | -1.1 | 30.4 | 28.4 | -2.0 | 58.5 | 51.8 | -6.7 |
| ROID* ( WACV 24 ) | 17.8 | 17.7 | -0.1 | 29.5 | 29.3 | -0.2 | 56.1 | 55.5 | -0.6 | 18.0 | 18.1 | +0.1 | 30.2 | 30.1 | -0.1 | 58.7 | 58.3 | -0.4 |
| *+ReservoirTTA* | 17.8 | 16.8 | -1.0 | 29.6 | 27.8 | -1.8 | 56.4 | 52.1 | -4.3 | 17.9 | 16.8 | -1.1 | 29.6 | 27.8 | -1.8 | 57.0 | 53.0 | -4.0 |

Table 2: **Average classification error (%)** on ImageNet-C under recurring continual TTA (ViT-B/16). For CCC, we average over an adaptation window (e.g., steps 6701–40200). Means over 5 seeds. % Train Params = fraction of trainable parameters; Time = × vs. Source. Margins show negative/positive error changes vs. the base plug-in.

| | Recurring CSC | | | Recurring CDC | | | CCC | | | Complexity | |
| | *Recurring visit* | | | *Recurring visit* | | | *Adaptation Step* | | | % **Train** | |
| **Method** | 1 | 20 | $\Delta$ | 1 | 20 | $\Delta$ | 6.7k | 40.2k | 80k | **Params** | **Time** |
|---|---|---|---|---|---|---|---|---|---|---|---|
| Source | 48.8 | 48.8 | 0.0 | 48.8 | 48.8 | 0.0 | 51.9 | 49.3 | 50.7 | 0.000 | 1.0 |
| **Continual TTA** | | | | | | | | | | | |
| ETA (ICML 22) | 38.9 | 48.4 | +9.5 | 42.9 | 35.9 | -7.0 | 42.7 | 40.6 | 40.8 | 0.044 | 2.0 |
| +*CoLA* ( NeurIPS 24 ) | 41.0 | 33.8 | -7.2 | 40.9 | 34.9 | -6.0 | 44.8 | 40.5 | 40.2 | 6.241 | 2.0 |
| *+ReservoirTTA* | 39.4 | 31.9 | -7.5 | 41.6 | 32.9 | -8.7 | 44.1 | 40.2 | 38.5 | 0.705 | 3.0 |
| **Prompt-based TTA** | | | | | | | | | | | |
| DPCore | 40.2 | 46.0 | +5.8 | 42.8 | 47.6 | +4.8 | 42.2 | 42.3 | 43.1 | 1.053 | 3.8 |
| VPT+*ReservoirTTA* | 38.0 | 33.7 | -4.3 | 38.0 | 34.3 | -3.7 | 42.9 | 40.2 | 39.2 | 0.113 | 3.0 |

(VPT) [14]—used by DPCore for fair comparison—keeps the error at 39.2%. This gap shows that the domain identification mechanism in ReservoirTTA outperforms those in CoLA and DPCore by avoiding gradual degradation and accelerating performance improvement. As shown in Appendix D, our style features form higher-quality clusters than the ViT-based features used in CoLA and DPCore. Moreover, our model discovery mechanism is less sensitive to test batch order and estimates the number of domains more accurately (see Figure 4 and Appendix I), resulting in fewer trainable parameters compared to DPCore and COLA. See Tables 12–25 in Appendix K for full results across all 20 recurrences, CNN backbones (group norm), DomainNet-126, and PACS.

**Segmentation.** Table 3 shows segmentation results on Cityscapes→ACDC under recurring CSC. Methods such as TENT, CoTTA*, and BECoTTA suffer mIoU declines when re-encountering each domain, indicating limited re-adaptation. By contrast, plugging in ReservoirTTA consistently boosts mIoU and limits performance drift. For example, TENT + ReservoirTTA gains over two percentage points on Snow by the 10th revisit, while CoTTA* + ReservoirTTA and LORA + ReservoirTTA show steady improvements across all conditions compared to BECoTTA. This demonstrates that our domain-specific reservoir effectively preserves and reapplies learned knowledge over multiple visits (visit Appendix J, Figure 18 for segmentation visualizations).

Table 3: **Semantic segmentation results (mIoU %)** on Cityscapes→ACDC under recurring CSC. Each target domain (Fog→Night→Rain→Snow) is revisited over 10 iterations. For fair comparison, stochastic restoration is disabled in CoTTA* and BECoTTA to ensure reproducibility. Best and second-best results are shown in bold and underline, respectively. Positive and Negative margins indicate mIoU changes relative to the plug-in method.

| | Fog | | | | Night | | | | Rain | | | | Snow | | | |
|---|---|---|---|---|---|---|---|---|---|---|---|---|---|---|---|---|
| | *Recurring visit* ⟶ | | | | *Recurring visit* ⟶ | | | | *Recurring visit* ⟶ | | | | *Recurring visit* ⟶ | | | |
| **Method** | **1** | **4** | **7** | **10** | **1** | **4** | **7** | **10** | **1** | **4** | **7** | **10** | **1** | **4** | **7** | **10** |
| Source | 69.1 | 69.1 | 69.1 | 69.1 | 40.3 | 40.3 | 40.3 | 40.3 | 59.7 | 59.7 | 59.7 | 59.7 | 57.8 | 57.8 | 57.8 | 57.8 |
| TENT ( ICLR 21 ) | 69.0 | 68.0 | 66.7 | 65.3 | 40.2 | 38.0 | 36.0 | 34.1 | 60.0 | 59.9 | 58.5 | 56.7 | 57.5 | 55.9 | 53.8 | 52.0 |
| *+ReservoirTTA* | 69.1 | 68.8 | 68.1 | 67.6 | 40.2 | 39.3 | 38.9 | 38.5 | 59.9 | 59.9 | 59.2 | 58.0 | 57.6 | 57.0 | 56.0 | 54.7 |
| | +0.1 | +0.8 | +1.4 | +2.3 | +0.0 | +1.3 | +2.9 | +4.4 | -0.1 | +0.0 | +0.7 | +1.3 | +0.1 | +1.1 | +2.2 | +2.6 |
| CoTTA* ( CVPR 22 ) | 71.5 | 71.4 | 71.3 | 71.2 | 41.4 | 40.8 | 40.3 | 39.9 | 62.7 | 63.0 | 63.2 | 63.3 | 59.9 | 59.9 | 59.8 | 59.8 |
| *+ReservoirTTA* | **72.9** | **72.8** | **72.8** | **72.7** | 41.1 | 40.7 | 40.5 | 40.2 | **64.4** | 64.6 | 64.6 | 64.7 | 60.2 | 60.1 | 60.0 | 60.0 |
| | +1.4 | +1.4 | +1.5 | +1.5 | -0.3 | -0.1 | +0.2 | +0.3 | +1.7 | +1.6 | +1.4 | +1.4 | +0.3 | +0.2 | +0.2 | +0.2 |
| BECoTTA ( ICML 24 ) | 72.0 | 72.3 | 72.7 | 72.5 | 41.2 | 41.4 | 40.7 | 40.8 | 63.4 | 64.0 | 64.6 | 64.8 | 60.3 | **61.3** | 61.0 | 60.3 |
| LORA *+ReservoirTTA* | 72.7 | 72.7 | 72.4 | 72.4 | **41.5** | **41.4** | **41.5** | **41.4** | 64.3 | **64.6** | **64.7** | **64.9** | **61.0** | 61.2 | **61.1** | **61.2** |
| | +0.7 | +0.4 | -0.3 | -0.1 | +0.3 | +0.0 | +0.8 | +0.6 | +0.9 | +0.6 | +0.1 | +0.1 | +0.7 | -0.1 | +0.1 | +0.9 |

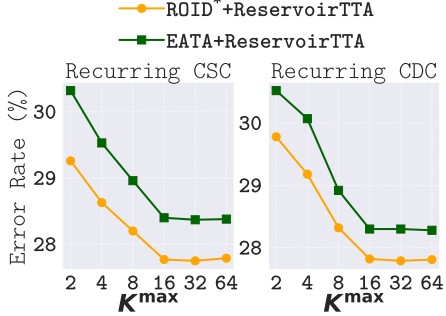

Figure 3: **Sensitivity to reservoir size** $K^{\mathbf{max}}$ on CIFAR100-C (recurring CSC/CDC).

Table 4: **Component-wise ablation of ReservoirTTA with EATA [27].** Average error (%) on CIFAR-100-C (CSC, CDC at visit 20) and ImageNet-C (CCC). Negative margins show improvement over EATA. We ablate Model Reservoir (MR), Style Reservoir (SR), and Ensembling (EMR). Best results in bold. Time is a multiplicative factor relative to EATA.

| | Components | | | Recurring CSC | | Recurring CDC | | CCC | | |
|---|---|---|---|---|---|---|---|---|---|---|
| Method | MR | SR | EMR | Error | Gain | Err | Gain | Error | Gain | Time |
| EATA | ✗ | ✗ | ✗ | 30.5 | - | 31.1 | - | 60.3 | - | 1.0 |
| EATA+ | ✔ | ✗ | ✗ | 28.6 | -1.9 | 28.5 | -2.6 | 59.5 | -0.8 | 1.1 |
| | ✔ | ✔ | ✗ | 28.4 | -2.1 | 28.5 | -2.6 | 59.1 | -1.2 | 1.3 |
| **EATA+*ReservoirTTA*** | ✔ | ✔ | ✔ | **28.4** | **-2.1** | **28.4** | **-2.7** | **58.8** | **-1.5** | 1.3 |

## 5.2 Analyses and Ablations

**Component-wise Ablation Analysis and Runtime.** Table 4 quantifies the impact of ReservoirTTA's components on CIFAR-100-C (CSC, CDC) and ImageNet-C (CCC), along with runtime relative to EATA. Incorporating the Model Reservoir (MR) enables domain-specific adaptation, yielding a 1.9% gain at visit 20 in CSC. The Style Reservoir (SR) has limited effect on CSC/CDC—frequent, abrupt domain resets make styles easily separable, so a single embedding suffices—but helps in CCC, where gradual drift benefits from accumulated history (+1.2%). EMR ensembling is marginal on CSC/CDC (assignments are near one-hot) yet gives +1.5% on CCC by leveraging multiple models. As shown in Appendix I, reusing specialists via MR outperforms simple weight resets: RDumb's blind reset 32.2% → domain-aware reset 31.2% → MR 28.4% (CSC, visit 20). Overall, ReservoirTTA improves accuracy by +2.1% (CSC), +2.7% (CDC), and +1.5% (CCC) at only $1.3\times$ EATA's runtime.

**Sensitivity Analysis of ReservoirTTA Hyperparameters.** Figure 3 highlights the importance of setting $K^{\max}$, with classification error stabilizing for $K^{\max} \geq 16$ on CIFAR-100-C consistent with its domain structure. Overestimating the number of domains helps avoid premature merging and improves specialization. Moreover, as shown in Figure 4, ReservoirTTA is robust to batch-order variability, reliably estimating the number of domains after the first recurrence in the CSC setting even without constraining $K^{\max}$. Across subsequent recurrences, the detected domain count increases marginally, consistent with the stable performance reported above. Additional ablations on CIFAR-100-C and CCC (see Appendix E) show that ReservoirTTA is robust to key hyperparameters: performance remains stable across style reservoir sizes $M$, and variations in source sample count or quantile $q_{\text{th}}$ for threshold $\tau$ in Equation (7) affect error by less than 1%. As shown in Appendix I, larger batch sizes improve performance, but ReservoirTTA consistently outperforms baselines across

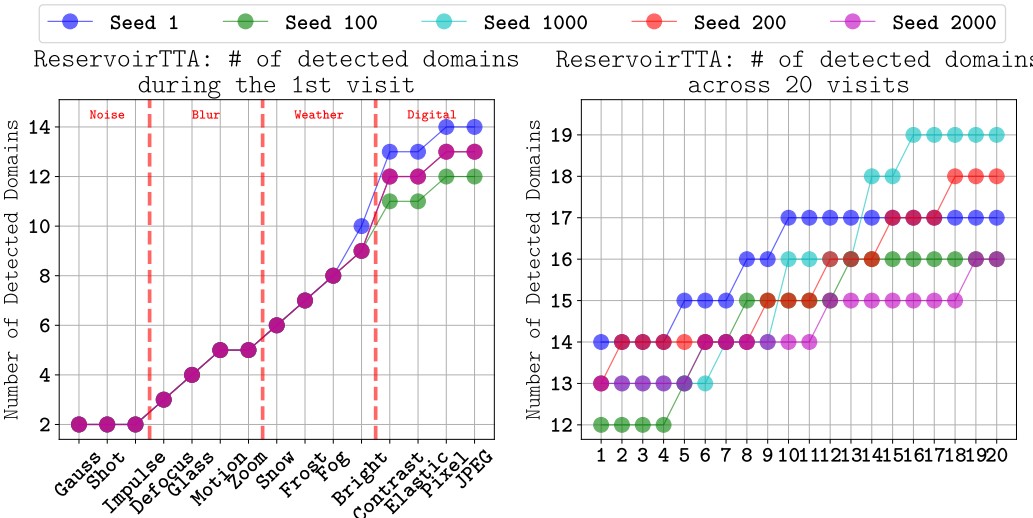

Figure 4: **Domain detection sensitivity of ReservoirTTA in recurring CSC on CIFAR-100-C.**
The plot shows the number of detected domains during the first recurrence (Left) and subsequent
recurrences (Right) across different batch-order seeds. ReservoirTTA is largely insensitive to batch
order, accurately estimating the number of domains, which remains stable over time.

settings. Notably, it requires less reliance on weight ensembling or Fisher regularization, indicating
strong inherent variance control.

**Additional Analysis.** We further test two challenging TTA settings: (i) gradual shifts with smoothly
varying severity, where ReservoirTTA adapts well to subtle transitions, and (ii) temporally corre-
lated streams with category bias, where pairing with sampling/stabilization mitigates bias. Results
(Appendix K) show effectiveness across diverse, realistic test-time scenarios.

## 6    Conclusion

We present ReservoirTTA as a novel framework to extend test-time adaptation from single model to
multiple model adaptation by decoupling updates across a reservoir of domain-specialized models.
Rather than forcing a single model to adapt continuously, our approach selectively updates the
most relevant component based on dynamic clustering of style features. Furthermore our approach
identifies, assigns, and updates its specialists fully at test time without needing multiple source
domains for training. This design not only stabilizes the adaptation process but also curtails the
accumulation of errors and catastrophic forgetting that typically plague single-model methods. Our
theoretical analysis and comprehensive experiments underscore the framework's ability to maintain
robust performance even under prolonged and unpredictable domain shifts.

**Limitations.**  Our plug-in approach improves adaptation by decoupling domain-specific updates
but introduces additional computational overhead due to the Model Reservoir, online clustering, and
refinement using a Style Reservoir. This can increase computation by up to 30% on top of lightweight
methods such as EATA and ETA. However, the memory overhead remains low, as only trainable
parameters and not full models are duplicated for the Model Reservoir.

A further limitation—common to most TTA methods—is updating parameters on every incoming
batch regardless of convergence. Introducing an adaptive update trigger (to switch optimization
on/off) could markedly reduce runtime overhead and improve practicality in resource-constrained
settings.

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

# Appendix

In this appendix, we provide further details on related work (Appendix A), our theoretical analysis (Appendix B), our algorithm (Appendix C), style features quality (Appendix D), domain identification via online clustering (Appendix E), model reservoir initialization (Appendix F), baselines (Appendix G), implementation details (Appendix H), additional ablation studies (Appendix I), as well as extra qualitative (Appendix J) and quantitative results (Appendix K). **Note:** The numbering of tables, figures, and equations in this appendix continues from the main document.

## A    Related Work

**Continual Test-Time Adaptation.** Early TTA approaches adapt pre-trained models to a fixed, stationary target domain by updating only a small subset of parameters—typically the affine parameters of BatchNorm—using techniques such as pseudolabeling or entropy minimization [40, 25]. However, these *single–target TTA* methods are prone to error accumulation when faced with prolonged or recurring continual domain shifts. To address these limitations, *continual TTA (CTTA)* methods (e.g., CoTTA [42], RoTTA [45], and EcoTTA [36]) employ teacher–student frameworks, self–distillation, and regularization techniques to improve robustness in adapting pre-trained models across evolving target domains. Nevertheless, most CTTA methods update a single shared model across domains, leading to slow convergence on brief exposures, catastrophic forgetting, and negative transfer when domains differ. Although they can reduce short-term performance fluctuations, these approaches remain prone to cumulative errors over time and assume each environment appears only once—a condition that rarely holds in practice. Recent works propose distinct paradigms. SAR [28] uses sharpness-aware entropy minimization to stabilize adaptation by suppressing noisy gradients. BECoTTA [20] updates domain-specific low-rank experts through adaptive routing, reducing interference but relies on access to multi-source data before TTA deployment. CoLA [3] enables collaborative test–time adaptation by sharing domain knowledge vectors across devices to improve efficiency, though its success depends on stable inter-device communication in resource-constrained environments. An alternative approach, known as *prompt-based TTA*, leverages visual prompt–based learning [47, 5] to enable domain-specific adaptation without altering the core network. Methods such as DPCore [47], VDP [7], DePT [8], and PAINT [5] employ learnable tokens or pixel-level prompts, though they typically require a warm-up phase to compute alignment statistics and reliably initialize the prompts.

**Robust and Persistent Test–Time Adaptation.** *Persistent TTA* approaches are designed to sustain robust performance over extended periods. Techniques such as active sample selection, Fisher–based regularization, and weight ensembling have been explored in methods like EATA [27] and ROID [24] to prevent catastrophic forgetting and model collapse. Other approaches, such as reset-based methods like RDumb [31], use teacher–student updates or periodic resets to counteract adaptation error accumulation. Similarly, PeTTA [12] dynamically adjusts its adaptation strategy to prevent model collapse without frequent resets. In parallel, Test–Time Ensemble (TTE) [16] leverages linear mode connectivity to form an adaptive *weight-space* ensemble (EMA modulated by a divergence term) together with a dropout ensemble, and applies de-biased reverse-KL distillation to stabilize updates; TTE plugs into standard TTA objectives and improves robustness on continual, non-i.i.d. streams. However, these strategies often exhibit limited re–adaptation capability when encountering previously seen distributions and require precise hyperparameter tuning or source data for initialization. In contrast, our ReservoirTTA is designed to maintain consistent long-term adaptation performance without these constraints.

## B    Details on Theoretical Analysis

### B.1    TTA Variance Bound via Source Weighted Ensembling and Fisher Regularization

**Theorem 2** (Recursive Weight Ensembling Update). *Let the update rule be defined as*

$$\hat{\theta}_t = \theta_{t-1} - \eta \nabla \mathcal{L}_{TTA}(\theta_{t-1}, \mathbf{x}_{t-1}), \tag{15}$$

$$\theta_t = \alpha \cdot \hat{\theta}_t + (1 - \alpha) \cdot \theta_0, \tag{16}$$

*with initial parameter $\theta_0 \in \mathbb{R}^d$, step size $\eta > 0$, and ensembling parameter $\alpha \in [0, 1]$. Then for all $t \geq 1$, the iterates $\theta_t$ admit the closed-form recursion*

$$\theta_t = \theta_0 - \eta \sum_{i=0}^{t-1} \alpha^{t-i} \nabla \mathcal{L}_{TTA}(\theta_i, \mathbf{x}_i). \tag{17}$$

*Proof.* We proceed by induction on $t$.

**Base Case ($t = 1$).** From (15), we have:

$$\hat{\theta}_1 = \theta_0 - \eta \nabla \mathcal{L}_{\text{TTA}}(\theta_0, \mathbf{x}_0).$$

Using (16):

$$\theta_1 = \alpha \hat{\theta}_1 + (1 - \alpha)\theta_0 = \alpha(\theta_0 - \eta \nabla \mathcal{L}_{\text{TTA}}(\theta_0, \mathbf{x}_0)) + (1 - \alpha)\theta_0.$$

Simplifying:

$$\theta_1 = \theta_0 - \eta \alpha \nabla \mathcal{L}_{\text{TTA}}(\theta_0, \mathbf{x}_0),$$

which matches (17) for $t = 1$.

**Inductive Step.** Assume the result holds for some $t \geq 1$, i.e.,

$$\theta_t = \theta_0 - \eta \sum_{i=0}^{t-1} \alpha^{t-i} \nabla \mathcal{L}_{\text{TTA}}(\theta_i, \mathbf{x}_i).$$

Then by (15) and (16):

$$\hat{\theta}_{t+1} = \theta_t - \eta \nabla \mathcal{L}_{\text{TTA}}(\theta_t, \mathbf{x}_t), \quad \theta_{t+1} = \alpha \hat{\theta}_{t+1} + (1 - \alpha)\theta_0.$$

Substituting:

$$\theta_{t+1} = \alpha(\theta_t - \eta \nabla \mathcal{L}_{\text{TTA}}(\theta_t, \mathbf{x}_t)) + (1 - \alpha)\theta_0$$
$$= \alpha \theta_t + (1 - \alpha)\theta_0 - \eta \alpha \nabla \mathcal{L}_{\text{TTA}}(\theta_t, \mathbf{x}_t).$$

Now apply the inductive hypothesis:

$$\theta_{t+1} = \alpha \left( \theta_0 - \eta \sum_{i=0}^{t-1} \alpha^{t-i} \nabla \mathcal{L}_{\text{TTA}}(\theta_i, \mathbf{x}_i) \right) + (1 - \alpha)\theta_0 - \eta \alpha \nabla \mathcal{L}_{\text{TTA}}(\theta_t, \mathbf{x}_t)$$

$$= \theta_0 - \eta \sum_{i=0}^{t-1} \alpha^{t+1-i} \nabla \mathcal{L}_{\text{TTA}}(\theta_i, \mathbf{x}_i) - \eta \alpha \nabla \mathcal{L}_{\text{TTA}}(\theta_t, \mathbf{x}_t)$$

$$= \theta_0 - \eta \sum_{i=0}^{t} \alpha^{(t+1)-i} \nabla \mathcal{L}_{\text{TTA}}(\theta_i, \mathbf{x}_i),$$

which completes the inductive step.

**Conclusion.** By mathematical induction, Equation (17) holds for all $t \geq 1$. $\square$

**Proposition 3** (Bounded Variance under Source-Weighted Ensembling). *Assume that the per-step gradient noise has average variance $\bar{V}$, i.e.,*

$$\mathbb{E}\left[ \text{Var}[\nabla \mathcal{L}_{TTA}(\theta_i, \mathbf{x}_i)] \right] \approx \bar{V}$$

*Then, under the weight ensembling update in (15)–(16), the variance of $\theta_t$ satisfies*

$$\text{Var}[\theta_t] \approx \eta^2 \bar{V} \cdot \frac{\alpha^2(1 - \alpha^{2t})}{1 - \alpha^2}. \tag{18}$$

*Proof.* From Equation (17), the update rule for $\theta_t$ is:

$$\theta_t = \theta_0 - \eta \sum_{i=0}^{t-1} \alpha^{t-i} \nabla \mathcal{L}_{\text{TTA}}(\theta_i, \mathbf{x}_i).$$

Assuming independent gradient noise across time steps with constant average variance $\bar{V}$, the variance of the sum is:

$$\text{Var}[\theta_t] \approx \eta^2 \sum_{i=0}^{t-1} \alpha^{2(t-i)} \bar{V} = \eta^2 \bar{V} \sum_{i=0}^{t-1} \alpha^{2(t-i)} = \eta^2 \bar{V} \cdot \frac{\alpha^2(1 - \alpha^{2t})}{1 - \alpha^2}.$$

Finally, taking the limit as $t \to \infty$ (and assuming $\alpha < 1$), we observe that $\alpha^{2t} \to 0$, yielding the upper bound:

$$\text{Var}[\theta_t] \lesssim \eta^2 \bar{V} \cdot \frac{\alpha^2}{1 - \alpha^2}.$$

Thus, the variance remains bounded uniformly in $t$.

Additionally, consider the limit $\alpha \to 1$. In this case, the denominator approaches $0$ and we can expand the geometric sum:

$$\lim_{\alpha \to 1} \frac{\alpha^2(1 - \alpha^{2t})}{1 - \alpha^2} = \lim_{\alpha \to 1} \frac{1 - \alpha^{2t}}{1 - \alpha^2} \approx t,$$

which recovers the standard linear-in-$t$ variance growth:

$$\text{Var}[\theta_t] \approx \eta^2 \bar{V} \cdot t.$$

This confirms that weight ensembling curtails variance growth over time compared to unregularized adaptation. $\qquad\square$

**Proposition 4** (Fisher Regularization as Weighted Ensembling). *Let the TTA objective be augmented with Fisher regularization as follows:*

$$\mathcal{L}_{TTA\text{-}fis}(\theta, \mathbf{x}) = \mathcal{L}_{TTA}(\theta, \mathbf{x}) + \lambda \cdot \omega \cdot (\theta - \theta_0)^2, \tag{19}$$

*where $\lambda > 0$ is the regularization coefficient and $\omega$ is the (diagonal) Fisher information weight. Then the gradient descent update becomes:*

$$\begin{aligned}
\theta_t &= \theta_{t-1} - \eta \nabla \mathcal{L}_{TTA\text{-}fis}(\theta_{t-1}, \mathbf{x}_{t-1}) \\
&= \theta_{t-1} - \eta \nabla \mathcal{L}_{TTA}(\theta_{t-1}, \mathbf{x}_{t-1}) - 2\lambda\omega\eta(\theta_{t-1} - \theta_0) \\
&= (1 - 2\lambda\omega\eta) \cdot \theta_{t-1} - \eta \nabla \mathcal{L}_{TTA}(\theta_{t-1}, \mathbf{x}_{t-1}) + 2\lambda\omega\eta \cdot \theta_0.
\end{aligned} \tag{20}$$

*Define $\alpha = 1 - 2\lambda\omega\eta$. Then this update is equivalent to the weighted ensembling rule:*

$$\theta_t = \alpha \cdot \hat{\theta}_t + (1 - \alpha) \cdot \theta_0, \qquad \text{where} \quad \hat{\theta}_t = \theta_{t-1} - \eta \nabla \mathcal{L}_{TTA}(\theta_{t-1}, \mathbf{x}_{t-1}). \tag{21}$$

*Proof.* Starting from the Fisher-regularized objective, the gradient is:

$$\nabla \mathcal{L}_{\text{TTA-fis}}(\theta, \mathbf{x}) = \nabla \mathcal{L}_{\text{TTA}}(\theta, \mathbf{x}) + 2\lambda\omega(\theta - \theta_0).$$

Substituting into the gradient update rule:

$$\theta_t = \theta_{t-1} - \eta \left[ \nabla \mathcal{L}_{\text{TTA}}(\theta_{t-1}, \mathbf{x}_{t-1}) + 2\lambda\omega(\theta_{t-1} - \theta_0) \right],$$

which simplifies to:

$$\theta_t = (1 - 2\lambda\omega\eta) \cdot \theta_{t-1} - \eta \nabla \mathcal{L}_{\text{TTA}}(\theta_{t-1}, \mathbf{x}_{t-1}) + 2\lambda\omega\eta \cdot \theta_0.$$

Letting $\alpha = 1 - 2\lambda\omega\eta$, we get:

$$\theta_t = \alpha \cdot \theta_{t-1} - \eta \nabla \mathcal{L}_{\text{TTA}}(\theta_{t-1}, \mathbf{x}_{t-1}) + (1 - \alpha) \cdot \theta_0.$$

Now define $\hat{\theta}_t = \theta_{t-1} - \eta \nabla \mathcal{L}_{\text{TTA}}(\theta_{t-1}, \mathbf{x}_{t-1})$, and substitute:

$$\theta_t = \alpha \cdot \hat{\theta}_t + (1 - \alpha) \cdot \theta_0,$$

which matches the ensembling formulation in Equation (16). $\qquad\square$

**Remark 1** (Bias-Variance Tradeoff). *The source-weighted ensembling update controls the variance of $\theta_t$ over time, as shown in Equation (18), but introduces bias toward the source model $\theta_0$. In the extreme case $\alpha = 0$, no adaptation occurs, resulting in maximal bias. As $\alpha \to 1$, the method recovers standard SGD, minimizing bias but allowing unbounded variance. In practice, setting $\alpha$ close to 1 balances low variance with limited bias.*

**Proof of Theorem 1** (Bound on Divergence Probability). *Let $\theta_t$ denote the model parameters at time $t$, and let $\theta_{Task}^*$ be the task-specific optimum. Suppose that $\mathbb{E}[\theta_t] \to \theta_{Task}^*$ and $\|\mathbb{E}[\theta_t] - \theta_{Task}^*\| < \|\theta_0 - \theta_{Task}^*\|$. Then for any threshold $\beta > \|\theta_0 - \theta_{Task}^*\|$, the probability of divergence from the stability region is bounded by:*

$$\Pr\big[\|\theta_t - \theta_{Task}^*\| > \beta\big] \;\leq\; \frac{\mathrm{Var}[\theta_t]}{\big(\beta - \|\theta_0 - \theta_{Task}^*\|\big)^2}. \tag{22}$$

*Proof.* We begin by decomposing the distance between $\theta_t$ and $\theta_{\mathrm{Task}}^*$:

$$\|\theta_t - \theta_{\mathrm{Task}}^*\| = \|\theta_t - \mathbb{E}[\theta_t] + \mathbb{E}[\theta_t] - \theta_{\mathrm{Task}}^*\|$$
$$\leq \|\theta_t - \mathbb{E}[\theta_t]\| + \|\mathbb{E}[\theta_t] - \theta_{\mathrm{Task}}^*\| \quad \text{(triangle inequality).} \tag{23}$$

By the assumption that $\|\mathbb{E}[\theta_t] - \theta_{\mathrm{Task}}^*\| < \|\theta_0 - \theta_{\mathrm{Task}}^*\|$, we have:

$$\Pr\big[\|\theta_t - \theta_{\mathrm{Task}}^*\| > \beta\big] \leq \Pr\left[\|\theta_t - \mathbb{E}[\theta_t]\| + \|\mathbb{E}[\theta_t] - \theta_{\mathrm{Task}}^*\| > \beta\right]$$
$$< \Pr\left[\|\theta_t - \mathbb{E}[\theta_t]\| + \|\theta_0 - \theta_{\mathrm{Task}}^*\| > \beta\right] \tag{24}$$
$$= \Pr\left[\|\theta_t - \mathbb{E}[\theta_t]\| > \beta - \|\theta_0 - \theta_{\mathrm{Task}}^*\|\right]. \tag{25}$$

Now apply Chebyshev's inequality:

$$\Pr\left[\|\theta_t - \mathbb{E}[\theta_t]\| > \beta - \|\theta_0 - \theta_{\mathrm{Task}}^*\|\right] \leq \frac{\mathrm{Var}[\theta_t]}{\big(\beta - \|\theta_0 - \theta_{\mathrm{Task}}^*\|\big)^2}. \tag{26}$$

This completes the proof. $\qquad\square$

### B.2 Comparison of Single Model TTA and Model Reservoir TTA

In Figure 5, we compare the adaptation trajectories of a single model TTA approach and our proposed Model Reservoir TTA framework under recurring continual domain shifts. Consider an example with three domains, $\mathcal{D}_1$, $\mathcal{D}_2$, and $\mathcal{D}_3$, each associated with stability radii $\beta_1$, $\beta_2$, and $\beta_3$, respectively. Let $\theta_{\mathrm{Task}_i}^* = \arg\min_\theta \mathcal{L}_{\mathrm{Task}_i}(\theta)$ for $i = 1, 2, 3$ denote the task-optimal parameters that minimize the latent task loss in each domain. If the shift between optimal parameters exceeds the stability radius for a transition,

$$\|\theta_{\mathrm{Task}_i}^* - \theta_{\mathrm{Task}_{i+1}}^*\| > \beta_{i+1} \quad \text{for } i = 1, 2,$$

then adaptation on $\mathcal{D}_1$ yields parameters near $\theta_{\mathrm{Task}_1}^*$, but a subsequent shift to $\mathcal{D}_2$ may cause the adapted parameters to drift outside the stability region of $\mathcal{D}_2$. This drift is compounded in a recurring continual TTA setting, where the test stream follows $\mathcal{D}_1 \to \mathcal{D}_2 \to \mathcal{D}_3 \hookrightarrow \mathcal{D}_1$. The left panel of Figure 5 illustrates that even with sample filtering and weight ensembling, a single model TTA approach accumulates error at each domain transition and ultimately fails to re-adapt properly (i.e., the trajectory drifts away from the $\mathcal{D}_1$ optimum, converging instead near the source model). In contrast, the right panel demonstrates that our Model Reservoir TTA framework maintains separate, domain-specific trajectories that remain bounded within their respective stability regions, thereby enabling efficient re-adaptation when a previously encountered domain reoccurs.

## C  Algorithm of ReservoirTTA

For clarity, we detail the complete ReservoirTTA workflow in Algorithm 1.

## D  Style Features Quality

In this section, we comprehensively evaluate the quality of style feature representations for ReservoirTTA by conducting quantitative ablation studies on VGG19 layer configurations and various style feature operations, and by presenting qualitative t-SNE [38] visualizations of the extracted features.

**VGG-19 Configuration: Shallow vs. Deep.** Table 5 reports an ablation study on the impact of various VGG19 layer configurations for the extraction of style features. VGG19 is organized into four hierarchical blocks from which we extract style features at specific layers: Shallow ([2, 5, 7]), Middle-1 ([10, 12, 14, 16]), Middle-2 ([19, 21, 23, 25]), and Deep ([28, 30, 32, 34]), where the indices denote

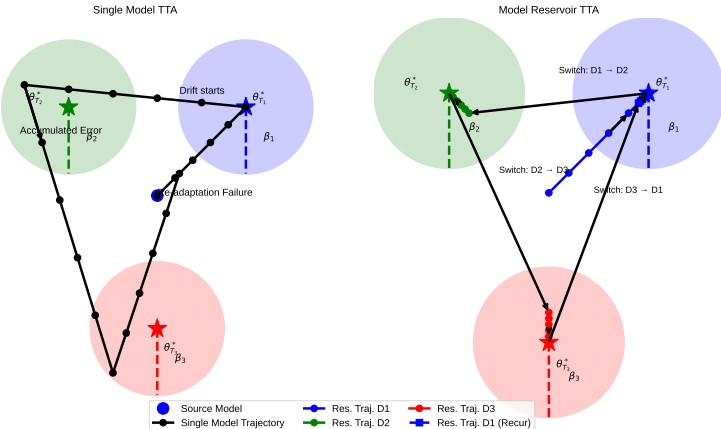

Figure 5: **Comparison of single model TTA vs. ReservoirTTA.** (Left) A single model TTA approach experiences error accumulation and drift when transitioning between domains; specifically, when $|\theta^*_{\text{Task}_i} - \theta^*_{\text{Task}_{i+1}}| > \beta_{i+1}, \quad i = 1, 2$, the model fails to re-adapt to $\mathcal{D}_1$ and its trajectory drifts away from the $\mathcal{D}_1$ optimum. (Right) In contrast, the Model Reservoir TTA framework maintains separate, domain-specific models, ensuring that each adaptation trajectory remains bounded within the stability region, thereby enabling efficient re-adaptation in recurring scenarios.

Table 5: **Ablation study on layer configurations for style feature extraction in ReservoirTTA.** The reported values represent the average classification error rate (%). Best results are shown in **bold**.

| Layer Configuration | Recurring CSC CIFAR100-C | | | Recurring CDC CIFAR100-C | | | CCC ImageNet-C | | |
|---|---|---|---|---|---|---|---|---|---|
| | *Recurring visit* ⟶ | | | *Recurring visit* ⟶ | | | *Adaptation Step* ⟶ | | |
| | 1 | 10 | 20 | 1 | 10 | 20 | 6.7k | 40.2k | 80k |
| **EATA+*ReservoirTTA*** | | | | | | | | | |
| Shallow | 30.56 | **28.39** | 28.38 | **30.44** | **28.39** | **28.40** | 61.57 | 58.16 | **58.09** |
| Middle-1 | 30.54 | 28.48 | 28.46 | 30.50 | 28.46 | 28.44 | 61.55 | 58.11 | 58.14 |
| Middle-2 | 30.67 | 28.92 | 28.75 | 30.62 | 29.02 | 28.84 | 61.78 | 58.43 | 58.43 |
| Deep | 30.73 | 28.88 | 28.65 | 30.60 | 28.77 | 28.60 | 61.82 | 58.94 | 59.28 |
| Mixed-1 | **30.51** | 28.47 | 28.44 | 30.43 | 28.44 | 28.45 | 61.46 | **58.03** | 57.92 |
| Mixed-2 | 30.83 | 28.61 | 28.56 | 30.52 | 28.63 | 28.55 | 61.94 | 58.56 | 58.52 |
| Mixed-3 | 30.56 | 28.51 | 28.48 | 30.48 | 28.58 | 28.49 | 61.42 | 58.26 | 58.35 |
| Mixed-4 | 30.57 | 28.41 | **28.36** | 30.49 | 28.49 | 28.43 | **61.13** | 58.15 | 58.11 |
| Avg. | $30.62_{\pm0.10}$ | $28.58_{\pm0.19}$ | $28.51_{\pm0.13}$ | $30.51_{\pm0.06}$ | $28.60_{\pm0.20}$ | $28.53_{\pm0.13}$ | $61.58_{\pm0.24}$ | $58.33_{\pm0.28}$ | $58.34_{\pm0.40}$ |
| **ROID*+*ReservoirTTA*** | | | | | | | | | |
| Shallow | 29.60 | **27.80** | **27.80** | 29.62 | **27.84** | 27.78 | 62.22 | 58.97 | 59.27 |
| Middle-1 | 29.65 | 27.84 | 27.83 | 29.64 | 27.86 | 27.83 | 62.19 | 58.94 | 59.30 |
| Middle-2 | 29.72 | 28.54 | 28.37 | 29.77 | 28.46 | 28.29 | 62.34 | 59.05 | 59.38 |
| Deep | 29.81 | 28.19 | 27.97 | 29.74 | 28.17 | 27.95 | 62.19 | 59.06 | 59.52 |
| Mixed-1 | **29.54** | 27.82 | 27.81 | **29.58** | 27.88 | 27.82 | **62.04** | **58.87** | **59.22** |
| Mixed-2 | 29.84 | 28.17 | 28.04 | 29.69 | 28.30 | 28.00 | 62.26 | 59.00 | 59.42 |
| Mixed-3 | 29.66 | 27.85 | 27.86 | 29.67 | 27.99 | 27.86 | 62.27 | 58.91 | 59.25 |
| Mixed-4 | 29.66 | 27.84 | 27.81 | 29.66 | 27.89 | 27.86 | 62.10 | 58.88 | 59.24 |
| Avg. | $29.68_{\pm0.10}$ | $28.00_{\pm0.25}$ | $27.94_{\pm0.18}$ | $29.67_{\pm0.06}$ | $28.05_{\pm0.22}$ | $27.92_{\pm0.15}$ | $62.20_{\pm0.09}$ | $58.96_{\pm0.07}$ | $59.33_{\pm0.10}$ |

the corresponding layers in the network. Additionally, we consider mixed configurations that integrate layers across different depths: Mixed-1 (Shallow + Middle-1), Mixed-2 (Deep + Middle-2), Mixed-3 (Shallow + Middle-1 + Middle-2 + Deep), and Mixed-4 (Shallow + Deep). We evaluated the four main configurations above (Shallow, Middle-1, Middle-2, Deep) along with several mixed configurations on three scenarios: recurring CSC and CDC on CIFAR100-C and CCC on ImageNet-C. The low standard deviations in the average error rates confirm that ReservoirTTA is robust to the choice of intermediate VGG19 layers. Table 6 reports the ablation study on style feature representations in ReservoirTTA. The average classification error rates (%) for five different operations—`mean`, `var`, `logvar`, `[mean, var]`, and `gram`—are evaluated. Notably, our `logvar`-based style representation delivers superior performance compared to the other operations.

**Style Extractor Choice: VGG-19 vs. Source.** We investigate whether using the source model for style feature extraction could replace the ImageNet-trained VGG. Following this suggestion, we replaced VGG with early layers of the source backbone for style calculation. On CIFAR100-C

---

**Algorithm 1** ReservoirTTA: Prolonged Test-Time Adaptation

---

**Require:** Pre-trained source model $f_{\theta_0}$; maximum domains $K^{\mathbf{max}}$; style reservoir size $M$; new-domain threshold $\tau$ (initialized using source examples); TTA objective $\mathcal{L}_{\mathrm{TTA}}(\cdot)$

**Ensure:** Updated model reservoir $\{\theta_t^1, \ldots, \theta_t^{K^{\mathrm{max}}}\}$ and final predictions for each batch

1: **Initialization:**
2: Initialize style centroids $\{c^1\} \leftarrow$ average source style features
3: Initialize model reservoir $\{\theta^1\} \leftarrow \theta_0$
4: Initialize style reservoir $R_0 \leftarrow \varnothing$           Capacity $= M$
5: Set current reservoir size $K_0 \leftarrow 1$

6: **for** each incoming test batch $\mathbf{B}_t = \{x_t^i\}_{i=1}^b$ **do**
7:     **(1) Domain Identification:**
8:     Extract style features from $\mathbf{B}_t$ and compute style vector $s_t$
9:     Update Style Reservoir $R_t$ via Reservoir Sampling
10:     **if** $|R_t| < M$ **then**
11:         Insert $s_t$ into $R_t$
12:     **else**
13:         **if** `rand()` $\leq M/t$ **then**
14:             Randomly replace an element in $R_t$ with $s_t$     Replace an element with $s_t$ with appropriate probability
15:         **else**
16:             Reject $s_t$          Do Nothing
17:         **end if**
18:     **end if**
19:     Compute distance $\Delta = \min_{1 \leq k \leq K_t} \|s_t - c^k\|$      See Equation (7)
20:     **if** $\Delta > \tau$ **and** $K_t < K^{\mathrm{max}}$ **then**
21:         $K_t \leftarrow K_t + 1$          New domain detected
22:         Set $c^{K_t} \leftarrow s_t$
23:         Initialize $\theta^{K_t} \leftarrow \arg\min_{\theta \in \{\theta^1, \ldots, \theta^{K_t-1}\}} \mathcal{L}_{\mathrm{MI}}\big(f_\theta(\mathbf{B}_t)\big)$ Select model with highest prediction mutual information with respect to uniform distribution U
24:     **end if**
25:     **(2) Update Style Centroids:**
26:     Compute soft assignment matrix $Q_t \in \mathbb{R}^{M \times K_t}$      See Equation (8)
27:     **for** $k = 1$ to $K_t$ **do**
28:         Update $c^k$ by gradient descent on $\mathcal{L}_{\mathrm{MI}}(Q_t)$      See Equation (10)
29:     **end for**
30:     **(3) Model Reservoir Update:**
31:     Compute soft assignment vector $q_t \in \mathbb{R}^{K_t}$
32:     Let $k^* = \arg\max_{1 \leq k \leq K_t} [q_t]_k$
33:     Update $\theta^{k^*} \leftarrow \theta^{k^*} - \eta \nabla_{\theta^{k^*}} \mathcal{L}_{\mathrm{TTA}}(\mathbf{B}_t, \theta^{k^*})$      Adaptation step for the selected domain
34:     **(4) Model Ensembling for Prediction:**
35:     Compute $\bar{\theta}_t = \sum_{k=1}^{K_t} [q_t]_k \, \theta^k$      Weighted ensembling over domain-specific models
36:     Predict $\hat{y}_t = f_{\bar{\theta}_t}(\mathbf{B}_t)$
37: **end for**
38: **return** $\{\theta^k\}_{k=1}^{K_t}$ and predictions $\hat{y}_t$

---

(CSC) with a ResNeXt-29 backbone, this swap increases mean error by +2.12% (Table 7). Source backbones lack the broad texture priors learned from ImageNet and would make our approach architecture-specific. By contrast, VGG delivers consistent gains across ResNeXt-29, ViT-B-16, and ResNet-50 (GN) (see Appendix K). These results highlight that the VGG-19 style extractor with the automatically chosen threshold $\tau$ introduces minimal, dataset-agnostic overhead while providing substantially better domain detection and adaptation performance.

**Style Features Quality.** Figure 6a presents the t-SNE visualization of style features extracted from three corruption benchmarks—CIFAR-10-C, CIFAR-100-C, and ImageNet-C—using our ReservoirTTA method. The datasets cover 15 distinct domains, which we organize into four categories: *Noise* (Gaussian Noise, Shot Noise, Impulse Noise), *Blur* (Defocus Blur, Glass Blur, Motion Blur, Zoom Blur), *Weather* (Snow, Frost, Fog, Brightness), and *Digital* (Contrast, Elastic, Pixelate, JPEG). As observed, samples from the same domain form well-defined clusters, while features from different domains remain clearly separated. This result confirms that style features effectively capture the inherent domain differences in these challenging corruption scenarios. In Figure 6b, we compare

Table 6: **Ablation study on style feature representation in ReservoirTTA.** Average classification error rate (%) is reported. We compare five style feature representations: `mean`, `var`, `logvar`, `gram` (diagonal of Gram matrix), and `[mean, var]` (concatenated). Features are computed over batch, height, and width. **Bold** indicates best results.

| | Recurring CSC | | | Recurring CDC | | | CCC | | |
|---|---|---|---|---|---|---|---|---|---|
| | CIFAR100-C | | | CIFAR100-C | | | ImageNet-C | | |
| | *Recurring visit* ⟶ | | | *Recurring visit* ⟶ | | | *Adaptation Step* ⟶ | | |
| **Representation** | **1** | **10** | **20** | **1** | **10** | **20** | **6.7k** | **40.2k** | **80k** |
| **EATA+*ReservoirTTA*** | | | | | | | | | |
| mean | 30.62 | 28.45 | 28.42 | 30.52 | 28.40 | 28.46 | 62.07 | 58.30 | 58.51 |
| var | 30.56 | 28.51 | 28.50 | 30.53 | 28.48 | 28.52 | 61.94 | 58.59 | 58.99 |
| [mean, var] | 30.59 | 28.45 | 28.40 | 30.51 | 28.39 | 28.44 | 62.50 | 58.28 | 58.44 |
| gram | 30.61 | 28.48 | 28.47 | 30.53 | 28.49 | 28.57 | 62.13 | 58.76 | 59.38 |
| logvar | **30.56** | **28.39** | **28.38** | **30.44** | **28.39** | **28.40** | **61.57** | **58.16** | **58.09** |
| **ROID*+*ReservoirTTA*** | | | | | | | | | |
| mean | 29.69 | 27.83 | 27.84 | 29.71 | 27.86 | 27.84 | 61.42 | 57.95 | 58.45 |
| var | 29.64 | 27.86 | 27.88 | 29.69 | 27.89 | 27.84 | 61.24 | 58.04 | 58.41 |
| [mean, var] | 29.62 | 27.83 | 27.84 | 29.68 | 27.86 | 27.82 | 61.31 | 57.94 | **58.40** |
| gram | 29.60 | 27.85 | 27.88 | 29.66 | 27.89 | 27.85 | 61.42 | 58.07 | 58.47 |
| logvar | **29.60** | **27.80** | **27.80** | **29.62** | **27.84** | **27.78** | **61.24** | **57.94** | 58.41 |

Table 7: Effect of using the source backbone (w/o VGG) vs. frozen VGG-19 for style feature extraction on CIFAR-100-C (CSC) with ResNeXt-29. Error (%) across 20 recurrences.

| | *Visits 1–10* | | | | | | | | | |
|---|---|---|---|---|---|---|---|---|---|---|
| **Method** | **1** | **2** | **3** | **4** | **5** | **6** | **7** | **8** | **9** | **10** |
| EATA | 30.51 | 30.29 | 30.39 | 30.39 | 30.45 | 30.47 | 30.38 | 30.44 | 30.47 | 30.53 |
| *+ReservoirTTA w/o VGG* | 31.62 | 30.86 | 30.75 | 30.77 | 30.64 | 30.67 | 30.64 | 30.70 | 30.68 | 30.64 |
| *+ReservoirTTA* | 30.56 | 29.07 | 28.75 | 28.58 | 28.52 | 28.46 | 28.41 | 28.41 | 28.39 | 28.39 |

| | *Visits 11–20* | | | | | | | | | | |
|---|---|---|---|---|---|---|---|---|---|---|---|
| **Method** | **11** | **12** | **13** | **14** | **15** | **16** | **17** | **18** | **19** | **20** | **Avg** |
| EATA | 30.51 | 30.46 | 30.47 | 30.51 | 30.51 | 30.48 | 30.51 | 30.54 | 30.47 | 30.47 | 30.46 |
| *+ReservoirTTA w/o VGG* | 30.65 | 30.65 | 30.61 | 30.60 | 30.61 | 30.61 | 30.56 | 30.53 | 30.52 | 30.51 | 30.69 |
| *+ReservoirTTA* | 28.38 | 28.40 | 28.37 | 28.37 | 28.37 | 28.40 | 28.36 | 28.43 | 28.37 | 28.38 | **28.57** |

the feature distributions produced by our method, ReservoirTTA, against two baseline approaches, DPCore and CoLA, under the recurring CSC setting in the CIFAR-100-C dataset. We show the evolution of the t-SNE plots at the first, 10th, and 20th visits, illustrating how the features adapt as domain shifts accumulate over time. Our method consistently maintains well-separated and dense clusters throughout the adaptation process, while the baselines exhibit less distinct clustering. This comparison further demonstrates the effectiveness of ReservoirTTA in maintaining domain distinctions even in evolving environments.

# E  Domain Identification via Online Clustering

**Effect of Style Reservoir Capacity.** The capacity of style reservoir $M$ determines how many style features are used to optimize the style centroids in Equation (9). As shown in Figure 7, in both recurring CSC and CDC settings, performance remains stable across different values of $M$, with low sensitivity to changes in this parameter. This suggests that even with an extremely small style reservoir size (e.g., $M = 1$), our style-based domain assignment optimization remains robust and efficient, without incurring significant memory overhead. In CCC, increasing $M$ leads to a reduction

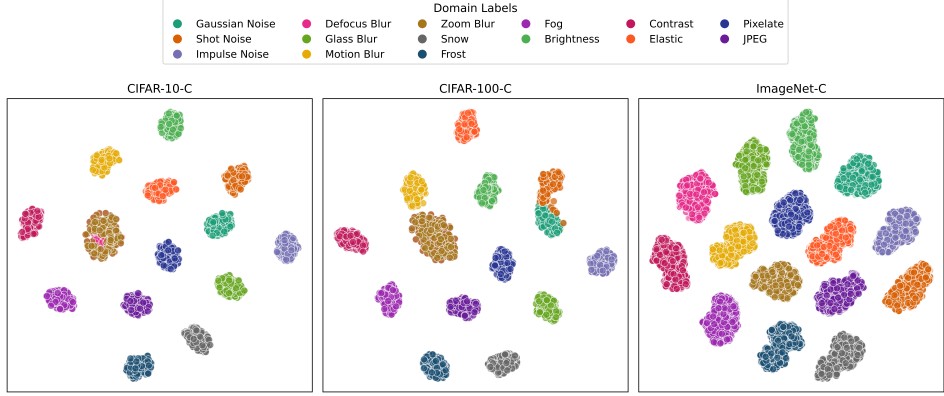

(a) **t-SNE visualization of style features.** Style features from CIFAR-10-C, CIFAR-100-C, and ImageNet-C (15 corruption domains) form distinct clusters, showing effective domain separation.

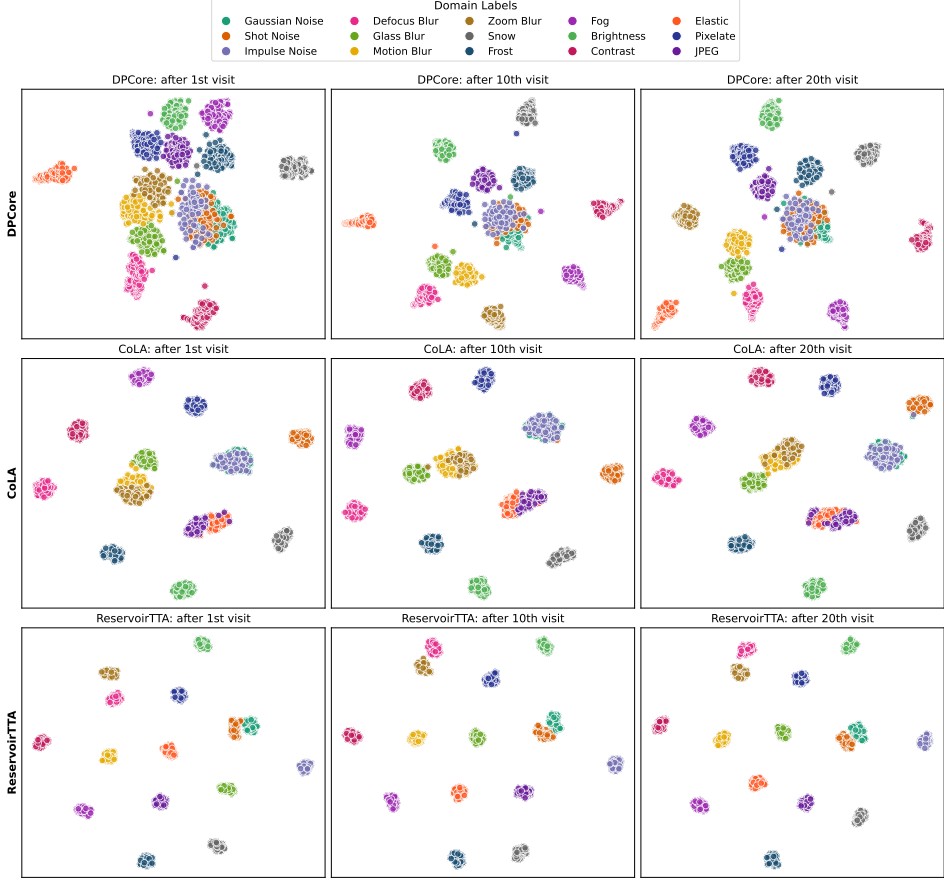

(b) **t-SNE comparison in recurring CSC.** ReservoirTTA is compared with DPCore and CoLA at visits 1, 10, and 20.

Figure 6: **t-SNE visualization and comparison.** (a) Style features form distinct clusters, and (b) ReservoirTTA maintains well-separated clusters over time in the recurring CSC setting.

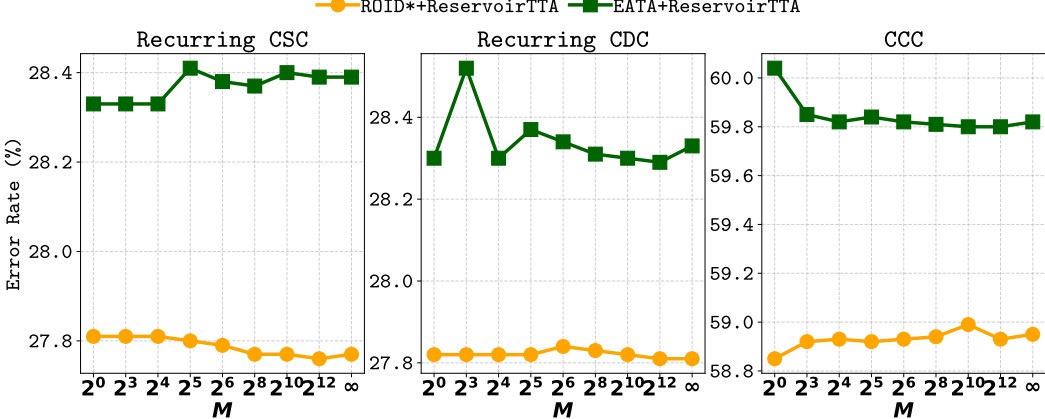

Figure 7: **Sensitivity study to size of the Style Reservoir** $M$ on CIFAR-100-C under recurring CSC and CDC settings, and on CCC.

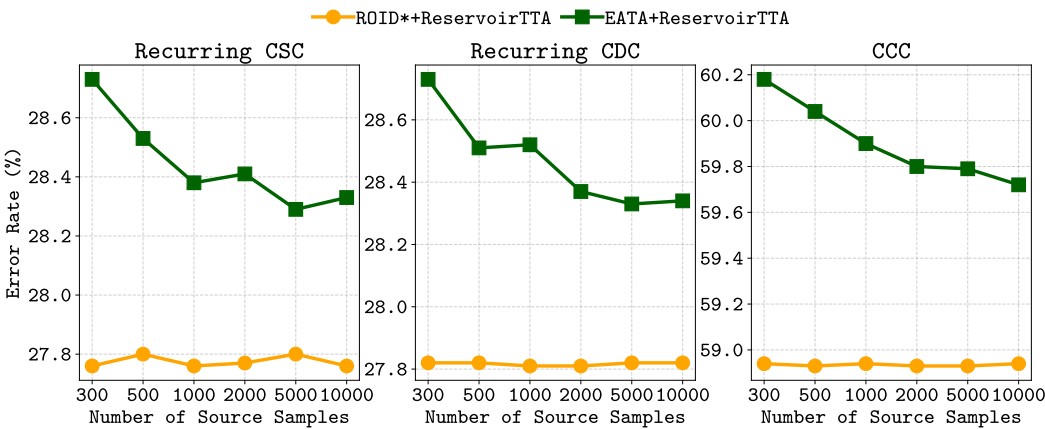

Figure 8: **Sensitivity study to the number of source examples used to compute the threshold** $\tau$ on CIFAR-100-C under recurring CSC and CDC settings, and on CCC.

in errors, confirming our ablation study analysis that a larger style reservoir allows for a more comprehensive observation of domain shifts, thereby improving performance. We set $M = 1024$ for all main experiments.

**Effect of Number of Source Samples and Quantile Value.** In Figure 8, we vary the number of unlabeled source samples from 300 to 10,000. For ROID*+ReservoirTTA, the error remains nearly constant, while for EATA+ReservoirTTA a slight decrease in error is observed with more source samples. In addition, minor fluctuations—especially in the recurring CSC and CDC settings.

On the choice of the quantile threshold $q_{\text{th}}$: lowering $q_{\text{th}}$ increases sensitivity—detecting smaller shifts—but also raises false positives and spawns unnecessary domains; higher $q_{\text{th}}$ is more conservative and can delay detection of genuinely new domains. In a single sweep over the 15 CIFAR-100-C corruptions, $q_{\text{th}} = 0.5$ severely overestimates (detects $> 100$ domains), whereas $q_{\text{th}} = 1.0$ merges similar corruptions—[zoom, motion, glass blur] and [Gaussian, shot, impulse noise]—yielding 9 domains. Intermediate thresholds balance sensitivity and precision: $q_{\text{th}} = 0.99$ identifies 17 domains and $q_{\text{th}} = 0.999$ identifies 14. Importantly, Figure 9 shows that varying $q_{\text{th}}$ leads to only minor changes in error. Overall, our style-based domain identifier is robust to both the quantile setting and the number of source samples, with minimal impact on performance.

**Ablation on Online Clustering Strategies.** Table 8 compares different online clustering strategies (Sinkhorn–Knopp [1], Online K-Means, and our adaptive clustering scheme used for reservoirTTA) when integrated into ROID* as the baseline. We evaluate recurring CSC and CDC on CIFAR-100-C,

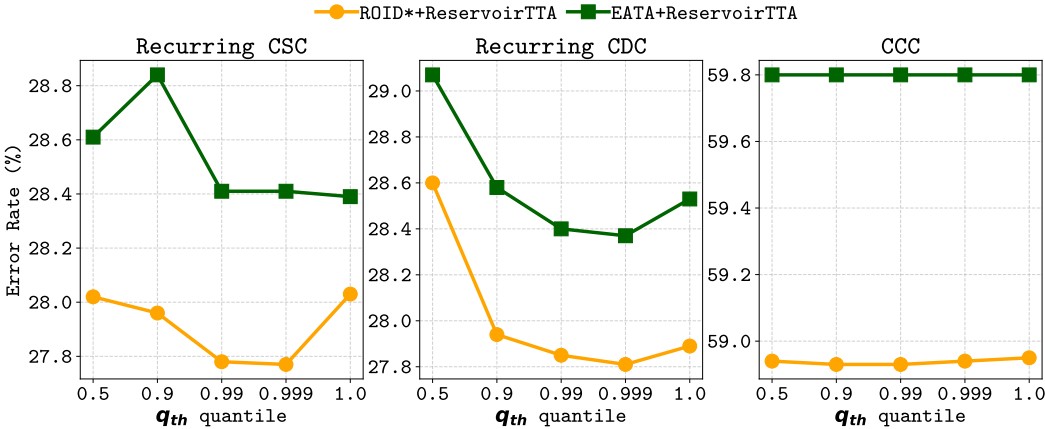

Figure 9: **Sensitivity study to the** $q_{th}$ **quantile used for threshold** $\tau$ **computation** on CIFAR-100-C under recurring CSC and CDC settings, and on CCC.

Table 8: **Ablation study on online clustering in ReservoirTTA.** Average classification error rates (%) are reported using ROID* as the baseline. Besides our reservoirTTA online clustering strategy, we also evaluate the online Sinkhorn-Knopp algorithm [1] and online K-Means for style clustering and domain assignment. Best results are highlighted in **bold**.

| | Recurring CSC | | | Recurring CDC | | | CCC | | |
|---|---|---|---|---|---|---|---|---|---|
| | CIFAR100-C | | | CIFAR100-C | | | ImageNet-C | | |
| | *Recurring visit* $\longrightarrow$ | | | *Recurring visit* $\longrightarrow$ | | | *Adaptation Step* $\longrightarrow$ | | |
| **Online clustering** | **1** | **10** | **20** | **1** | **10** | **20** | **6700** | **40200** | **80000** |
| *Baseline* | **29.45** | 29.20 | 29.25 | 30.17 | 30.08 | 30.12 | 61.89 | 58.33 | 58.76 |
| Sinkhorn | 29.78 | 27.81 | 27.80 | 29.82 | **27.83** | 27.80 | 63.35 | 70.01 | 71.73 |
| Online K-Means | 29.60 | 27.83 | 27.84 | **29.61** | 27.87 | 27.82 | 61.97 | 58.43 | 59.79 |
| **Ours** | 29.60 | **27.80** | **27.80** | 29.62 | 27.84 | **27.78** | **61.24** | **57.94** | **58.41** |

and CCC on ImageNet-C, reporting average classification error rates. Across all settings, our method consistently achieves the lowest error. In particular, for recurring CSC, our approach shows the best performance at visits 10 and 20, highlighting accurate domain partitioning and model assignment over multiple rounds. Sinkhorn–Knopp and Online K-Means provide moderate improvements but still trail behind our reservoir-based strategy. These findings confirm that our online clustering mechanism effectively separates target domains and assigns them to specialized models, leading to superior long-term adaptation.

**Ablation on Distance Metrics.** We investigate alternative metrics for centroid assignment within EATA+ReservoirTTA. Euclidean distance aligns naturally with our quantile-based new-domain detector $\tau$, yet cosine similarity and KL divergence might, in principle, offer advantages in high-dimensional or streaming regimes. To assess this, we compare soft assignments using Euclidean distance, cosine similarity, and KL divergence on CIFAR-100-C (CSC) over 20 recurrences. As summarized in Table 9, KL divergence outperforms cosine similarity—supporting its suitability for online adaptation—while Euclidean distance computed on log-variance features consistently achieves the lowest error across recurrences. These findings validate our choice to retain Euclidean distance and provide further empirical evidence of the method's robustness, alongside a clear head-to-head comparison of distance metrics.

**Model Assignments per Domain.** Figure 10 displays a heatmap illustrating how each target domain is assigned to different models in ReservoirTTA. The rows represent 15 corruption types (e.g., Gauss, Shot, Impulse, Defocus), while the columns correspond to 15 model indices. Each cell $[i, j]$ denotes the proportion of samples from the $i$-th domain that are routed to the $j$-th model, with row sums normalized to 1. We observe a clear one-to-one correspondence for most corruptions, though some exhibit similar style cues. Overall, the concentration along the diagonal reflects domain-specialized

Table 9: Cosine vs. Euclidean vs. KL-divergence soft assignments for EATA+ReservoirTTA on CIFAR-100-C (CSC). Error (%) across 20 recurrences.

| Variant | Visits 1–10 | | | | | | | | | |
| | **1** | **2** | **3** | **4** | **5** | **6** | **7** | **8** | **9** | **10** |
|---|---|---|---|---|---|---|---|---|---|---|
| Cosine distance | 31.94 | 31.04 | 30.91 | 30.81 | 30.76 | 30.77 | 30.74 | 30.77 | 30.60 | 30.58 |
| Euclidean distance | 31.76 | 30.37 | 30.19 | 29.96 | 29.81 | 29.93 | 29.92 | 29.84 | 29.89 | 29.86 |
| KL divergence | 32.01 | 30.59 | 30.14 | 30.16 | 30.19 | 30.00 | 30.04 | 30.09 | 29.99 | 30.05 |

| Variant | Visits 11–20 | | | | | | | | | | |
| | **11** | **12** | **13** | **14** | **15** | **16** | **17** | **18** | **19** | **20** | **Avg** |
|---|---|---|---|---|---|---|---|---|---|---|---|
| Cosine distance | 30.62 | 30.55 | 30.63 | 30.58 | 30.59 | 30.57 | 30.53 | 30.55 | 30.51 | 30.55 | 30.73 |
| Euclidean distance | 29.97 | 29.87 | 30.02 | 30.01 | 29.98 | 30.02 | 29.93 | 30.05 | 30.06 | 30.06 | **30.07** |
| KL divergence | 30.07 | 30.02 | 30.15 | 30.15 | 30.15 | 30.16 | 30.26 | 30.30 | 30.32 | 30.40 | 30.26 |

adaptation, whereas smaller off-diagonal entries highlight moderate sharing among visually related domains. This confirms that ReservoirTTA's style-based clustering reliably assigns test samples to the most relevant domain-specific model.

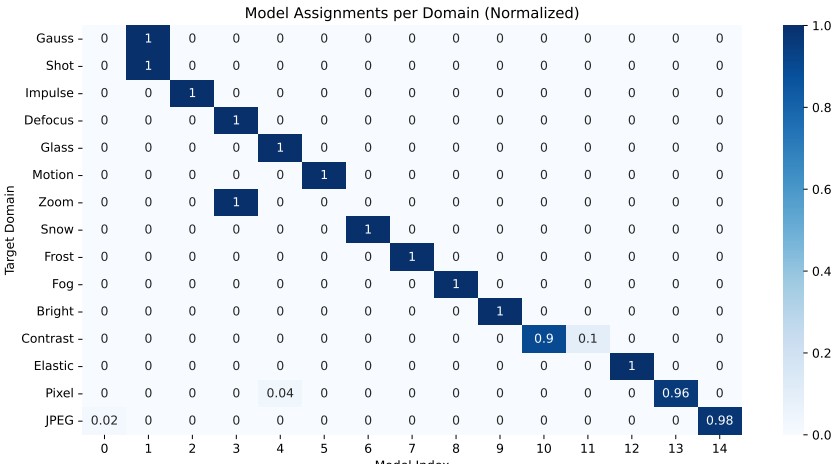

Figure 10: **Model assignments per domain**. The heatmap illustrates the assignment distribution of target domains to models in ReservoirTTA. The y-axis represents target domains, while the x-axis corresponds to model indices. Each entry $[i, j]$ denotes the proportion of samples from domain $i$ assigned to model $j$, with row sums normalized to one.

**Style Cluster Centroid Trajectories During Adaptation.** Figure 11 shows the evolution (via trajectories) of style cluster centroids during adaptation. Each centroid is assigned a unique ID, which corresponds to the model index presented in Figure 10. As shown, each centroid remains closely linked to its assigned domain throughout adaptation, indicating that each model adapts exclusively to that domain.

**FIFO vs. Reservoir Sampling for Domain Balancing.** Figure 12 compares two sampling strategies—FIFO and Reservoir sampling—for maintaining a domain-balanced Style Reservoir in recurring CSC on CIFAR-100-C. It shows the distribution of different domains over time for two reservoir sizes ($M$=256 and $M$=1024). The results reveal that Reservoir sampling produces a more uniform and stable domain distribution than FIFO, even as the reservoir size varies. These observations underscore the robustness of our approach and support the use of Reservoir sampling as the preferred update strategy.

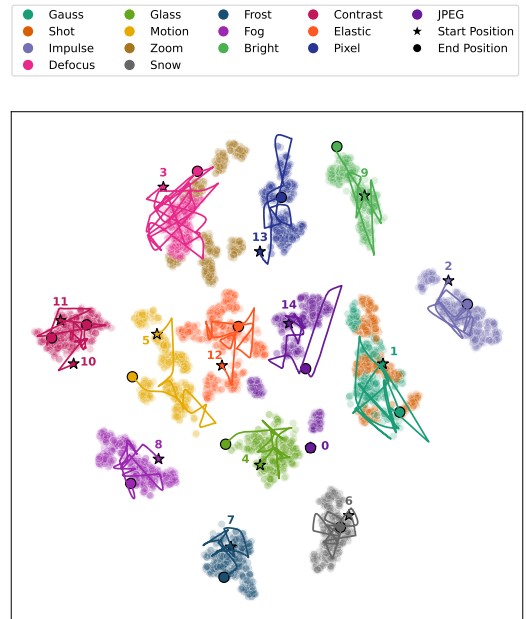

Figure 11: **t-SNE visualization of domain features and the trajectories of style cluster centroids in recurring CSC.**

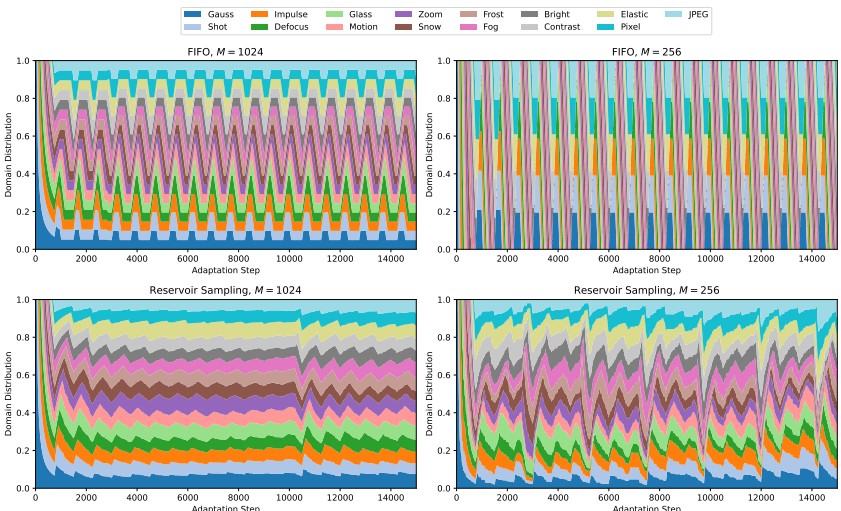

Figure 12: **Comparison of FIFO and Reservoir sampling for a domain-balanced style Reservoir in Recurring CSC on CIFAR-100-C.** The plot shows that ReservoirTTA achieves more stable domain balancing and is robust to changes in the style reservoir size ($M$). It displays the distribution of domains over time for $M = 256$ and $M = 1024$, demonstrating that our reservoir sampling yields a more uniform distribution than FIFO. This supports its use as the preferred update strategy.

## F   Model Reservoir Initialization

In this section, we explore strategies for initializing a new model when a new domain is detected.

**MI vs. Source Weight Initialization.** As shown in Figure 13, we evaluate two initialization approaches: (1) initializing with source model parameters and (2) initializing with reservoir parameters that maximize mutual information (Equation (11)) on the current batch. We evaluate these on CIFAR100-C under recurring CSC and CDC settings (20 recurrences) using ResNeXt-29, and on CCC with ResNet-50 and ImageNet-C under recurring CSC and CDC settings (20 recurrences) using

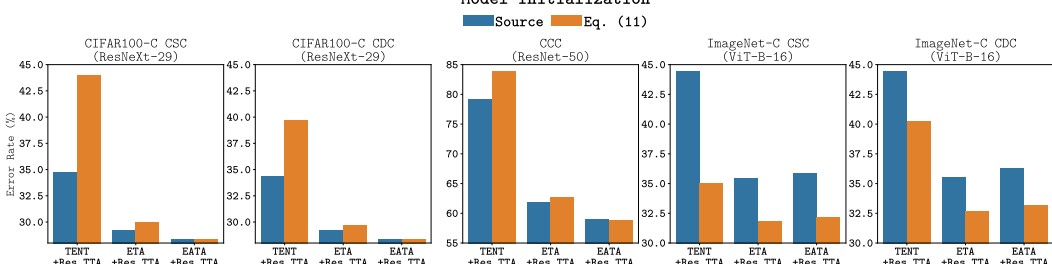

Figure 13: **Comparative study of model reservoir initializations in Recurring TTA.** We compare two initialization strategies—one using source parameters and one based on mutual information (Equation (11))—across various datasets, TTA settings, and backbones. For CSC and CDC, error rates (%) are reported at the 20th recurrence; for CCC, we report the average error rate (%) over the full dataset.

a ViT-B-16. Our results show that the optimal strategy depends on both the method and the backbone. For convolutional networks, TENT and ETA benefit from source initialization—likely because, without a regularization module, drifting too far from the source degrades performance—while EATA is largely unaffected. Conversely, for ViT-B-16, initializing via Equation (11) significantly reduces error rates after 20 recurrences for TENT, ETA, and EATA, suggesting that the transformer is less prone to parameter drift and can better leverage the most similar model in the reservoir, whereas source initialization may overwrite valuable acquired knowledge.

**MI vs Top-$k$ Model Weight Ensembling Initialization.** We assess the effect of top-$k$ ($k = 3$) ensemble initialization strategy for EATA+ReservoirTTA. On CIFAR-100-C (CSC) with a CNN backbone, we observe that EATA remains unaffected by the choice of initialization, yielding nearly identical performance across source, top-$k$, and MI initialization, consistent with our previous analysis. In contrast, on ImageNet-C with a ViT backbone, top-$k$ initialization improves performance by 3.37% compared to source initialization, but degrades performance over MI initialization (–0.27%).

# G   Baselines

Below is a summary of how the baseline TTA methods in our paper are categorized, along with a very short description of each method. Every baseline has been tested in a unified TTA repository[2] [24] (MIT license):

**1. Single-Target TTA. TENT** minimizes the entropy of prediction online.

**2. Continual TTA.** These methods are robust to continual TTA settings but may suffer performance degradation when run for a long time.

- **CoTTA***[42]: A variant of CoTTA that updates only affine parameters for a fair comparison. The CoTTA method combined weight averaging, predictions averaged over augmentations, and stochastic restoration within a mean-teacher framework. We employ the official `mmsegmentation` code provided by the authors[3] (MIT license).

- **RoTTA** [45]: Incorporates robust adaptation and reinitialization mechanisms to counteract the negative effects of prolonged adaptation.

- **ETA** [27]: Uses a sample-adaptive entropy minimization strategy that filters out test samples that are unreliable or redundant. This approach reduces the number of backward passes and error accumulation during test-time adaptation.

- **SAR** [28]: Employs sharpness-aware entropy minimization to improve reliability by mitigating the influence of noisy gradients.

---

[2]https://github.com/mariodoebler/test-time-adaptation
[3]https://github.com/qinenergy/cotta/issues/6

Table 10: **Hyperparameter settings for TTA experiments**. Following `RobustBench`, we use WideResNet-28 [46] for CIFAR-10-C, ResNeXt-29 [44] for CIFAR-100-C, and ResNet-50 [10] for ImageNet-C, DomainNet-126, and PACS. We also evaluate on ViT-B-16 [6] and ResNet-50 with GroupNorm [43] for ImageNet-C. ReservoirTTA-specific settings are provided below.

| Dataset / Backbone | Optimizer | Learning Rate | Batch Size | Notes |
|---|---|---|---|---|
| CIFAR-10-C (WideResNet-28) | Adam | $1 \times 10^{-3}$ | 200 | Except SAR [28] (SGD) |
| CIFAR-100-C (ResNeXt-29) | Adam | $1 \times 10^{-3}$ | 200 | Except SAR [28] (SGD) |
| ImageNet-C (ResNet-50) | SGD | $2.5 \times 10^{-4}$ | 64 | — |
| ImageNet-C (ViT-B-16) | SGD | $2.5 \times 10^{-4}$ | 64 | Except SAR [28] ($1 \times 10^{-4}$) |
| ImageNet-C (ResNet-50 with GroupNorm) | SGD | $2.5 \times 10^{-4}$ | 64 | — |
| CCC (ResNet-50) | SGD | $2.5 \times 10^{-4}$ | 64 | — |
| CCC (ViT-B-16) | SGD | $1 \times 10^{-4}$ | 64 | — |
| DomainNet-126 (ResNet-50) | SGD | $2.5 \times 10^{4}$ | 128 | — |
| PACS (ResNet-50) | Adam | $1 \times 10^{-3}$ | 64 | — |
| Cityscapes-to-ACDC (Segformer-B5) | Adam | $7.5 \times 10^{-6}$ | 1 | Except BECoTTA [20] ($6 \times 10^{-7}$) |
| **ReservoirTTA-specific Settings** | | | | |
| Max Reservoirs ($K^{\max}$) | — | — | — | 16 |
| Threshold ($\tau$) | — | — | — | 2000 source examples |
| Centroid Update Optimizer | AdamW | $1 \times 10^{-4}$ | — | — |
| Style Reservoir Size ($M$) | — | — | — | 1024 |
| VGG19 Layers for Style Feature Extraction | — | — | — | [2, 5, 7] |

**3. Persistent TTA.** These methods are designed for long-term stability, preventing model collapse and catastrophic forgetting across repeated domain shifts.

- **RDumb** [31]: Periodically resets model parameters to counteract error accumulation and avoid catastrophic forgetting.
- **PeTTA** [12]: Explicitly controls parameter drift during test-time adaptation by dynamically adjusting the update and regularization parameters, thereby preventing model collapse and sustaining performance over extended, recurring testing scenarios.
- **EATA** [27]: Extends ETA by incorporating a Fisher-based anti-forgetting regularizer [18] to prevent forgetting and ensure robust adaptation.
- **ROID**[*]: A modified version of ROID [24] that omits the augmentation consistency loss, enabling a fair comparison with other baselines which do not incorporate this extra regularization. Removing it isolates the impact of key components like weight ensembling and diversity weighting, which are common to the compared methods.

**4. Domain-Disentangled / Prompt-based TTA.** These methods incorporate mechanisms to disentangle domain-specific features or use visual prompts for more effective adaptation to domain shifts.

- **CoLA** [3]: Leverages collaborative adaptation by sharing domain knowledge vectors across devices for improved efficiency.
- **DPCore** [47]: Utilizes prompt-based learning by incorporating visual prompts to guide domain-specific adaptation.
- **BECoTTA** [20]: Uses input-dependent blending of experts to adapt to domain shifts, and is evaluated especially in the context of semantic segmentation.

# H   Implementation Details

Following the `RobustBench`[4] [11] (MIT license), we employ WideResNet-28 [46] on CIFAR-10-C, ResNeXt-29 [44] on CIFAR-100-C, and ResNet-50 [10] on ImageNet-C. We also evaluate on ViT-B-16 [6] and ResNet-50 with GroupNorm [43] for ImageNet-C to further assess generalizability. For CIFAR-10-C and CIFAR-100-C, TTA baselines (except SAR [28]) are optimized with the Adam optimizer [17] using a learning rate of $1 \times 10^{-3}$, a universal $\beta = (0.9, 0.999)$, and a batch size of 200, whereas SAR employs SGD [32]. For ImageNet-C, models are adapted with SGD at a batch

---

[4]`https://github.com/RobustBench/robustbench`

size of 64 and a learning rate of $2.5 \times 10^{-4}$ (adjusted to $1 \times 10^{-4}$ for ViT-B-16 in the CCC setting). For ReservoirTTA, we configure the system with a maximum of $K^{\max} = 16$ reservoirs, determine the threshold $\tau$ using 2000 source examples, and update centroids with AdamW [23] at a learning rate of $1 \times 10^{-4}$. Table 10 summarizes these settings. Concerning DomainNet-126, and PACS, we optimize the models with SGD at $2.5 \times 10^{-4}$ learning rate, and Adam at $1 \times 10^{-3}$ learning rate, respectively. Note CIFAR10-C, CIFAR100-C, and ImageNet-C are publicly available online[5] (Apache-2.0 license). CCC is also provided by Rdumb paper[6] [31] (MIT license). Both **DomainNet-126**[7] and **PACS**[8] are publicly available. All experiments were run on a single NVIDIA A100 Tensor Core GPU (80 GB VRAM) on our internal cluster.

# I    Additional Ablation Studies

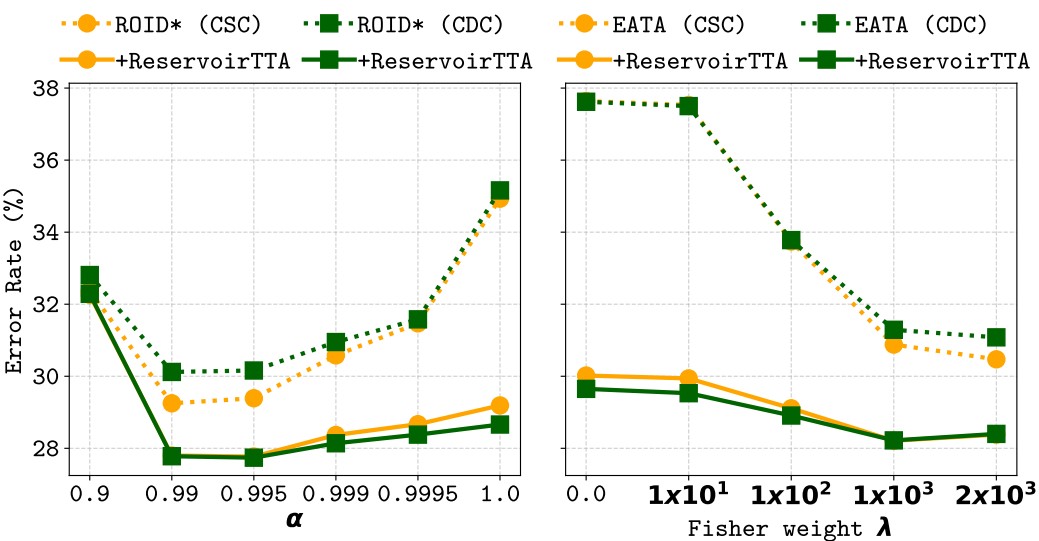

Figure 14: **Regularization weight sensitivity for ReservoirTTA with persistent TTA baselines on CIFAR-100-C Recurring CSC and CDC.** We compare the sensitivity of ROID* and ROID*+ReservoirTTA to the weight ensembling momentum parameter $\alpha$, and compare EATA and EATA+ReservoirTTA for various Fisher weight $\lambda$. Results are reported as the average classification error rate after the 20th visit, averaged over 5 seeds.

**Impact of ReservoirTTA on Regularization Weight Sensitivity of EATA and ROID*.** We analyze how ReservoirTTA affects the optimal hyperparameter $\alpha$ of ROID*, which controls the momentum in weight ensembling with the source model. We also examine its impact on Fisher weight $\lambda$, which regulates the distance to the source model parameters in EATA. As shown in Figure 14, ReservoirTTA reduces the sensitivity of both EATA and ROID* to these hyperparameters on CIFAR100-C recurring CSC and CDC settings. Furthermore, we observe that the best results with ReservoirTTA are achieved at a higher $\alpha$ for ROID*, shifting from 0.99 in ROID* to 0.995 in ROID*+ReservoirTTA. A similar trend is observed for EATA, where the optimal Fisher weight $\lambda$ decreases from 2000 in EATA to 1000 in EATA+ReservoirTTA. This effect can be attributed to the Model Reservoir, which exposes each model to a more stable distribution of domains over time, thereby reducing the need for strong regularization.

This effect is even more pronounced in ViTs. As shown in Figure 15, EATA achieves its best performance at $K^{\max}{=}1$ with a regularization weight of 2000.0, whereas ReservoirTTA ($K^{\max}{=}16$) reaches optimal performance with a regularization weight of 0. Interestingly, as $K^{\max}$ increases, the need for regularization diminishes. For ROID*, the trend differs. The momentum parameter $\alpha$

---

[5] https://github.com/hendrycks/robustness

[6] https://github.com/oripress/CCC

[7] https://ai.bu.edu/M3SDA/

[8] https://huggingface.co/datasets/flwrlabs/pacs

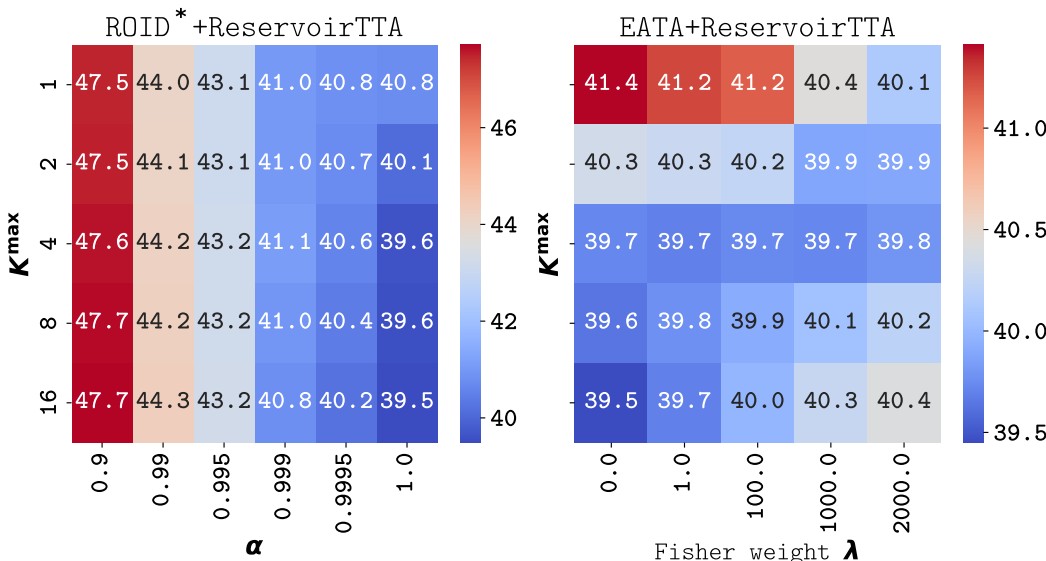

Figure 15: **Regularization weight sensitivity for ReservoirTTA on CCC using ViT-B-16.** Classification error rates (%) on the CCC benchmark are reported for varying values of $K^{max}$ and $\alpha$ for ROID*+ReservoirTTA, and for varying values of $K^{max}$ and Fisher weight $\lambda$ for EATA+ReservoirTTA.

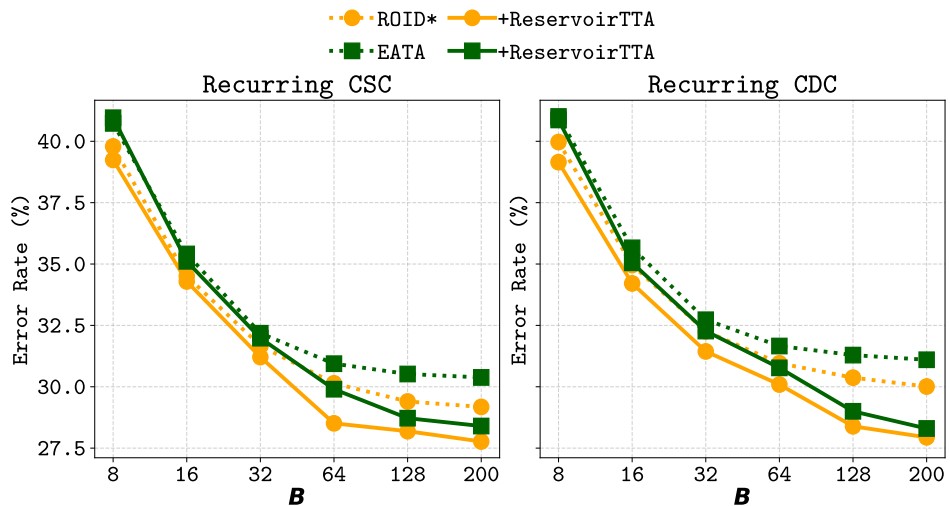

Figure 16: **Sensitivity study to the batch size $B$ on CIFAR-100-C under CSC and CDC settings.**

should be set to 1.0 for ViT to achieve the best results, suggesting that regularization is unnecessary for this architecture. Moreover, further increasing $K^{max}$ reduces the error rate by 0.6% on CCC, demonstrating the effectiveness of ReservoirTTA in this setting.

**Effect of Batch Size.** As illustrated in Figure 16, we explore batch sizes from 8 to 200 for both EATA + ReservoirTTA and ROID* + ReservoirTTA on CIFAR-100-C under recurring CSC and CDC settings. We observe a consistent decrease in error rate with larger batches, which likely stems from more stable gradient estimates, reduced variance in style features, and more accurate domain assignments. Conversely, very small batches can lead to noisy updates and suboptimal domain clustering. Although a large batch is important for reliable performance, memory constraints and real-time needs may limit its size in practice.

**Sensitivity to Various Sequence Orders.** We examine how DPCore, CoLA, and ReservoirTTA respond to different test batch orders during domain discovery. As shown in Figure 17, we measure the

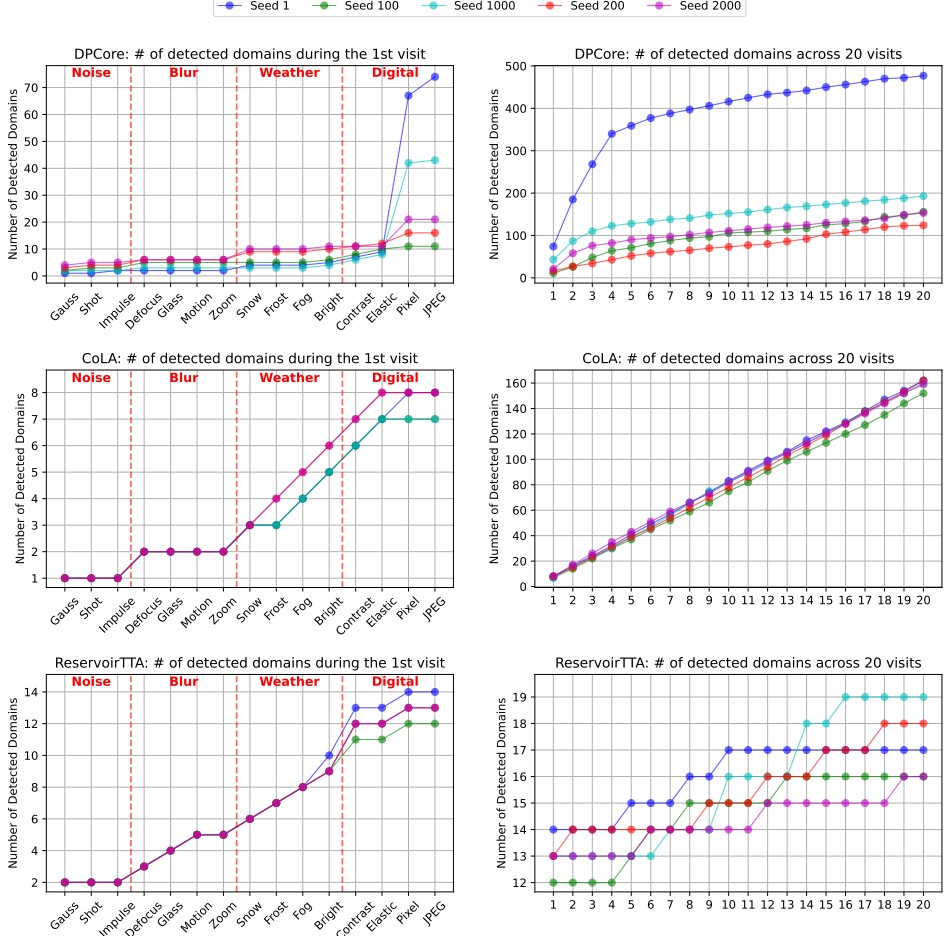

Figure 17: **Domain detection sensitivity comparison in recurring CSC on CIFAR-100-C**. The plot tracks the number of detected domains over time across different seeds. ReservoirTTA achieves the most accurate detection, while DPCore and CoLA are overly sensitive—detecting more than a hundred domains after 20 visits despite the dataset having only 15. This over-detection leads to extra domain-specific parameters (e.g., prompt, affine parameters), increasing memory overhead.

number of detected domains after the first visit and across 20 visits in recurring CSC on CIFAR100-C using five seeds—each corresponding to a distinct batch order while keeping the domain sequence fixed. Our findings reveal that DPCore is highly sensitive to batch order, undermining its reliability. Although CoLA yields consistent detection across seeds, its number of detected domains continuously increases, reaching 160 models after 20 visits. In contrast, ReservoirTTA shows minimal sensitivity, with detected domains stabilizing near the ground truth of 15 after 20 visits. These results underscore the superiority of our style quantifier, online clustering, and model discovery mechanisms over those used in CoLA and DPCore.

**Resetting to Source vs. Reusing Specialists.** We analyze reset-based strategies to isolate the benefit of reusing adapted specialists versus simply resetting to source parameters. On CIFAR-100-C (CSC), blind periodic resets every 1,000 iterations (RDumb) yield an average error of 32.17%. Domain-aware resets that trigger only when the VGG-based detector flags a domain change (RDumb w/ VGG) improve to 31.17% ($-1.0\%$). By contrast, ReservoirTTA+EATA reactivates the specialist previously adapted to a recurring domain—avoiding costly re-adaptation and forgetting—and achieves 28.57% ($-2.60\%$ vs. domain-aware, $-3.60\%$ vs. blind). See Table 11 for the comparison results.

Table 11: Comparison of reset-based baselines and ReservoirTTA on CIFAR100-C (CSC). Error (%) at visits 1–20 and mean.

| Method | Visits 1–10 | | | | | | | | | |
| | 1 | 2 | 3 | 4 | 5 | 6 | 7 | 8 | 9 | 10 |
|---|---|---|---|---|---|---|---|---|---|---|
| RDumb | 31.95 | 31.49 | 32.34 | 32.85 | 31.96 | 31.51 | 32.24 | 32.99 | 32.07 | 31.48 |
| RDumb w/ VGG | 31.11 | 31.13 | 31.17 | 31.20 | 31.19 | 31.16 | 31.20 | 31.21 | 31.20 | 31.21 |
| EATA+ReservoirTTA | 30.56 | 29.07 | 28.75 | 28.58 | 28.52 | 28.46 | 28.41 | 28.41 | 28.39 | 28.39 |

| Method | Visits 11–20 | | | | | | | | | | |
| | 11 | 12 | 13 | 14 | 15 | 16 | 17 | 18 | 19 | 20 | Avg |
|---|---|---|---|---|---|---|---|---|---|---|---|
| RDumb | 32.29 | 32.82 | 32.07 | 31.48 | 32.41 | 32.88 | 32.15 | 31.43 | 32.22 | 32.86 | 32.17 |
| RDumb w/ VGG | 31.12 | 31.17 | 31.16 | 31.17 | 31.11 | 31.17 | 31.17 | 31.16 | 31.20 | 31.19 | 31.17 |
| EATA+ReservoirTTA | 28.38 | 28.40 | 28.37 | 28.37 | 28.37 | 28.40 | 28.36 | 28.43 | 28.37 | 28.38 | **28.57** |

## J  Additional Qualitative Results

Figure 18 shows pseudo segmentation label visualization on the ACDC dataset [34]. The first two columns display outputs from TENT and BECoTTA, respectively, while the third column presents refined labels from BECoTTA combined with ReservoirTTA. The inclusion of ReservoirTTA yields more detailed and accurate segmentation labels, improving overall prediction quality.

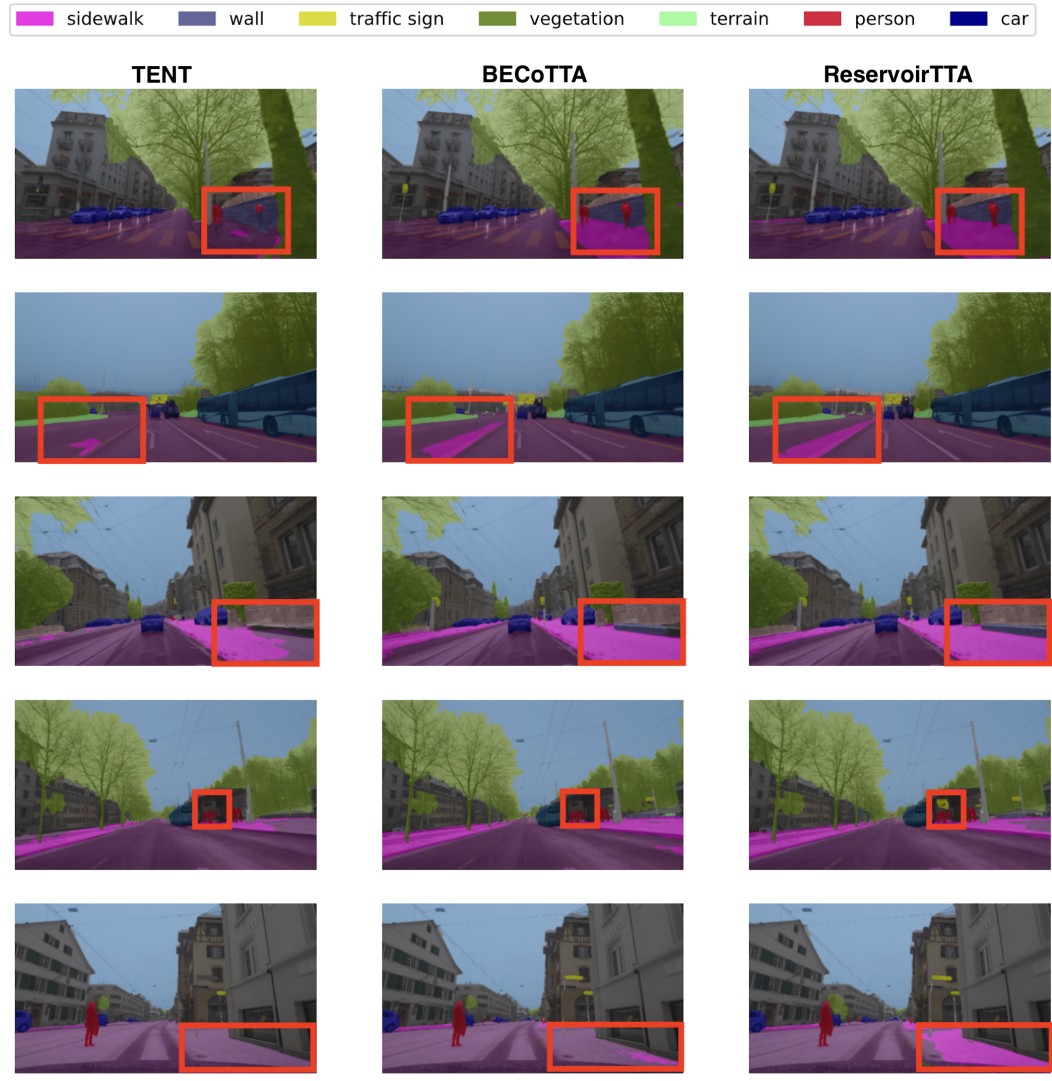

Figure 18: **Pseudo segmentation label visualization on ACDC [34]**. The first two columns show results from TENT and BECoTTA, respectively, while the third column presents results produced by BECoTTA+ReservoirTTA. Incorporating ReservoirTTA yields more fine-grained and accurate labels than the baselines. **Zoom in for best view**.

## K  Additional Quantitative Results

**Additional Evaluations on Recurring Domain Shifts.** In Tables 12–23, we present extensive quantitative evaluations on classification benchmarks under various recurring domain shift scenarios (CSC, CDC, and CCC) and across different architectures. Overall, the results confirm that ReservoirTTA significantly reduces error rates and prevents catastrophic forgetting compared to existing TTA methods. These evaluations reinforce our main findings, demonstrating that ReservoirTTA's architecture-agnostic design allows seamless adoption in both CNN- and transformer-based networks.

**Additional Evaluation on Object-Level Style Shifts.** To further assess the versatility of our framework beyond scene-level corruptions, we evaluate on datasets with pronounced object-level domain gaps, namely DomainNet-126 (126 classes) and PACS (7 classes). On DomainNet-126, we follow a CSC protocol by training a ResNet-50 on each source domain (real, painting, clipart, sketch), then adapt over a 20-cycle recurring sequence of the remaining three target domains. A similar protocol is applied to PACS, using photo, art painting, cartoon, and sketch as sources. As reported in Tables 24 and 25, integrating ReservoirTTA with EATA consistently reduces error across both datasets, yielding average improvements of 0.79% on DomainNet-126 and 0.70% on PACS, with the largest gains observed when adapting from Real on DomainNet-126 (–1.60%) and from Cartoon on PACS (–1.93%). These results show that a framework motivated by scene-level shifts transfers robustly to object-level distribution changes.

**Evaluation Under Gradual Shift.** To assess the ability of our framework to handle gradual changes in corruption severity, we evaluate on a gradual domain shift stream constructed from CIFAR-100-C (CSC) using a ResNeXt-29 backbone. Each corruption type cycles through severity levels $1 \rightarrow 2 \rightarrow 3 \rightarrow 4 \rightarrow 5 \rightarrow 4 \rightarrow 3 \rightarrow 2 \rightarrow 1$ over 20 recurrences. Even under these subtle transitions, ReservoirTTA consistently improves over EATA. As shown in Table 26, EATA averages $25.41\%$ error, while ReservoirTTA achieves $25.06\%$ with $K_{\max} = 16$ ($-0.35\%$) and $24.95\%$ with $K_{\max} = 64$ (additional $-0.10\%$). Thus, our method adapts to gradually evolving severities, not only abrupt domain changes.

**Can ReservoirTTA Help Tackle Temporally Correlated Test Streams?** ReservoirTTA raises the question of whether it can tackle the challenge of a temporally correlated testing stream. As observed in Table 27—on CIFAR-10 $\rightarrow$ CIFAR-10-C under recurring continual CSC, with **temporal correlation** of image categories (Dirichlet coefficient = 0.1)— where we integrate ReservoirTTA with state-of-the-art TTA method, RoTTA [45], which is designed to address temporal correlations—yields similar gains. RoTTA tackles the instability caused by temporally correlated test samples by introducing mechanisms that stabilize the online adaptation process via smoothing the updates over time. In particular, RoTTA uses an exponential moving average (EMA)–based update to accumulate robust, long-term statistics that mitigate the bias introduced by correlated mini-batches. This principle is further enforced by using robust batch normalization (RBN) and by maintaining a memory bank through category-balanced sampling, which ensures a representative snapshot of the test distribution.

*Although tackling temporally correlated test streams is not the primary focus of ReservoirTTA, it is important to note that during test time, when images are highly correlated, extracting style information using channel-wise feature statistics (logvar) can become biased toward the majority category. To address this issue, we adopt RoTTA's Category-Balanced Sampling with Timeliness and Uncertainty (CSTU) strategy, which resamples images from the online stream in a way that maintains category balance. By integrating this strategy with our method, we enhance stability over an extended, temporally correlated test stream, and instead of using the ensemble output from our reservoir models, we make predictions using the mean-teacher approach as introduced in RoTTA. While methods such as RoTTA are potential candidates for integrating ReservoirTTA, their need for separate memory banks per model makes integration computationally prohibitive and beyond the scope of our paper, which focuses on prolonged domain recurrence rather than temporally correlated test streams.*

Table 12: **Average classification error rates (%) for *WideResNet-28* on CIFAR-10 → CIFAR-10-C at severity level 5 under recurring continual CSC.** The lowest error is highlighted in **bold**. Results are averaged over 5 seeds.

| Method | *Recurring TTA visit* 1 | 2 | 3 | 4 | 5 | 6 | 7 | 8 | 9 | 10 | 11 | 12 | 13 | 14 | 15 | 16 | 17 | 18 | 19 | 20 | Avg |
|---|---|---|---|---|---|---|---|---|---|---|---|---|---|---|---|---|---|---|---|---|---|
| Source | 43.52 | | | | | | | | | | | | | | | | | | | | 43.52 |
| **Single-Target TTA** | | | | | | | | | | | | | | | | | | | | | |
| Tent | 19.28 | 23.63 | 30.63 | 39.91 | 49.74 | 57.08 | 62.22 | 66.46 | 72.86 | 74.92 | 77.66 | 81.65 | 84.09 | 85.32 | 85.39 | 86.13 | 86.71 | 87.70 | 88.14 | 87.84 | 67.37 |
| + *ReservoirTTA* | 18.28 | 17.67 | 17.57 | 17.51 | 17.48 | 17.48 | 17.49 | 17.47 | 17.45 | 17.47 | 17.48 | 17.50 | 17.49 | 17.49 | 17.47 | 17.50 | 17.50 | 17.53 | 17.55 | 17.56 | 17.55 |
| **Continual TTA** | | | | | | | | | | | | | | | | | | | | | |
| CoTTA* | 18.75 | 18.51 | 18.43 | 18.46 | 18.48 | 18.57 | 18.79 | 18.93 | 19.18 | 19.41 | 19.66 | 19.92 | 20.22 | 20.44 | 20.80 | 21.06 | 21.38 | 21.75 | 22.07 | 22.40 | 19.86 |
| RoTTA | 19.35 | 16.74 | 16.77 | 17.01 | 17.17 | 17.27 | 17.45 | 17.55 | 17.64 | 17.77 | 17.86 | 17.90 | 18.00 | 17.99 | 18.09 | 18.12 | 18.19 | 18.23 | 18.27 | 18.35 | 17.79 |
| ETA | 17.78 | 18.89 | 19.87 | 20.74 | 21.52 | 22.47 | 23.88 | 25.24 | 25.83 | 26.49 | 26.56 | 27.22 | 27.46 | 27.12 | 27.50 | 27.27 | 28.24 | 28.59 | 29.51 | 30.89 | 25.15 |
| + *ReservoirTTA* | 17.49 | **16.71** | **16.56** | **16.47** | **16.43** | **16.39** | **16.38** | **16.36** | **16.39** | **16.37** | **16.37** | **16.38** | **16.38** | **16.39** | **16.38** | **16.39** | **16.38** | **16.39** | **16.42** | **16.41** | **16.47** |
| SAR | 20.39 | 20.35 | 20.38 | 20.40 | 20.38 | 20.38 | 20.38 | 20.37 | 20.37 | 20.36 | 20.39 | 20.39 | 20.40 | 20.37 | 20.39 | 20.41 | 20.39 | 20.36 | 20.39 | 20.39 | 20.38 |
| + *ReservoirTTA* | 20.35 | 20.32 | 20.36 | 20.38 | 20.33 | 20.36 | 20.36 | 20.34 | 20.32 | 20.35 | 20.36 | 20.36 | 20.37 | 20.33 | 20.36 | 20.38 | 20.36 | 20.33 | 20.36 | 20.36 | 20.35 |
| **Persistent TTA** | | | | | | | | | | | | | | | | | | | | | |
| RDumb | 17.78 | 17.39 | 18.22 | 18.23 | 17.85 | 17.53 | 18.42 | 18.32 | 17.78 | 17.38 | 18.08 | 18.37 | 17.83 | 17.44 | 18.11 | 18.28 | 17.95 | 17.60 | 18.43 | 18.41 | 17.97 |
| PeTTA | 22.98 | 20.33 | 18.93 | 18.24 | 17.80 | 17.49 | 17.35 | 17.27 | 17.23 | 17.18 | 17.16 | 17.16 | 17.14 | 17.09 | 17.11 | 17.14 | 17.17 | 17.16 | 17.17 | 17.20 | 17.82 |
| EATA | **17.46** | 17.53 | 17.67 | 17.63 | 17.61 | 17.73 | 17.75 | 17.62 | 17.55 | 17.66 | 17.70 | 17.60 | 17.69 | 17.78 | 17.71 | 17.81 | 17.79 | 17.76 | 17.84 | 17.76 | 17.68 |
| + *ReservoirTTA* | 17.53 | 16.78 | 16.63 | 16.58 | 16.55 | 16.52 | 16.48 | 16.46 | 16.46 | 16.45 | 16.44 | 16.46 | 16.44 | 16.41 | 16.45 | 16.42 | 16.43 | 16.44 | 16.44 | 16.44 | 16.54 |
| ROID* | 17.75 | 17.62 | 17.54 | 17.56 | 17.60 | 17.59 | 17.59 | 17.68 | 17.58 | 17.49 | 17.59 | 17.52 | 17.56 | 17.50 | 17.60 | 17.60 | 17.58 | 17.55 | 17.52 | 17.65 | 17.58 |
| + *ReservoirTTA* | 17.75 | 17.08 | 16.91 | 16.92 | 16.89 | 16.85 | 16.83 | 16.81 | 16.80 | 16.79 | 16.81 | 16.77 | 16.81 | 16.78 | 16.79 | 16.78 | 16.80 | 16.78 | 16.79 | | 16.88 |

Table 13: **Average classification error rates (%) for *WideResNet-28* on CIFAR-10 → CIFAR-10-C at severity level 5 under recurring continual CDC.** The lowest error is highlighted in **bold**. Results are averaged over 5 seeds.

| Method | *Recurring TTA visit* 1 | 2 | 3 | 4 | 5 | 6 | 7 | 8 | 9 | 10 | 11 | 12 | 13 | 14 | 15 | 16 | 17 | 18 | 19 | 20 | Avg |
|---|---|---|---|---|---|---|---|---|---|---|---|---|---|---|---|---|---|---|---|---|---|
| Source | 43.52 | | | | | | | | | | | | | | | | | | | | 43.52 |
| **Single-Target TTA** | | | | | | | | | | | | | | | | | | | | | |
| Tent | 20.49 | 26.20 | 37.00 | 48.75 | 55.66 | 61.89 | 68.99 | 72.51 | 74.38 | 77.05 | 79.47 | 81.39 | 82.01 | 82.51 | 83.81 | 86.02 | 87.11 | 87.14 | 86.79 | 86.95 | 69.30 |
| + *ReservoirTTA* | 18.15 | 17.45 | 17.32 | 17.32 | 17.31 | 17.29 | 17.28 | 17.26 | 17.30 | 17.28 | 17.30 | 17.30 | 17.30 | 17.32 | 17.34 | 17.33 | 17.34 | 17.33 | 17.37 | 17.37 | 17.36 |
| **Continual TTA** | | | | | | | | | | | | | | | | | | | | | |
| CoTTA* | 18.75 | 18.46 | 18.37 | 18.35 | 18.41 | 18.55 | 18.71 | 18.92 | 19.14 | 19.35 | 19.60 | 19.85 | 20.16 | 20.41 | 20.73 | 21.01 | 21.30 | 21.67 | 21.99 | 22.29 | 19.80 |
| RoTTA | 21.86 | 19.07 | 18.52 | 19.01 | 19.25 | 19.52 | 19.04 | 19.42 | 19.68 | 20.06 | 19.91 | 20.05 | 20.29 | 20.55 | 20.25 | 20.45 | 20.15 | 20.51 | 20.50 | 20.38 | 19.92 |
| ETA | 17.90 | 18.93 | 19.86 | 20.98 | 21.57 | 22.36 | 23.33 | 24.51 | 25.70 | 26.33 | 26.33 | 27.23 | 27.51 | 28.29 | 27.92 | 28.98 | 30.20 | 31.64 | 32.88 | 33.49 | 25.80 |
| + *ReservoirTTA* | **17.40** | **16.44** | **16.31** | **16.26** | **16.22** | **16.21** | **16.20** | **16.18** | **16.21** | **16.21** | **16.21** | **16.22** | **16.21** | **16.21** | **16.22** | **16.23** | **16.24** | **16.23** | **16.24** | **16.27** | **16.30** |
| SAR | 20.41 | 20.41 | 20.41 | 20.38 | 20.38 | 20.41 | 20.41 | 20.40 | 20.41 | 20.40 | 20.41 | 20.40 | 20.40 | 20.43 | 20.41 | 20.37 | 20.42 | 20.42 | 20.40 | 20.39 | 20.41 |
| + *ReservoirTTA* | 20.38 | 20.34 | 20.38 | 20.35 | 20.38 | 20.36 | 20.38 | 20.36 | 20.36 | 20.35 | 20.34 | 20.37 | 20.36 | 20.37 | 20.34 | 20.32 | 20.35 | 20.36 | 20.33 | 20.35 | 20.35 |
| **Persistent TTA** | | | | | | | | | | | | | | | | | | | | | |
| RDumb | 17.90 | 18.05 | 18.00 | 18.08 | 17.86 | 17.99 | 18.15 | 17.86 | 17.87 | 18.00 | 17.87 | 18.27 | 18.34 | 17.94 | 17.90 | 18.23 | 18.27 | 17.97 | 17.94 | 18.12 | 18.03 |
| PeTTA | 27.16 | 24.30 | 22.53 | 22.57 | 21.41 | 21.92 | 20.76 | 20.66 | 21.00 | 21.67 | 20.93 | 21.27 | 20.76 | 21.55 | 20.62 | 20.65 | 20.80 | 21.56 | 21.15 | 20.81 | 21.70 |
| EATA | 17.70 | 17.87 | 17.85 | 17.74 | 17.87 | 17.96 | 17.82 | 17.82 | 17.96 | 17.95 | 18.05 | 17.87 | 17.92 | 17.82 | 17.74 | 17.90 | 17.87 | 17.87 | 17.85 | 17.88 | 17.87 |
| + *ReservoirTTA* | 17.46 | 16.62 | 16.48 | 16.45 | 16.42 | 16.42 | 16.39 | 16.39 | 16.38 | 16.37 | 16.37 | 16.37 | 16.37 | 16.36 | 16.37 | 16.40 | 16.37 | 16.38 | 16.39 | 16.39 | 16.46 |
| ROID* | 18.04 | 17.91 | 17.95 | 17.93 | 17.91 | 18.04 | 17.96 | 17.96 | 18.00 | 18.00 | 17.97 | 18.01 | 18.04 | 18.02 | 17.93 | 17.99 | 17.98 | 17.91 | 17.89 | 18.07 | 17.98 |
| + *ReservoirTTA* | 17.86 | 17.12 | 16.96 | 16.86 | 16.84 | 16.84 | 16.82 | 16.78 | 16.81 | 16.82 | 16.80 | 16.81 | 16.77 | 16.76 | 16.77 | 16.75 | 16.77 | 16.75 | 16.73 | 16.78 | 16.87 |

Table 14: **Average classification error rates (%) for *ResNeXt-29* on CIFAR-100 → CIFAR-100-C at severity level 5 under recurring continual CSC.** The lowest error is highlighted in **bold**. Results are averaged over 5 seeds.

| Method | *Recurring TTA visit* 1 | 2 | 3 | 4 | 5 | 6 | 7 | 8 | 9 | 10 | 11 | 12 | 13 | 14 | 15 | 16 | 17 | 18 | 19 | 20 | Avg |
|---|---|---|---|---|---|---|---|---|---|---|---|---|---|---|---|---|---|---|---|---|---|
| Source | 46.45 | | | | | | | | | | | | | | | | | | | | 46.45 |
| **Single-Target TTA** | | | | | | | | | | | | | | | | | | | | | |
| Tent | 61.41 | 97.28 | 98.18 | 98.62 | 98.74 | 98.81 | 98.84 | 98.86 | 98.87 | 98.81 | 98.88 | 98.93 | 98.92 | 98.95 | 99.00 | 98.99 | 99.00 | 99.00 | 99.00 | 99.00 | 96.90 |
| + *ReservoirTTA* | 38.06 | 39.42 | 40.68 | 41.46 | 42.04 | 42.42 | 42.67 | 42.95 | 43.12 | 43.25 | 43.40 | 43.48 | 43.62 | 43.64 | 43.74 | 43.78 | 43.83 | 43.86 | 43.92 | 43.97 | 42.67 |
| **Continual TTA** | | | | | | | | | | | | | | | | | | | | | |
| CoTTA* | 35.10 | 36.08 | 37.29 | 38.73 | 40.22 | 41.83 | 43.50 | 45.21 | 46.86 | 48.59 | 50.41 | 52.10 | 53.87 | 55.55 | 57.33 | 59.04 | 60.72 | 62.37 | 63.93 | 65.46 | 49.71 |
| RoTTA | 34.80 | 33.12 | 34.77 | 36.27 | 37.81 | 39.43 | 40.90 | 42.19 | 43.68 | 45.04 | 46.50 | 48.17 | 49.66 | 51.15 | 52.65 | 53.96 | 55.31 | 56.61 | 57.90 | 59.10 | 45.95 |
| ETA | 31.95 | 33.05 | 33.77 | 34.30 | 34.66 | 34.91 | 35.38 | 35.62 | 35.91 | 35.92 | 36.17 | 36.50 | 36.63 | 36.79 | 36.89 | 37.17 | 37.20 | 37.44 | 37.43 | 37.64 | 35.77 |
| + *ReservoirTTA* | 31.59 | 30.12 | 29.77 | 29.69 | 29.66 | 29.65 | 29.67 | 29.69 | 29.78 | 29.79 | 29.82 | 29.83 | 29.86 | 29.91 | 29.93 | 29.96 | 29.98 | 29.99 | 30.04 | 30.02 | 29.94 |
| SAR | 31.92 | 32.77 | 35.01 | 37.41 | 39.92 | 43.03 | 47.26 | 52.68 | 59.40 | 61.16 | 43.38 | 32.79 | 34.81 | 36.93 | 39.34 | 42.14 | 45.70 | 50.42 | 56.81 | 60.36 | 44.16 |
| + *ReservoirTTA* | 31.91 | 30.90 | 30.50 | 30.30 | 30.14 | 30.13 | 30.11 | 30.09 | 30.09 | 30.12 | 30.17 | 30.16 | 30.20 | 30.24 | 30.27 | 30.33 | 30.39 | 30.37 | 30.47 | 30.48 | 30.37 |
| **Persistent TTA** | | | | | | | | | | | | | | | | | | | | | |
| RDumb | 31.95 | 31.49 | 32.34 | 32.85 | 31.96 | 31.51 | 32.24 | 32.99 | 32.07 | 31.48 | 32.29 | 32.82 | 32.07 | 31.48 | 32.41 | 32.88 | 32.15 | 31.43 | 32.22 | 32.86 | 32.17 |
| PeTTA | 39.44 | 34.17 | 33.08 | 32.76 | 32.72 | 32.74 | 32.72 | 32.73 | 32.74 | 32.73 | 32.81 | 32.80 | 32.80 | 32.77 | 32.80 | 32.87 | 32.91 | 32.88 | 32.86 | 32.91 | 33.21 |
| EATA | 30.51 | 30.29 | 30.39 | 30.39 | 30.45 | 30.47 | 30.38 | 30.44 | 30.47 | 30.53 | 30.51 | 30.46 | 30.47 | 30.51 | 30.51 | 30.48 | 30.51 | 30.54 | 30.47 | 30.47 | 30.46 |
| +*ReservoirTTA* | 30.56 | 29.07 | 28.75 | 28.58 | 28.52 | 28.46 | 28.41 | 28.41 | 28.39 | 28.39 | 28.38 | 28.40 | 28.37 | 28.37 | 28.37 | 28.40 | 28.36 | 28.43 | 28.37 | 28.38 | 28.57 |
| ROID* | **29.45** | 29.25 | 29.26 | 29.27 | 29.26 | 29.23 | 29.18 | 29.23 | 29.18 | 29.20 | 29.30 | 29.19 | 29.20 | 29.27 | 29.25 | 29.29 | 29.27 | 29.22 | 29.23 | 29.25 | 29.25 |
| + *ReservoirTTA* | 29.60 | **28.20** | **27.96** | **27.90** | **27.83** | **27.83** | **27.79** | **27.81** | **27.77** | **27.80** | **27.80** | **27.78** | **27.79** | **27.79** | **27.76** | **27.79** | **27.77** | **27.78** | **27.81** | **27.80** | **27.92** |

Table 15: **Average classification error rates (%) for *ResNeXt-29* on CIFAR-100 → CIFAR-100-C at severity level 5 under recurring continual CDC.** The lowest error is highlighted in **bold**. Results are averaged over 5 seeds.

| Method | Recurring TTA visit 1 | 2 | 3 | 4 | 5 | 6 | 7 | 8 | 9 | 10 | 11 | 12 | 13 | 14 | 15 | 16 | 17 | 18 | 19 | 20 | Avg |
|---|---|---|---|---|---|---|---|---|---|---|---|---|---|---|---|---|---|---|---|---|---|
| Source | | | | | | | | | | 46.45 | | | | | | | | | | | 46.45 |
| **Single-Target TTA** | | | | | | | | | | | | | | | | | | | | | |
| Tent | 60.21 | 96.26 | 97.79 | 98.29 | 98.45 | 98.47 | 98.51 | 98.52 | 98.59 | 98.75 | 98.79 | 98.83 | 98.80 | 98.82 | 98.87 | 98.87 | 98.84 | 98.85 | 98.87 | 98.87 | 96.61 |
| + *ReservoirTTA* | 33.90 | 34.86 | 35.92 | 36.66 | 37.17 | 37.57 | 37.86 | 38.12 | 38.24 | 38.41 | 38.54 | 38.62 | 38.78 | 38.99 | 39.12 | 39.25 | 39.38 | 39.55 | 39.63 | 39.73 | 38.02 |
| **Continual TTA** | | | | | | | | | | | | | | | | | | | | | |
| CoTTA* | 35.09 | 36.07 | 37.25 | 38.67 | 40.19 | 41.74 | 43.40 | 44.98 | 46.71 | 48.38 | 50.10 | 51.75 | 53.55 | 55.16 | 56.81 | 58.61 | 60.22 | 61.85 | 63.51 | 65.13 | 49.46 |
| RoTTA | 36.82 | 34.81 | 36.50 | 39.18 | 41.02 | 43.22 | 44.10 | 46.65 | 49.10 | 53.29 | 56.45 | 57.31 | 60.21 | 65.22 | 65.83 | 69.21 | 70.08 | 71.15 | 73.27 | 73.82 | 54.36 |
| ETA | 32.36 | 33.19 | 33.75 | 34.26 | 34.66 | 35.06 | 35.38 | 35.66 | 35.87 | 36.22 | 36.27 | 36.49 | 36.60 | 36.75 | 36.97 | 37.24 | 37.17 | 37.29 | 37.43 | 37.62 | 35.81 |
| + *ReservoirTTA* | 30.87 | 29.78 | 29.48 | 29.35 | 29.31 | 29.30 | 29.30 | 29.39 | 29.35 | 29.36 | 29.40 | 29.40 | 29.45 | 29.46 | 29.47 | 29.50 | 29.55 | 29.53 | 29.57 | 29.57 | 29.52 |
| SAR | 31.64 | 32.42 | 34.60 | 36.75 | 39.00 | 41.78 | 45.16 | 49.83 | 55.04 | 61.63 | 59.75 | 36.30 | 32.83 | 34.82 | 37.10 | 39.76 | 42.68 | 46.54 | 51.75 | 57.81 | 43.36 |
| + *ReservoirTTA* | 31.72 | 30.67 | 30.18 | 29.91 | 29.72 | 29.69 | 29.60 | 29.56 | 29.56 | 29.59 | 29.59 | 29.60 | 29.62 | 29.64 | 29.65 | 29.71 | 29.73 | 29.75 | 29.77 | 29.79 | 29.85 |
| **Persistent TTA** | | | | | | | | | | | | | | | | | | | | | |
| RDumb | 32.36 | 32.32 | 32.41 | 32.60 | 32.34 | 32.27 | 32.42 | 32.65 | 32.19 | 32.37 | 32.50 | 32.74 | 32.23 | 32.14 | 32.46 | 32.87 | 32.27 | 32.24 | 32.23 | 32.63 | 32.41 |
| PeTTA | 42.13 | 36.61 | 35.15 | 35.09 | 34.84 | 35.16 | 34.52 | 34.75 | 34.94 | 35.27 | 34.89 | 35.19 | 34.93 | 35.22 | 35.00 | 35.12 | 34.99 | 35.45 | 35.21 | 35.34 | 35.49 |
| EATA | 31.01 | 30.88 | 30.85 | 30.88 | 30.86 | 31.01 | 30.88 | 30.98 | 31.01 | 31.05 | 31.09 | 30.93 | 30.96 | 31.07 | 30.99 | 31.02 | 31.03 | 30.97 | 31.07 | 31.08 | 30.98 |
| +*ReservoirTTA* | 30.44 | 29.07 | 28.74 | 28.61 | 28.50 | 28.52 | 28.40 | 28.40 | 28.35 | 28.39 | 28.38 | 28.34 | 28.38 | 28.37 | 28.40 | 28.40 | 28.41 | 28.40 | 28.38 | 28.40 | 28.56 |
| ROID* | 30.17 | 30.01 | 29.94 | 29.97 | 29.86 | 30.11 | 29.87 | 30.02 | 29.96 | 30.08 | 30.04 | 29.99 | 29.92 | 30.03 | 29.97 | 29.97 | 29.92 | 30.02 | 30.00 | 30.12 | 30.00 |
| + *ReservoirTTA* | **29.62** | **28.27** | **28.01** | **27.95** | **27.88** | **27.89** | **27.84** | **27.87** | **27.83** | **27.84** | **27.83** | **27.78** | **27.79** | **27.79** | **27.82** | **27.82** | **27.83** | **27.82** | **27.81** | **27.78** | **27.95** |

Table 16: **Average classification error rates (%) for *ResNet-50-BN* on ImageNet → ImageNet-C at severity level 5 under recurring continual CSC.** The lowest error is highlighted in **bold**. Results are averaged over 5 seeds.

| Method | Recurring TTA visit 1 | 2 | 3 | 4 | 5 | 6 | 7 | 8 | 9 | 10 | 11 | 12 | 13 | 14 | 15 | 16 | 17 | 18 | 19 | 20 | Avg |
|---|---|---|---|---|---|---|---|---|---|---|---|---|---|---|---|---|---|---|---|---|---|
| Source | | | | | | | | | | 82.03 | | | | | | | | | | | 82.03 |
| **Single-Target TTA** | | | | | | | | | | | | | | | | | | | | | |
| Tent | 62.60 | 62.16 | 64.88 | 69.70 | 76.49 | 84.36 | 92.50 | 97.33 | 98.80 | 99.14 | 99.24 | 99.31 | 99.33 | 99.35 | 99.38 | 99.41 | 99.44 | 99.44 | 99.46 | 99.48 | 90.09 |
| + *ReservoirTTA* | 62.60 | 59.78 | 58.61 | 57.83 | 57.34 | 57.09 | 56.89 | 56.83 | 56.82 | 56.80 | 56.88 | 56.93 | 57.07 | 57.16 | 57.30 | 57.43 | 57.66 | 57.79 | 58.07 | 58.22 | 57.76 |
| **Continual TTA** | | | | | | | | | | | | | | | | | | | | | |
| CoTTA* | 67.58 | 65.65 | 64.32 | 63.49 | 62.99 | 62.59 | 62.37 | 62.16 | 62.08 | 61.93 | 61.89 | 61.82 | 61.76 | 61.78 | 61.70 | 61.74 | 61.74 | 61.74 | 61.70 | 61.68 | 62.64 |
| RoTTA | 67.27 | 61.59 | 61.60 | 63.76 | 66.26 | 69.53 | 75.71 | 79.94 | 90.41 | 95.85 | 96.59 | 97.15 | 97.59 | 98.08 | 98.36 | 98.67 | 98.86 | 99.07 | 99.23 | 99.37 | 85.74 |
| ETA | 60.00 | 58.11 | 58.01 | 58.22 | 58.22 | 58.25 | 58.41 | 58.54 | 58.65 | 58.80 | 58.84 | 58.95 | 59.04 | 59.11 | 59.25 | 59.28 | 59.31 | 59.35 | 59.47 | 59.40 | 58.86 |
| + *ReservoirTTA* | 59.77 | 56.09 | 54.95 | 54.38 | 53.95 | 53.76 | 53.52 | 53.43 | 53.36 | 53.30 | 53.22 | 53.19 | 53.14 | 53.15 | 53.14 | 53.11 | 53.15 | 53.11 | 53.10 | 53.10 | 53.90 |
| SAR | 61.87 | 59.56 | 59.28 | 59.30 | 59.26 | 59.41 | 59.60 | 59.82 | 60.15 | 60.45 | 60.89 | 61.29 | 61.89 | 62.47 | 62.98 | 63.65 | 64.52 | 65.17 | 66.04 | 67.06 | 61.73 |
| + *ReservoirTTA* | 62.21 | 59.16 | 57.81 | 56.91 | 56.16 | 55.69 | 55.23 | 54.88 | 54.65 | 54.37 | 54.18 | 54.01 | 53.83 | 53.73 | 53.58 | 53.50 | 53.41 | 53.30 | 53.22 | 53.13 | 55.15 |
| **Persistent TTA** | | | | | | | | | | | | | | | | | | | | | |
| RDumb | 59.82 | 58.24 | 57.23 | 57.02 | 58.22 | 59.78 | 59.05 | 57.26 | 57.11 | 57.62 | 59.62 | 59.17 | 57.94 | 57.17 | 56.78 | 56.36 | 59.71 | 58.49 | 57.34 | 56.78 | 58.14 |
| PeTTA | 67.50 | 68.60 | 66.69 | 64.94 | 63.64 | 62.57 | 61.94 | 61.96 | 61.24 | 61.09 | 60.90 | 60.51 | 60.33 | 60.41 | 60.33 | 60.33 | 60.24 | 60.03 | 60.31 | 60.11 | 62.18 |
| EATA | 57.52 | 55.91 | 55.64 | 55.54 | 55.45 | 55.45 | 55.60 | 55.56 | 55.59 | 55.57 | 55.59 | 55.61 | 55.71 | 55.77 | 55.70 | 55.77 | 55.78 | 55.83 | 55.94 | 55.94 | 55.77 |
| + *ReservoirTTA* | 58.03 | 54.32 | 53.22 | **52.60** | **52.19** | **51.96** | **51.73** | **51.59** | **51.46** | **51.40** | **51.31** | **51.25** | **51.19** | **51.13** | **51.15** | **51.09** | **51.07** | **51.03** | **51.00** | 51.00 | 51.99 |
| ROID* | **56.07** | 55.45 | 55.40 | 55.47 | 55.38 | 55.44 | 55.42 | 55.38 | 55.42 | 55.43 | 55.33 | 55.44 | 55.44 | 55.44 | 55.36 | 55.35 | 55.43 | 55.40 | 55.45 | 55.48 | 55.45 |
| + *ReservoirTTA* | 56.42 | **53.42** | **52.94** | 52.71 | 52.53 | 52.48 | 52.45 | 52.36 | 52.23 | 52.26 | 52.20 | 52.12 | 52.18 | 52.08 | 52.10 | 52.04 | 52.06 | 52.13 | 52.10 | 52.07 | 52.54 |

Table 17: **Average classification error rates (%) for *ResNet-50-BN* on ImageNet → ImageNet-C at severity level 5 under recurring continual CDC.** The lowest error is highlighted in **bold**. Results are averaged over 5 seeds.

| Method | Recurring TTA visit 1 | 2 | 3 | 4 | 5 | 6 | 7 | 8 | 9 | 10 | 11 | 12 | 13 | 14 | 15 | 16 | 17 | 18 | 19 | 20 | Avg |
|---|---|---|---|---|---|---|---|---|---|---|---|---|---|---|---|---|---|---|---|---|---|
| Source | | | | | | | | | | 82.03 | | | | | | | | | | | 82.03 |
| **Single-Target TTA** | | | | | | | | | | | | | | | | | | | | | |
| Tent | 61.98 | 61.72 | 64.26 | 68.94 | 75.61 | 82.84 | 90.14 | 95.76 | 98.26 | 99.07 | 99.32 | 99.40 | 99.45 | 99.46 | 99.47 | 99.50 | 99.51 | 99.52 | 99.54 | 99.54 | 89.66 |
| + *ReservoirTTA* | 62.43 | 59.20 | 57.84 | 57.16 | 56.78 | 56.53 | 56.38 | 56.25 | 56.25 | 56.35 | 56.41 | 56.47 | 56.61 | 56.70 | 56.81 | 56.95 | 57.04 | 57.15 | 57.19 | 57.52 | 57.20 |
| **Continual TTA** | | | | | | | | | | | | | | | | | | | | | |
| CoTTA* | 67.73 | 65.68 | 64.26 | 63.43 | 62.92 | 62.49 | 62.21 | 62.06 | 61.86 | 61.84 | 61.71 | 61.70 | 61.57 | 61.56 | 61.50 | 61.56 | 61.50 | 61.50 | 61.47 | 61.50 | 62.50 |
| RoTTA | 71.61 | 66.61 | 67.14 | 68.85 | 73.20 | 76.11 | 85.69 | 92.94 | 93.53 | 95.32 | 97.13 | 97.69 | 98.08 | 98.54 | 98.88 | 99.07 | 99.24 | 99.37 | 99.46 | 99.51 | 88.90 |
| ETA | 59.33 | 57.94 | 58.24 | 58.43 | 58.70 | 58.69 | 58.81 | 58.96 | 59.08 | 59.26 | 59.36 | 59.35 | 59.41 | 59.60 | 59.74 | 59.76 | 59.83 | 59.97 | 59.84 | 60.07 | 59.22 |
| + *ReservoirTTA* | 58.63 | 55.01 | 53.87 | 53.25 | 53.06 | 52.68 | 52.57 | 52.46 | 52.36 | 52.38 | 52.31 | 52.36 | 52.26 | 52.17 | 52.13 | 52.13 | 52.18 | 52.17 | 52.04 | 52.18 | 52.91 |
| SAR | 61.48 | 59.47 | 59.21 | 59.33 | 59.49 | 59.52 | 59.75 | 59.85 | 60.14 | 60.88 | 61.08 | 61.59 | 62.07 | 63.19 | 63.83 | 64.49 | 65.11 | 66.17 | — | — | 61.48 |
| + *ReservoirTTA* | 62.63 | 59.52 | 58.00 | 57.04 | 56.46 | 55.88 | 55.58 | 55.24 | 54.92 | 54.75 | 54.56 | 54.44 | 54.28 | 54.10 | 54.00 | 53.89 | 53.81 | 53.75 | 53.53 | 53.56 | 55.50 |
| **Persistent TTA** | | | | | | | | | | | | | | | | | | | | | |
| RDumb | 59.57 | 59.34 | 59.56 | 59.39 | 59.43 | 59.47 | 59.61 | 59.40 | 59.39 | 59.18 | 59.12 | 59.25 | 59.57 | 59.00 | 59.14 | 58.79 | 59.01 | 59.56 | 59.29 | 59.47 | 59.33 |
| PeTTA | 71.60 | 66.52 | 66.47 | 66.04 | 66.98 | 66.75 | 67.56 | 68.14 | 67.84 | 68.56 | 67.70 | 68.12 | 67.94 | 68.69 | 69.05 | 68.67 | 68.56 | 69.98 | 68.75 | 69.52 | 68.17 |
| EATA | 58.46 | 56.97 | 56.85 | 56.73 | 56.84 | 56.78 | 56.86 | 56.80 | 56.84 | 56.79 | 56.76 | 56.89 | 56.84 | 56.82 | 56.88 | 56.99 | 57.01 | 56.98 | 56.85 | 57.02 | 56.95 |
| + *ReservoirTTA* | 58.53 | 54.86 | 53.73 | **53.08** | **52.85** | **52.55** | **52.42** | **52.31** | **52.13** | **52.05** | **51.97** | **51.99** | **51.93** | **51.82** | **51.76** | **51.74** | **51.73** | **51.73** | **51.59** | **51.81** | 52.63 |
| ROID* | 58.70 | 58.23 | 58.26 | 58.38 | 58.37 | 58.26 | 58.48 | 58.28 | 58.39 | 58.32 | 58.27 | 58.22 | 58.26 | 58.27 | 58.37 | 58.41 | 58.46 | 58.37 | 58.04 | 58.25 | 58.33 |
| + *ReservoirTTA* | **56.95** | **53.99** | **53.62** | 53.37 | 53.37 | 53.10 | 53.16 | 53.03 | 52.93 | 52.90 | 52.98 | 53.05 | 52.89 | 52.88 | 52.83 | 52.91 | 52.90 | 52.89 | 52.87 | 53.01 | 53.28 |

Table 18: **Average classification error rates (%) for *ResNet-50-GN* on ImageNet → ImageNet-C at severity level 5 under recurring continual CSC.** The lowest error is highlighted in **bold**. Results are averaged over 5 seeds.

| Method | 1 | 2 | 3 | 4 | 5 | 6 | 7 | 8 | 9 | 10 | 11 | 12 | 13 | 14 | 15 | 16 | 17 | 18 | 19 | 20 | Avg |
|---|---|---|---|---|---|---|---|---|---|---|---|---|---|---|---|---|---|---|---|---|---|
| Source | | | | | | | | | | 72.55 | | | | | | | | | | | 72.55 |
| **Single-Target TTA** | | | | | | | | | | | | | | | | | | | | | |
| Tent | 71.51 | 75.15 | 79.55 | 81.37 | 82.05 | 82.62 | 85.06 | 89.84 | 93.86 | 96.07 | 97.28 | 97.96 | 98.37 | 98.65 | 98.81 | 98.93 | 98.99 | 99.04 | 99.07 | 99.11 | 91.17 |
| + *ReservoirTTA* | 70.87 | 69.84 | 71.98 | 76.40 | 76.82 | 77.16 | 77.47 | 77.76 | 77.99 | 78.14 | 78.28 | 78.42 | 78.51 | 78.63 | 78.72 | 78.79 | 78.86 | 78.90 | 78.96 | 78.97 | 77.07 |
| **Continual TTA** | | | | | | | | | | | | | | | | | | | | | |
| CoTTA* | 72.81 | 72.52 | 72.44 | 72.50 | 72.67 | 72.96 | 72.87 | 72.97 | 73.28 | 73.34 | 73.55 | 73.64 | 73.54 | 73.57 | 73.51 | 73.74 | 74.00 | 74.06 | 74.04 | 74.26 | 73.31 |
| ETA | 65.58 | 65.15 | 63.10 | 63.16 | 62.12 | 62.02 | 61.96 | 62.34 | 61.94 | 62.45 | 61.62 | 61.44 | 61.64 | 61.37 | 61.29 | 61.20 | 60.98 | 60.62 | 61.32 | 61.44 | 62.14 |
| + *ReservoirTTA* | 65.61 | 62.03 | 60.95 | 60.04 | 59.58 | 59.09 | 58.58 | 58.21 | 58.18 | 57.93 | 57.87 | 57.77 | 57.68 | 57.81 | 57.51 | 57.41 | 57.62 | 57.72 | 57.53 | 57.29 | 58.82 |
| SAR | 67.92 | 65.45 | 64.65 | 64.29 | 64.20 | 64.46 | 64.85 | 66.46 | 79.56 | 74.84 | 65.94 | 64.80 | 64.37 | 64.16 | 64.22 | 64.58 | 65.91 | 71.91 | 70.09 | 78.91 | 67.58 |
| + *ReservoirTTA* | 68.13 | 65.75 | 64.22 | 63.13 | 62.42 | 61.96 | 61.70 | 61.50 | 61.39 | 61.37 | 61.35 | 61.40 | 61.49 | 61.61 | 61.80 | 62.06 | 62.35 | 62.73 | 63.14 | 63.40 | 62.64 |
| **Persistent TTA** | | | | | | | | | | | | | | | | | | | | | |
| RDumb | 65.38 | 65.82 | 65.73 | 66.28 | 66.29 | 66.49 | 66.45 | 66.85 | 65.83 | 66.17 | 66.24 | 65.42 | 66.00 | 66.05 | 66.29 | 66.72 | 65.83 | 65.98 | 66.01 | 66.08 | 66.10 |
| EATA | 62.90 | 62.02 | 61.63 | 61.18 | 60.86 | 60.91 | 60.56 | 60.71 | 60.23 | 60.23 | 60.42 | 59.93 | 59.97 | 59.86 | 60.09 | 59.53 | 60.07 | 60.08 | 60.04 | 59.85 | 60.55 |
| + *ReservoirTTA* | 63.41 | **58.46** | **56.86** | **56.26** | **55.85** | **55.64** | **55.44** | **55.46** | **55.65** | **55.58** | **55.53** | **55.44** | **55.37** | **55.36** | **55.29** | **55.34** | **55.31** | **55.24** | **55.20** | **55.22** | **56.10** |
| ROID* | 62.39 | 62.54 | 62.25 | 62.66 | 62.43 | 62.58 | 62.93 | 62.26 | 62.84 | 62.78 | 62.05 | 62.55 | 62.69 | 62.98 | 62.57 | 62.73 | 62.72 | 62.56 | 62.50 | 62.84 | 62.59 |
| +*ReservoirTTA* | **62.37** | 58.76 | 58.17 | 58.29 | 57.97 | 57.92 | 57.86 | 57.90 | 57.70 | 57.77 | 57.74 | 57.67 | 57.69 | 57.72 | 57.82 | 57.53 | 57.70 | 57.65 | 57.57 | 57.77 | 58.08 |

Table 19: **Average classification error rates (%) for *ResNet-50-GN* on ImageNet → ImageNet-C at severity level 5 under recurring continual CDC.** The lowest error is highlighted in **bold**. Results are averaged over 5 seeds.

| Method | 1 | 2 | 3 | 4 | 5 | 6 | 7 | 8 | 9 | 10 | 11 | 12 | 13 | 14 | 15 | 16 | 17 | 18 | 19 | 20 | Avg |
|---|---|---|---|---|---|---|---|---|---|---|---|---|---|---|---|---|---|---|---|---|---|
| Source | | | | | | | | | | 72.55 | | | | | | | | | | | 72.55 |
| **Single-Target TTA** | | | | | | | | | | | | | | | | | | | | | |
| Tent | 75.13 | 84.52 | 90.38 | 92.70 | 94.70 | 96.57 | 97.89 | 98.52 | 98.89 | 99.14 | 99.30 | 99.38 | 99.43 | 99.46 | 99.50 | 99.52 | 99.53 | 99.55 | 99.56 | 99.58 | 96.16 |
| + *ReservoirTTA* | 71.79 | 73.99 | 77.76 | 79.30 | 79.81 | 80.09 | 80.81 | 81.91 | 82.56 | 83.10 | 83.50 | 83.80 | 84.09 | 84.37 | 84.47 | 84.03 | 84.18 | 84.11 | 84.49 | 84.61 | 81.64 |
| **Continual TTA** | | | | | | | | | | | | | | | | | | | | | |
| CoTTA* | 72.88 | 72.69 | 72.68 | 72.85 | 73.02 | 73.16 | 73.29 | 73.34 | 73.48 | 73.80 | 74.00 | 74.20 | 74.37 | 74.43 | 74.54 | 74.81 | 75.11 | 75.33 | 75.48 | 75.68 | 73.96 |
| ETA | 66.95 | 64.42 | 63.11 | 62.21 | 61.54 | 61.01 | 61.19 | 61.19 | 60.96 | 61.19 | 61.68 | 61.23 | 61.19 | 60.89 | 61.26 | 61.16 | 61.62 | 61.33 | 61.53 | 61.90 | 61.89 |
| + *ReservoirTTA* | 66.00 | 62.27 | 60.77 | 59.87 | 59.37 | 58.70 | 58.64 | 58.40 | 58.05 | 57.74 | 57.72 | 57.68 | 57.57 | 57.48 | 57.37 | 57.28 | 57.23 | 57.24 | 57.06 | 57.11 | 58.68 |
| SAR | 68.41 | 65.18 | 64.33 | 63.95 | 64.03 | 63.59 | 63.61 | 63.58 | 63.77 | 63.96 | 65.04 | 67.01 | 79.86 | 72.86 | 65.41 | 64.26 | 63.86 | 63.59 | 63.46 | 63.52 | 65.66 |
| + *ReservoirTTA* | 70.75 | 68.91 | 67.63 | 66.65 | 66.06 | 65.43 | 65.16 | 65.07 | 64.85 | 64.72 | 64.74 | 65.18 | 65.27 | 65.24 | 65.36 | 65.51 | 65.58 | 65.69 | 65.70 | 65.79 | 65.96 |
| **Persistent TTA** | | | | | | | | | | | | | | | | | | | | | |
| RDumb | 67.36 | 67.18 | 67.05 | 66.48 | 66.24 | 67.08 | 67.61 | 66.97 | 67.02 | 66.48 | 66.76 | 66.77 | 67.09 | 66.69 | 67.23 | 66.82 | 67.11 | 67.19 | 66.68 | 66.41 | 66.91 |
| EATA | 64.57 | 62.31 | 61.91 | 61.31 | 61.32 | 61.04 | 60.96 | 61.04 | 60.73 | 60.30 | 60.10 | 60.31 | 60.21 | 59.74 | 60.16 | 59.94 | 59.94 | 59.86 | 59.79 | 59.98 | 60.78 |
| + *ReservoirTTA* | 66.05 | 60.47 | **58.22** | **57.21** | **56.58** | **56.03** | **56.05** | **55.83** | **55.54** | **55.35** | **55.19** | **55.06** | **54.96** | **54.85** | **54.81** | **54.77** | **54.76** | **54.74** | **54.69** | **54.78** | **56.30** |
| ROID* | 66.63 | 65.37 | 66.38 | 66.16 | 65.97 | 66.00 | 66.93 | 65.73 | 66.48 | 66.10 | 65.82 | 66.00 | 66.01 | 66.47 | 66.32 | 66.28 | 66.50 | 66.46 | 66.03 | 65.91 | 66.18 |
| +*ReservoirTTA* | **63.41** | **59.65** | 59.10 | 58.93 | 58.80 | 58.51 | 58.67 | 58.49 | 58.68 | 58.38 | 58.34 | 58.42 | 58.29 | 58.28 | 58.15 | 58.11 | 57.98 | 58.02 | 58.14 | 58.36 | 58.74 |

Table 20: **Average classification error rates (%) for *ViT-B-16* on ImageNet → ImageNet-C at severity level 5 under recurring continual CSC.** The lowest error is highlighted in **bold**. Results are averaged over 5 seeds.

| Method | 1 | 2 | 3 | 4 | 5 | 6 | 7 | 8 | 9 | 10 | 11 | 12 | 13 | 14 | 15 | 16 | 17 | 18 | 19 | 20 | Avg |
|---|---|---|---|---|---|---|---|---|---|---|---|---|---|---|---|---|---|---|---|---|---|
| Source | | | | | | | | | | 48.75 | | | | | | | | | | | 48.75 |
| **Single-Target TTA** | | | | | | | | | | | | | | | | | | | | | |
| Tent | 42.35 | 38.91 | 38.25 | 38.05 | 37.82 | 37.74 | 37.69 | 37.64 | 37.58 | 37.55 | 37.53 | 37.53 | 37.53 | 37.53 | 37.48 | 37.55 | 37.59 | 37.60 | 37.60 | 37.64 | 37.96 |
| + *ReservoirTTA* | 42.87 | 40.76 | 39.58 | 38.75 | 38.26 | 37.81 | 37.40 | 37.07 | 36.79 | 36.53 | 36.29 | 36.10 | 35.91 | 35.75 | 35.60 | 35.46 | 35.34 | 35.22 | 35.13 | 35.04 | 37.08 |
| **Continual TTA** | | | | | | | | | | | | | | | | | | | | | |
| CoTTA* | 48.55 | 48.06 | 47.69 | 47.26 | 46.94 | 46.64 | 46.35 | 46.12 | 46.00 | 46.03 | 46.08 | 46.12 | 46.32 | 46.30 | 46.31 | 46.34 | 46.45 | 46.40 | 46.38 | 46.46 | 46.64 |
| ETA | 38.91 | 36.69 | 36.22 | 35.93 | 35.87 | 35.69 | 35.71 | 35.67 | 35.47 | 35.55 | 35.36 | 35.32 | 38.54 | 44.64 | 38.43 | 36.21 | 38.69 | 48.53 | 48.34 | 48.43 | 38.71 |
| + *CoLA* | 40.98 | 38.47 | 36.74 | 35.87 | 35.44 | 34.96 | 34.76 | 34.58 | 34.44 | 34.41 | 34.27 | 34.21 | 34.13 | 34.04 | 33.93 | 34.04 | 33.89 | 33.90 | 33.90 | 33.82 | 35.04 |
| + *ReservoirTTA* | 39.40 | 36.46 | 34.96 | 34.13 | 33.56 | 33.19 | 32.87 | 32.62 | 32.47 | 32.35 | 32.26 | 32.19 | 32.08 | 32.02 | 31.96 | 31.97 | 31.93 | 31.89 | 31.89 | 31.86 | 33.10 |
| SAR | 44.52 | 41.85 | 40.84 | 40.37 | 40.14 | 40.02 | 39.89 | 39.83 | 39.76 | 39.72 | 39.69 | 39.67 | 39.67 | 39.65 | 39.62 | 39.58 | 39.57 | 39.54 | 39.52 | 39.50 | 40.15 |
| + *ReservoirTTA* | 43.44 | 41.56 | 40.54 | 39.85 | 39.38 | 38.93 | 38.57 | 38.28 | 38.00 | 37.78 | 37.56 | 37.37 | 37.20 | 37.05 | 36.93 | 36.79 | 36.68 | 36.55 | 36.45 | 36.33 | 38.26 |
| **Persistent TTA** | | | | | | | | | | | | | | | | | | | | | |
| RDumb | 39.32 | 39.23 | 38.88 | 38.69 | 38.64 | 39.07 | 41.49 | 40.62 | 38.65 | 38.98 | 38.69 | 39.46 | 39.19 | 38.85 | 38.47 | 38.66 | 39.06 | 42.09 | 38.95 | 38.68 | 39.28 |
| EATA | 39.11 | 37.07 | 36.15 | 35.70 | 35.37 | 35.10 | 34.96 | 34.88 | 34.76 | 34.58 | 34.63 | 34.51 | 34.69 | 34.62 | 34.40 | 34.53 | 35.30 | 34.37 | 34.39 | 34.60 | 35.19 |
| + *ReservoirTTA* | 39.76 | 37.23 | 35.98 | **35.11** | 34.63 | 34.18 | 33.81 | 33.54 | 33.33 | 33.19 | 33.03 | 32.86 | 32.77 | 32.65 | 32.55 | 32.44 | 32.40 | 32.33 | 32.28 | 32.24 | 33.81 |
| ROID* | 40.14 | 40.04 | 40.02 | 40.06 | 40.07 | 39.98 | 40.03 | 40.01 | 40.07 | 40.12 | 40.06 | 40.04 | 40.00 | 40.07 | 40.04 | 40.04 | 40.04 | 40.76 | 40.06 | 40.03 | 40.08 |
| + *ReservoirTTA* | 40.09 | 38.30 | 38.04 | 37.94 | 37.94 | 37.90 | 37.87 | 37.90 | 37.85 | 37.88 | 37.83 | 37.83 | 37.80 | 37.77 | 37.78 | 37.78 | 37.77 | 37.77 | 37.77 | 37.77 | 37.98 |
| **Prompt-based TTA** | | | | | | | | | | | | | | | | | | | | | |
| DPCore | 40.21 | 43.25 | 44.12 | 44.40 | 44.50 | 44.94 | 44.84 | 44.83 | 44.79 | 44.90 | 44.89 | 44.99 | 45.06 | 45.07 | 45.47 | 45.33 | 45.45 | 45.84 | 45.79 | 45.95 | 44.73 |
| *VPT+ReservoirTTA* | **37.98** | **36.05** | **35.43** | 35.22 | **34.96** | **34.83** | **34.63** | **34.51** | **34.40** | **34.31** | **34.38** | **34.25** | **34.12** | **34.04** | **34.00** | **33.90** | **33.85** | **33.77** | **33.77** | **33.72** | **34.61** |

Table 21: **Average classification error rates (%) for *ViT-B-16* on ImageNet → ImageNet-C at severity level 5 under recurring continual CDC.** The lowest error is highlighted in **bold**. Results are averaged over 5 seeds.

| Method | Recurring TTA visit 1 | 2 | 3 | 4 | 5 | 6 | 7 | 8 | 9 | 10 | 11 | 12 | 13 | 14 | 15 | 16 | 17 | 18 | 19 | 20 | Avg |
|---|---|---|---|---|---|---|---|---|---|---|---|---|---|---|---|---|---|---|---|---|---|
| Source | | | | | | | | | | 48.75 | | | | | | | | | | | 48.75 |
| **Single-Target TTA** | | | | | | | | | | | | | | | | | | | | | |
| Tent | 44.21 | 39.96 | 39.28 | 39.03 | 38.93 | 38.82 | 38.75 | 38.67 | 38.63 | 38.59 | 38.58 | 38.58 | 38.59 | 38.58 | 38.59 | 38.59 | 38.61 | 38.59 | 38.66 | 38.68 | 39.05 |
| + *ReservoirTTA* | 46.64 | 44.93 | 43.90 | 43.34 | 42.84 | 42.51 | 42.02 | 41.84 | 41.54 | 41.33 | 41.11 | 40.94 | 40.77 | 40.68 | 40.49 | 40.44 | 40.30 | 40.28 | 40.23 | 40.19 | 41.82 |
| **Continual TTA** | | | | | | | | | | | | | | | | | | | | | |
| CoTTA* | 48.37 | 47.90 | 47.43 | 46.97 | 46.63 | 46.34 | 45.91 | 45.69 | 45.49 | 45.44 | 45.51 | 45.62 | 45.54 | 45.67 | 45.61 | 45.68 | 45.70 | 45.66 | 45.90 | 45.94 | 46.15 |
| ETA | 42.89 | 37.66 | 36.99 | 36.74 | 36.48 | 36.25 | 36.20 | 36.24 | 35.96 | 36.08 | 36.08 | 35.94 | 35.93 | 36.91 | 36.40 | 36.46 | 36.29 | 35.87 | 36.06 | 35.87 | 36.66 |
| + *CoLA* | 40.90 | 37.61 | 36.67 | 36.10 | 35.86 | 35.64 | 35.47 | 35.33 | 35.25 | 35.13 | 35.20 | 35.11 | 35.12 | 35.02 | 35.10 | 35.10 | 35.11 | 34.97 | 35.03 | 34.87 | 35.73 |
| + *ReservoirTTA* | 41.55 | 38.17 | 36.62 | 35.60 | 35.03 | 34.51 | 34.14 | 33.96 | 33.79 | 33.60 | 33.46 | 33.55 | 33.42 | 33.33 | 33.18 | 33.10 | 33.07 | 33.07 | 32.96 | 32.92 | 34.45 |
| SAR | 45.86 | 42.16 | 41.12 | 40.60 | 40.32 | 40.14 | 40.03 | 39.96 | 39.91 | 39.80 | 39.77 | 39.73 | 39.71 | 39.66 | 39.68 | 39.64 | 39.60 | 39.60 | 39.57 | 39.57 | 40.32 |
| + *ReservoirTTA* | 45.47 | 43.00 | 41.89 | 41.27 | 40.75 | 40.28 | 39.96 | 39.70 | 39.46 | 39.25 | 39.09 | 38.95 | 38.80 | 38.55 | 38.43 | 38.26 | 38.13 | 38.04 | 37.88 | 37.91 | 39.75 |
| **Persistent TTA** | | | | | | | | | | | | | | | | | | | | | |
| RDumb | 43.91 | 40.27 | 39.32 | 39.32 | 39.74 | 40.37 | 40.43 | 39.53 | 39.16 | 39.03 | 38.88 | 39.53 | 39.55 | 39.26 | 40.30 | 39.15 | 39.51 | 40.07 | 39.74 | 39.20 | 39.81 |
| EATA | 39.89 | 37.20 | 36.34 | 35.76 | 35.62 | 35.28 | 35.20 | 35.00 | 34.87 | 34.80 | 35.24 | 34.66 | 34.62 | 34.97 | 34.54 | 34.58 | 34.55 | 34.51 | 34.46 | 34.53 | 35.33 |
| + *ReservoirTTA* | 42.08 | 39.54 | 38.14 | 37.29 | 36.62 | 36.08 | 35.79 | 35.56 | 35.24 | 35.14 | 35.04 | 34.99 | 34.80 | 34.66 | 34.49 | 34.37 | 34.30 | 34.26 | 34.16 | 34.19 | 35.84 |
| ROID* | 42.09 | 41.12 | 41.73 | 46.97 | 46.24 | 41.18 | 42.15 | 45.99 | 44.52 | 41.81 | 41.99 | 41.12 | 40.91 | 41.28 | 42.01 | 42.81 | 41.45 | 41.43 | 41.21 | 41.12 | 42.46 |
| + *ReservoirTTA* | 40.35 | 38.47 | 38.04 | 38.01 | 38.00 | 37.92 | 37.86 | 37.89 | 37.90 | 37.92 | 37.89 | 37.85 | 37.83 | 37.83 | 37.85 | 37.81 | 37.78 | 37.75 | 37.77 | 37.82 | 38.03 |
| **Prompt-based TTA** | | | | | | | | | | | | | | | | | | | | | |
| DPCore | 42.79 | 45.68 | 46.94 | 47.14 | 44.86 | 42.40 | 42.25 | 41.90 | 43.09 | 43.42 | 44.07 | 45.85 | 45.53 | 45.40 | 46.93 | 46.43 | 47.41 | 47.46 | 47.33 | 47.58 | 45.22 |
| *VPT+ReservoirTTA* | **38.01** | **36.30** | **35.87** | **35.75** | **35.56** | **35.34** | **35.35** | **35.17** | **34.92** | **34.78** | **34.78** | **34.74** | **34.72** | **34.55** | **34.47** | 34.45 | 34.48 | 34.32 | 34.32 | 34.32 | **35.11** |

Table 22: **Average classification error rates (%) for *ResNet-50* in the CCC setting.** Each column shows the average error over an adaptation interval (e.g., the second column covers steps 6701–13400), with each adaptation step using a mini-batch of 64 images. The lowest error is highlighted in **bold**.

| Method | CCC Adaptation Step 6700 | 13400 | 20100 | 26800 | 33500 | 40200 | 46900 | 53600 | 60200 | 66800 | 73400 | 80000 | Avg |
|---|---|---|---|---|---|---|---|---|---|---|---|---|---|
| Source | 83.06 | 83.18 | 83.39 | 82.89 | 82.53 | 83.55 | 83.52 | 83.00 | 83.60 | 83.39 | 83.32 | 83.34 | 83.23 |
| **Single-Target TTA** | | | | | | | | | | | | | |
| TENT | 83.62 | 99.46 | 99.58 | 99.58 | 99.61 | 99.67 | 99.65 | 99.63 | 99.70 | 99.65 | 99.79 | 99.62 | 98.30 |
| + *ReservoirTTA* | 68.26 | 67.44 | 67.97 | 74.21 | 77.42 | 75.32 | 75.62 | 82.24 | 83.41 | 86.48 | 87.82 | 92.04 | 78.19 |
| **Continual TTA** | | | | | | | | | | | | | |
| CoTTA* | 71.14 | 64.98 | 65.03 | 67.93 | 66.57 | 65.51 | 65.81 | 69.75 | 65.17 | 65.97 | 67.23 | 66.71 | 66.82 |
| RoTTA | 71.08 | 70.48 | 69.46 | 75.42 | 80.96 | 86.56 | 92.88 | 98.28 | 97.61 | 98.69 | 99.56 | 99.31 | 86.69 |
| ETA | 64.69 | 66.61 | 66.37 | 69.73 | 72.95 | 72.99 | 72.95 | 76.99 | 76.43 | 75.83 | 77.77 | 78.65 | 72.66 |
| + *ReservoirTTA* | 63.07 | 60.35 | 59.52 | 62.74 | 62.62 | 60.52 | 60.14 | 65.12 | 60.03 | 61.17 | 61.64 | 61.90 | 61.57 |
| SAR | 65.41 | 69.47 | 82.62 | 97.78 | 72.72 | 66.80 | 78.04 | 94.28 | 73.83 | 64.02 | 75.19 | 90.22 | 77.53 |
| + *ReservoirTTA* | 65.49 | 61.35 | 60.54 | 63.14 | 62.63 | 60.89 | 59.89 | 64.71 | 60.05 | 61.57 | 61.59 | 61.14 | 61.92 |
| **Persistent TTA** | | | | | | | | | | | | | |
| RDumb | 62.33 | 59.05 | 58.32 | 60.98 | 60.17 | 58.37 | 58.61 | 63.21 | 57.94 | 59.08 | 60.03 | 59.83 | 59.83 |
| PeTTA | 68.91 | 62.93 | 61.18 | 64.10 | 63.18 | 62.62 | 63.33 | 66.15 | 61.95 | 61.69 | 63.39 | 63.75 | 63.60 |
| EATA | **60.85** | 57.95 | 57.51 | 60.33 | 59.71 | 58.07 | 58.51 | 62.91 | 58.39 | 58.96 | 59.99 | 60.23 | 59.45 |
| + *ReservoirTTA* | 61.68 | 58.12 | 57.29 | 59.83 | 58.98 | 56.96 | **56.99** | **61.19** | **56.60** | 57.31 | **58.17** | **57.86** | **58.42** |
| ROID* | 61.89 | 58.23 | 57.42 | 60.14 | 58.82 | 57.02 | 57.80 | 62.20 | 56.73 | 58.01 | 58.94 | 58.84 | 58.84 |
| + *ReservoirTTA* | 61.24 | **57.76** | **56.83** | **59.73** | **58.64** | **56.74** | 57.30 | 61.76 | 56.65 | **57.71** | 58.55 | 58.45 | 58.45 |

Table 23: **Average classification error rates (%) for *ViT-B-16* in the CCC setting.** Each column displays the average error over an adaptation interval (e.g., the second column covers steps 6701–13400), with each adaptation step performed on a mini-batch of 64 images. The lowest error is highlighted in **bold**.

| Method | CCC Adaptation Step 6700 | 13400 | 20100 | 26800 | 33500 | 40200 | 46900 | 53600 | 60200 | 66800 | 73400 | 80000 | Avg |
|---|---|---|---|---|---|---|---|---|---|---|---|---|---|
| Source | 53.27 | 57.65 | 52.95 | 50.45 | 56.05 | 54.92 | 57.53 | 56.04 | 54.36 | 50.75 | 51.05 | 48.11 | 53.59 |
| **Single-Target TTA** | | | | | | | | | | | | | |
| Tent | 46.28 | 45.30 | 40.81 | 41.49 | 44.52 | 43.08 | 42.88 | 46.76 | 59.94 | 99.80 | 99.93 | 99.88 | 59.22 |
| + *ReservoirTTA* | 47.74 | 46.30 | 41.50 | 41.94 | 43.97 | 41.29 | 41.55 | 44.75 | 40.89 | 40.50 | 39.49 | 39.82 | 42.48 |
| **Continual TTA** | | | | | | | | | | | | | |
| CoTTA* | 50.97 | 49.45 | 44.57 | 45.53 | 49.08 | 45.34 | 49.45 | 50.97 | 48.26 | 48.48 | 44.43 | 43.99 | 47.54 |
| ETA | 42.66 | 42.24 | 38.97 | 39.81 | 41.92 | 39.86 | 40.84 | 44.10 | 40.66 | 40.32 | 39.36 | 39.28 | 40.84 |
| + *CoLA* | 44.78 | 42.88 | 38.91 | 39.71 | 41.47 | 39.57 | 40.38 | 43.10 | 39.89 | 39.67 | 38.93 | 38.91 | 40.68 |
| + *ReservoirTTA* | 44.08 | 42.52 | 38.47 | 39.45 | 41.83 | 38.51 | **38.78** | **41.96** | **38.78** | **37.76** | **36.77** | **36.88** | **39.65** |
| SAR | 46.83 | 46.97 | 42.49 | 43.19 | 45.52 | 43.80 | 44.73 | 47.75 | 44.43 | 43.76 | 43.39 | 43.84 | 44.73 |
| + *ReservoirTTA* | 47.84 | 46.76 | 42.29 | 43.11 | 45.55 | 42.47 | 43.13 | 46.69 | 42.91 | 42.40 | 41.09 | 41.24 | 43.79 |
| **Persistent TTA** | | | | | | | | | | | | | |
| RDumb | 44.30 | 44.84 | 40.79 | 41.99 | 44.03 | 42.41 | 43.97 | 47.35 | 41.61 | 41.35 | 41.27 | 40.79 | 42.89 |
| EATA | 43.47 | **41.93** | **38.23** | **39.09** | 41.08 | 38.54 | 39.41 | 42.41 | 39.52 | 38.69 | 37.90 | 38.09 | 39.86 |
| + *ReservoirTTA* | 44.07 | 42.27 | 38.61 | 39.13 | **41.07** | 38.39 | 39.05 | 42.13 | 38.95 | 38.18 | 37.41 | 37.52 | 39.73 |
| ROID* | 46.01 | 45.91 | 41.94 | 42.66 | 44.73 | 42.21 | 43.70 | 47.40 | 42.50 | 42.91 | 42.00 | 42.00 | 43.66 |
| +*ReservoirTTA* | 44.91 | 44.77 | 40.85 | 41.95 | 43.87 | 41.23 | 42.65 | 46.49 | 41.88 | 41.76 | 41.15 | 41.17 | 42.72 |
| **Prompt-based TTA** | | | | | | | | | | | | | |
| DPCore | **42.16** | 42.62 | 40.20 | 42.70 | 44.82 | 41.30 | 43.42 | 46.14 | 42.46 | 42.06 | 42.01 | 42.72 | 42.72 |
| *VPT+ReservoirTTA* | 42.91 | 42.19 | 38.89 | 40.33 | 41.68 | **37.84** | 39.59 | 42.57 | 39.11 | 38.55 | 37.70 | 37.58 | 39.91 |

Table 24: Comparison on DomainNet-126 under CSC setting (error rates %). Each source domain is trained separately; adaptation occurs over a recurring cycle of the remaining three domains repeated 20 times.

| Source Domain | real | painting | clipart | sketch | |
|---|---|---|---|---|---|
| Target Domain | clipart → painting → sketch (×20) | real → sketch → clipart (×20) | sketch → real → painting (×20) | painting → real → clipart (×20) | Avg |
| Source | 45.16 | 41.57 | 49.52 | 45.33 | 45.40 |
| EATA | 38.49 | 33.49 | 38.34 | 33.34 | 35.92 |
| +*ReservoirTTA* | **36.89** | **32.25** | 38.65 | **32.74** | **35.13** |
| Δ | -1.60 | -1.24 | +0.31 | 0.60 | -0.79 |

Table 25: Comparison on PACS under CSC setting (error rates %). Each source domain is trained separately; adaptation occurs over a recurring cycle of the remaining three domains repeated 20 times.

| Source Domain | photo | art painting | cartoon | sketch | |
|---|---|---|---|---|---|
| Target Domain | cartoon → art painting → sketch (×20) | sketch → photo → cartoon (×20) | photo → sketch → art painting (×20) | cartoon → art painting → photo (×20) | Avg |
| Source | 56.97 | 32.21 | 23.71 | 72.06 | 46.24 |
| EATA | 38.22 | 22.80 | 18.82 | 37.68 | 29.38 |
| +*ReservoirTTA* | **37.79** | **22.55** | **18.63** | **35.75** | **28.68** |
| Δ | -0.43 | -0.25 | -0.19 | -1.93 | -0.70 |

Table 26: Performance under gradual shift on CIFAR-100-C (CSC, ResNeXt-29). Each corruption cycles severities $1 \to 2 \to 3 \to 4 \to 5 \to 4 \to 3 \to 2 \to 1$. Error (%) at visits 1–20 and mean.

| Method | Visits 1–10 | | | | | | | | | |
| | 1 | 2 | 3 | 4 | 5 | 6 | 7 | 8 | 9 | 10 |
|---|---|---|---|---|---|---|---|---|---|---|
| Source | 33.58 | 33.58 | 33.58 | 33.58 | 33.58 | 33.58 | 33.58 | 33.58 | 33.58 | 33.58 |
| EATA | 25.35 | 25.44 | 25.41 | 25.41 | 25.40 | 25.43 | 25.39 | 25.44 | 25.39 | 25.38 |
| +*ReservoirTTA* ($K_{max} = 16$) | 25.17 | 25.06 | 24.99 | 25.03 | 25.00 | 25.02 | 25.01 | 25.09 | 25.09 | 25.05 |
| +*ReservoirTTA* ($K_{max} = 64$) | 25.17 | 24.99 | 24.92 | 24.93 | 24.94 | 24.93 | 24.92 | 24.95 | 24.95 | 24.91 |

| Method | Visits 11–20 | | | | | | | | | | Avg |
| | 11 | 12 | 13 | 14 | 15 | 16 | 17 | 18 | 19 | 20 | |
|---|---|---|---|---|---|---|---|---|---|---|---|
| Source | 33.58 | 33.58 | 33.58 | 33.58 | 33.58 | 33.58 | 33.58 | 33.58 | 33.58 | 33.58 | 33.58 |
| EATA | 25.45 | 25.42 | 25.44 | 25.44 | 25.43 | 25.42 | 25.42 | 25.43 | 25.38 | 25.40 | 25.41 |
| +*ReservoirTTA* ($K_{max} = 16$) | 25.07 | 25.08 | 25.05 | 25.10 | 25.07 | 25.08 | 25.09 | 25.05 | 25.06 | 25.07 | 25.06 |
| +*ReservoirTTA* ($K_{max} = 64$) | 24.94 | 24.91 | 24.94 | 24.94 | 24.94 | 24.93 | 24.97 | 24.97 | 24.97 | 24.96 | **24.95** |

Table 27: **Average classification error(%)** for *WideResNet-28* on CIFAR-10 $\to$ CIFAR-10-C (severity 5) under recurring CSC with **temporal correlation** (Dirichlet = 0.1).

| Method | Visits 1–10 | | | | | | | | | |
| | 1 | 2 | 3 | 4 | 5 | 6 | 7 | 8 | 9 | 10 |
|---|---|---|---|---|---|---|---|---|---|---|
| Source | 43.52 | 43.52 | 43.52 | 43.52 | 43.52 | 43.52 | 43.52 | 43.52 | 43.52 | 43.52 |
| RoTTA | 26.62 | 25.17 | 25.05 | 25.56 | 25.20 | 25.26 | 25.88 | 27.07 | 28.50 | 28.78 |
| +*ReservoirTTA* | 26.19 | 25.32 | 25.66 | 25.69 | 25.72 | 25.88 | 25.26 | 25.63 | 26.44 | 25.85 |

| Method | Visits 11–20 | | | | | | | | | | Avg |
| | 11 | 12 | 13 | 14 | 15 | 16 | 17 | 18 | 19 | 20 | |
|---|---|---|---|---|---|---|---|---|---|---|---|
| Source | 43.52 | 43.52 | 43.52 | 43.52 | 43.52 | 43.52 | 43.52 | 43.52 | 43.52 | 43.52 | 43.52 |
| RoTTA | 30.28 | 30.19 | 32.06 | 32.96 | 34.30 | 35.66 | 35.38 | 36.70 | 37.93 | 38.87 | 30.37 |
| +*ReservoirTTA* | 26.67 | 25.78 | 25.94 | 25.56 | 26.35 | 26.95 | 25.87 | 25.54 | 25.86 | 25.67 | 25.89 -4.48 |

