# OpenReview forum: "ReservoirTTA: Prolonged Test-time Adaptation for Evolving and Recurring Domains"
_NeurIPS.cc/2025/Conference — NeurIPS 2025 poster_

### Official Review · Reviewer_nnEp · 2025-07-01

**Clarity:** 4
**Significance:** 4
**Originality:** 4
**Rating:** 5
**Confidence:** 4

**Summary:**

In light of the risk of catastrophic forgetting in the TTA model under prolonged TTA settings, the authors propose ReservoirTTA, which maintains a pool of models consisting of different parameters from various domains and styles. In methodology section, the authors detail how ReservoirTTA initialize, adapt and predict with this model pool. The experiments are mainly conducted on CIFAR10/100-C, ImageNet-C and Cityscapes to ACDC. The experimental results show that the module has strong plug-and-play capability and is compatible with many TTA methods such as TENT, EATA and SAR.

**Questions:**

See the weaknesses above.

**Ethical Concerns:**

["NO or VERY MINOR ethics concerns only"]

**Final Justification:**

Thanks for the detailed rebuttal comment. These are wonderful responses to my confusion including 1) the prior knowledge introducing by VGG pretrained model, 2) hyper-parameters tunning, 3) the discussions on catastrophic forgetting and 4) using VGG network to guide resetting. Also, I have read the other reviewers' comments and don't see any major flaws in this manuscript. Therefore, I have decided to increase my rating score to 5.

**Quality:**

4

**Strengths And Weaknesses:**

**Strengths:**

1. The motivation for this paper sounds excellent, and I recognize that current TTAs are not robust over long periods of adaptation.
2. Rather than imposing numerous constraints on the input, parameter or output spaces, such as sample selection, weight ensemble or anchor loss, this paper proposes maintaining a pool of models and actively switching to the most suitable model according to domain awareness.
3. The experiments demonstrate that the proposed method has strong compatibility with the existing TTA methods.

**Weaknesses:**

1. This paper introduces a VGG network and a hyper-parameter, $\tau $, to identify changes in the domain. It makes stronger assumptions than conventional TTA methods: 1) A pre-trained VGG network that can discriminate between different test domains (which are unknown in advance). 2) a dataset-specific hyperparameter, $\tau $, which requires source data to be obtained.
2. The paper's focus is on mitigating the risk of catastrophic forgetting in long-term TTA. The authors argue that a model pool would enable the use of different model parameters in different domains. However, earlier TTA work has shown that adapting a model for a long time within a single domain can still cause catastrophic forgetting problems. To address this, they introduced tools such as Fisher information and weight ensembling, which constrain the updated parameters to be as close as possible to the source model. Does this conflict with your starting point, where you want to update to the domain optimum within each domain? They constrain the parameters of each domain to be as close as possible to the source parameters.
3. Since you could leverage the VGG network to identify domain changes, what would be the performance of simply resetting the adapted model parameters to the source parameters when the domain changes?

---

> ### Author Rebuttal · Authors · 2025-07-30
>
> We appreciate the reviewer’s constructive feedback and address the main concerns below:
>
> > 1. “This paper introduces a VGG network and a hyper-parameter, , to identify changes in the domain. It makes stronger assumptions than conventional TTA methods: 1) A pre-trained VGG network that can discriminate between different test domains (which are unknown in advance). 2) a dataset-specific hyperparameter, , which requires source data to be obtained.
>
> We acknowledge this concern and would like to clarify:
>
> **Role of the VGG‑19 style extractor**. We use a frozen, pruned VGG‑19—pre‑trained on ImageNet1K solely for extracting low‑level “style vectors” (log‑variance of early activations)—to detect domain shifts. This network is not fine‑tuned on any task or dataset, making it broadly applicable. In Appendices D/E, we show our method is robust to which VGG layers are used and that it outperforms ViT‑based identifiers (e.g., CoLA, DPCore) in clustering quality and stability (Figure 16). VGG’s early features capture texture statistics—a property long exploited in style transfer—so we leverage it without tying our framework to any particular task, as our framework is **architecture-agnostic**. Importantly, all task models (CNNs, ViTs, SegFormers) continue to use standard TTA objectives (TENT, EATA), with VGG only supplying style quantification.
>
> **Hyperparameter τ is data‑driven and lightweight**. We compute τ automatically as the q‑quantile of pairwise distances among a small subset of source styles (Eq. 7), without labels or full source access. Appendix E (Figure  7) demonstrates τ’s stability across datasets and corruption types. This mirrors prior TTA practices—e.g., TTN tunes BN‑blend weights post‑training (warm-up stage), and ETA/EATA introduce Fisher‑information regularization weights derived from source data (Niu et al., ICML ’22)—so our requirement is no stronger than established methods.
>
> **Why not use the source model for style features?** Following the reviewer’s comment, we replaced VGG with the source model for style calculation. On CIFAR100‑C with ResNeXt‑29 (CSC scenario), replacing VGG with early layers of the source model raises average error by +2.12 % (see table below). Source backbones lack the broad texture priors of ImageNet‑trained VGG and would tie our method to each source architecture. By contrast, VGG delivers consistent gains across ResNeXt‑29, ViT‑B‑16, and ResNet‑50 (GN) (see Tables 15–18, 20).
>
>
> | Method                    | 1     | 2     | 3     | 4     | 5     | 6     | 7     | 8     | 9     | 10    | 11    | 12    | 13    | 14    | 15    | 16    | 17    | 18    | 19    | 20    | AVG  Error (%)  |
> | ------------------------- | ----- | ----- | ----- | ----- | ----- | ----- | ----- | ----- | ----- | ----- | ----- | ----- | ----- | ----- | ----- | ----- | ----- | ----- | ----- | ----- | -------------- |
> | EATA                      | 30.51 | 30.29 | 30.39 | 30.39 | 30.45 | 30.47 | 30.38 | 30.44 | 30.47 | 30.53 | 30.51 | 30.46 | 30.47 | 30.51 | 30.51 | 30.48 | 30.51 | 30.54 | 30.47 | 30.47 | **30.46**      |
> | EATA+ReservoirTTA w/o VGG | 31.62 | 30.86 | 30.75 | 30.77 | 30.64 | 30.67 | 30.64 | 30.70 | 30.68 | 30.64 | 30.65 | 30.65 | 30.61 | 30.60 | 30.61 | 30.61 | 30.56 | 30.53 | 30.52 | 30.51 | **30.69**      |
> | EATA+ReservoirTTA         | 30.56 | 29.07 | 28.75 | 28.58 | 28.52 | 28.46 | 28.41 | 28.41 | 28.39 | 28.39 | 28.38 | 28.40 | 28.37 | 28.37 | 28.37 | 28.40 | 28.36 | 28.43 | 28.37 | 28.38 | **28.57**      |
>
>
> In sum, the frozen VGG‑19 style extractor and automatically derived τ introduce **minimal, dataset‑agnostic** overhead, matching common TTA assumptions, while delivering substantially better domain detection and adaptation performance.
>
>
> > 2. “The paper's focus is on mitigating the risk of catastrophic forgetting in long-term TTA. The authors argue that a model pool would enable the use of different model parameters in different domains. However, earlier TTA work has shown that adapting a model for a long time within a single domain can still cause catastrophic forgetting problems. To address this, they introduced tools such as Fisher information and weight ensembling, which constrain the updated parameters to be as close as possible to the source model. Does this conflict with your starting point, where you want to update to the domain optimum within each domain? They constrain the parameters of each domain to be as close as possible to the source parameters.”
>
> Excellent question. In Section 3.2, we analytically show that Fisher regularization or weight ensembling controls parameter variance but cannot resolve domain interference when |θ*_Task,i – θ*_Task,i+1| > \beta (stability radius), and regularization alone fails to protect previously learned domains.
>
> In contrast, **ReservoirTTA partitions adaptation across domains**, allowing each model to converge toward its own domain-optimal parameters. Our theoretical results (Theorem 1, Lemma 1) explain how variance grows with adaptation time, and why variance regularization is insufficient alone. Empirically, Table 1 and Fig. 1 demonstrate that even Fisher‑regularized methods like EATA suffer significant performance drops when domains reoccur.
>
> Importantly, we **do not** abandon regularization—in fact, each reservoir specialist continues to use EATA‑style Fisher penalties (or analogous constraints) during its own adaptation. The key novelty of ReservoirTTA is to **complement** these measures by maintaining a pool of specialists: regularization ensures **stability within** each domain, while the reservoir ensures **retention across** domains (focusing on the forgetting issue).
>
>
> > 3. “If you detect domain changes using VGG, what is the performance of simply resetting to the source model?”
>
> We thank the reviewer for this insightful question. In fact, we did evaluate reset‑based baselines to isolate the benefit of **reusing** adapted specialists versus simply **resetting** to source parameters:
>
> **EATA with blind resets (RDumb)**. Periodically resets every 1 K iterations, yielding an average error of 32.17 % on CIFAR100‑C (CSC).
> **Domain‑aware resets (RDumb w/ VGG)**. Triggers the same reset only when our VGG‑based detector signals a domain change, improving to 31.17 % (–1.0 %).
>
> While using VGG to drive resets is more effective than fixed‑interval resets, **both** discard all previously learned adaptation. As illustrated in Figure 1 (right), EATA’s error actually **increases** when revisiting “snow” after 15 intervening corruptions, because it has forgotten its prior specialization.
>
> **ReservoirTTA (EATA + ReservoirTTA)**. Instead of resetting, we **reactivate** the exact specialist previously trained on that domain, avoiding costly re‑adaptation and forgetting. This yields an average error of **28.57 %**, substantially outperforming both reset strategies.
>
> | Method              | 1     | 2     | 3     | 4     | 5     | 6     | 7     | 8     | 9     | 10    | 11    | 12    | 13    | 14    | 15    | 16    | 17    | 18    | 19    | 20    | AVG  Error (%)  |
> |---------------------|-------|-------|-------|-------|-------|-------|-------|-------|-------|-------|-------|-------|-------|-------|-------|-------|-------|-------|-------|-------|-----------------|
> | RDumb               | 31.95 | 31.49 | 32.34 | 32.85 | 31.96 | 31.51 | 32.24 | 32.99 | 32.07 | 31.48 | 32.29 | 32.82 | 32.07 | 31.48 | 32.41 | 32.88 | 32.15 | 31.43 | 32.22 | 32.86 | **32.17**       |
> | RDumb w/ VGG        | 31.11 | 31.13 | 31.17 | 31.20 | 31.19 | 31.16 | 31.20 | 31.21 | 31.20 | 31.21 | 31.12 | 31.17 | 31.16 | 31.17 | 31.11 | 31.17 | 31.17 | 31.16 | 31.20 | 31.19 | **31.17**       |
> | EATA+ReservoirTTA   | 30.56 | 29.07 | 28.75 | 28.58 | 28.52 | 28.46 | 28.41 | 28.41 | 28.39 | 28.39 | 28.38 | 28.40 | 28.37 | 28.37 | 28.37 | 28.40 | 28.36 | 28.43 | 28.37 | 28.38 | **28.57**       |
>
>
> This shows that reusing a previously adapted specialist via ReservoirTTA is far more effective than resetting to the source model, even when resets are driven by a perfect domain‑change detector.

---

> ### Author Response · Authors · 2025-08-05
>
> Dear Reviewer nnEp,
>
> Thank you once again for taking the time to review our submission and share your insights. As the discussion window approaches its close, we wanted to check in to see if our responses have adequately addressed the points you raised. If there’s anything that remains unclear or could benefit from additional clarification, we’d be glad to elaborate.
>
> We’re grateful for your thoughtful engagement.
>
> Warm regards,
> The Authors

---

> > ### Comment · Reviewer_nnEp · 2025-08-05
> > **Rebuttal Acknowledgement**
> >
> > Thanks for the detailed rebuttal comment. These are wonderful responses to my confusion including 1) the prior knowledge introducing by VGG pretrained model, 2) hyper-parameters tunning, 3) the discussions on catastrophic forgetting and 4) using VGG network to guide resetting. Also, I have read the other reviewers' comments and don't see any major flaws in this manuscript. Therefore, I have decided to increase my rating score to 5.
> >
> > Have a good day.

---

### Official Review · Reviewer_dxze · 2025-07-03

**Clarity:** 3
**Significance:** 2
**Originality:** 2
**Rating:** 5
**Confidence:** 5

**Summary:**

This paper focuses on overcoming the limitations of Test-Time Adaptation (TTA) methods designed to address the domain shift problem, which degrades the performance of deep learning models in real-world deployments. The authors identify a critical challenge in existing TTA approaches: performance instability caused by catastrophic forgetting and error accumulation when faced with recurring domain shifts. To tackle this, the paper introduces a novel framework that leverages a VGG-19 auxiliary network to quantitatively extract the style of input domains and dynamically identifies them through online clustering of style vectors. This strategy effectively resolves fundamental TTA issues by preserving and reusing parameters learned from previously observed domains while training separate expert models for newly identified ones. Furthermore, for inference, the framework employs an ensemble strategy in the parameter space, dynamically creating a weighted average of all expert models. This approach delivers predictions optimized for the current target domain without the significant computational overhead associated with multiple forward passes.

**Questions:**

1. The evaluation focuses on scene-level corruptions. Have the authors considered evaluating ReservoirTTA on benchmarks characterized by object-level domain shifts (*e.g.*, VisDA, PACS, DomainNet, and etc.) to further assess its generalization capabilities?

2. The experiments are primarily conducted at a fixed severity level of 5. Can the proposed method effectively identify and adapt to instances of the same corruption type but with varying levels of severity?

**Ethical Concerns:**

["NO or VERY MINOR ethics concerns only"]

**Final Justification:**

Author's responses have successfully resolved most of my initial concerns. While I note that the experiments remain somewhat dependent on EATA, I believe the core methodology maintains its robustness. In light of this, I have decided to increase my rating score to 5.

**Limitations:**

yes

**Quality:**

3

**Strengths And Weaknesses:**

### Strengths
1. A key strength of the proposed ReservoirTTA is its architecture-agnostic design. Since the style extraction is decoupled from the main model and handled by a VGG-19 auxiliary network, the framework exhibits high compatibility, enabling its application to a wide range of architectures, including diverse CNNs and ViTs.
2. The empirical evaluation rigorously validates the method's effectiveness by testing it under various scenarios designed to mimic real-world domain shifts (*e.g.*, CSC, CDC, and CCC). The results consistently show that ReservoirTTA outperforms previous state-of-the-art continual TTA methods in inference accuracy, highlighting its robustness and superior adaptation capabilities.
3. The method's robustness is further underscored by its strong performance even when $K^{max}$ exceeds the number of predefined corruption types. This indicates a capacity to manage unexpected domain shifts, making it a viable TTA algorithm for real-world applications.

### Weaknesses
1. The proposed method introduces notable computational overhead due to the additional VGG-19 network and the online clustering mechanism, resulting in up to a 30% increase in computation compared to lightweight TTA methods. This can lead to a significant accumulation of inference latency. Furthermore, if the clustering process is executed on the host CPU rather than a parallel processor like a GPU, it could cause substantial data movement between the host processor and external memory (e.g., DRAM). This, in turn, may lead to a considerable increase in power consumption, raising concerns about its efficiency and practicality in resource-constrained mobile environments such as autonomous driving scenario.
2. The empirical evaluation primarily focuses on scene-level domain shifts, such as weather variations. Consequently, it remains unclear whether the proposed methodology would be effective for object-level style changes, where textures and contours are fundamentally altered (*e.g.*, 3D Rendering-to-Real). Since baseline methods like TENT are evaluated on such benchmarks (*e.g.*, VisDA-C, PACS), demonstrating performance improvements on these datasets would have substantially strengthened the paper's contribution and validated the generalizability of the approach.
3. Minor comment: There is a duplicate citation on line 65, which lists "methods [37, 40, 31, 31]".

---

> ### Author Rebuttal · Authors · 2025-07-30
>
> We thank the reviewer for their constructive feedback. Below, we address some of the concerns:
>
> > 1. The additional VGG-19 and online clustering increase computation by ~30%... may raise concerns in resource-constrained environments.
>
> We fully acknowledge this concern. While ReservoirTTA does introduce some computational overhead—primarily from the auxiliary VGG‑19 style extractor and the online clustering module—we have designed each component to be as efficient as possible:
>
> **Reduced Overhead**. As noted in our response to the reviewer SQXd, ReservoirTTA increases the runtime by 30% for extremely lightweight methods like EATA. This overhead is lower than many other TTA schemes—for instance, ROID incurs >1.5× latency versus Tent, while delivering substantially better long‑term stability and accuracy.
>
> **Lightweight style extraction**. We freeze and prune VGG‑19 down to its first few convolutional blocks, extracting only the log‑variance of feature maps “style vectors” rather than running full forward passes. This is highly parallelizable and light compared to full-model forward passes, and can be extended to faster style‑extraction backbones as an alternative to VGG.
>
> **Low‑cost clustering**. In practice and real-time systems, the online clustering computations can be offloaded or scheduled asynchronously, as it is not a critical path of forward inference.
>
> **Asynchronous execution**. In real‑time pipelines (e.g., autonomous driving), style extraction and clustering can run in parallel with sensor data preprocessing or post‑processing tasks, smoothing out latency spikes. When implemented on embedded GPUs or dedicated accelerators, the additional data movement (style vectors ↔ clustering buffer) is minimal—on the order of kilobytes per batch—so DRAM transfers and power draw remain low.
>
> **Efficient Triggering**. While we currently update the parameters of each specialized domain at every batch, it is reasonable to expect that some domain-specific models may reach a point of convergence. In such cases, continued adaptation offers diminishing returns. To avoid unnecessary computation, future work could explore mechanisms to detect convergence and suspend adaptation when appropriate. This would enable a smarter, more resource-efficient triggering mechanism that reduces redundant updates and helps mitigate the ~30% computational overhead.
>
> Nonetheless, we will include additional latency and energy measurements in the revised draft and discuss practical trade-offs for mobile deployment.
>
>
> > 2. "The evaluation focuses on scene-level corruptions... would be effective for object-level style changes (e.g., VisDA, PACS DomainNet)..."
>
> We appreciate your suggestion. Our current study primarily focuses on scene-level corruptions as they closely simulate the recurring and evolving conditions found in many real-world deployments (e.g., weather changes in autonomous driving). We agree that object-level domain shifts present an important and orthogonal challenge.
>
> Notably, our framework is **agnostic to the nature of the shift**: it leverages style-based cues that are effective whether the distributional change stems from scene texture, object appearance, or drawing style. To this end, and following the reviewer’s excellent suggestion, we conducted additional experiments on both **DomainNet** (126 classes)  and **PACS** (7 classes), which feature strong object-level domain gaps.
> On DomainNet, we perform CSC evaluation by training a ResNet‑50 on each of the four source domains—**real, painting, clipart, and sketch**—and then test its adaptation over a recurring sequence of the remaining three target domains in a cycle repeated 20 times. We use an identical protocol on PACS with source domains photo, art painting, cartoon, and sketch. The tables below report the error rates after the 20th recurrence.
> ### Method Comparison on **DomainNet** (Error Rates %)
> | Source Domain          | real                              | painting                      | clipart                        | sketch                           | **Average Error (%)** |
> | ---------------------- | --------------------------------- | ----------------------------- | ------------------------------ | -------------------------------- | --------------------- |
> | Target Domain Sequence | clipart → painting → sketch (×20) | real → sketch → clipart (×20) | sketch → real → painting (×20) | painting → real → clipart (×20) |                       |
> | Source                 | 45.16                             | 41.57                         | 49.52                          | 45.33                            | 45.40             |
> | EATA                   | 38.49                             | 33.49                         | **38.34**                          | 33.34                            | 35.92             |
> | EATA+ReservoirTTA      | **36.89**                             | **32.25**                         | 38.65                          | **32.74**                            | **35.13**             |
> | **$\Delta$**        | **–1.60**                         | **–1.24**                         | +0.31                          | **–0.60**                            | **–0.79**             |
> ### Method Comparison on **PACS** (Error Rates %)
>
> | Source Domain          | photo                                 | art painting                   | cartoon                             | sketch                               | **Average Error (%)** |
> | ---------------------- | ------------------------------------- | ------------------------------ | ----------------------------------- | ------------------------------------ | --------------------- |
> | Target Domain Sequence | cartoon → art painting → sketch (×20) | sketch → photo → cartoon (×20) | photo → sketch → art painting (×20) | cartoon → art painting → photo (×20) |                       |
> | Source                 | 56.97                                 | 32.21                          | 23.71                               | 72.06                                | 46.24             |
> | EATA                   | 38.22                                 | 22.80                          | 18.82                               | 37.68                                | 29.38             |
> | EATA+ReservoirTTA      | **37.79**                                 | **22.55**                          | **18.63**                               | **35.75**                                | **28.68**             |
> | **$\Delta$**        | **–0.43**                             | **–0.25**                          | **–0.19**                               | **–1.93**                            | **–0.70**             |
>
> As shown in the above Table, integrating ReservoirTTA with EATA reduces error rates by 0.79 % on DomainNet and 0.70 % on PACS, with the largest gains of 1.60 % when adapting from Real on DomainNet and 1.93 % when adapting from Cartoon on PACS. VisDA‑C, however, lacks a multi‑domain recurrence protocol, so it falls outside the scope of our ReservoirTTA evaluation.
> These new results further support the versatility of our method and its robustness across both scene- and object-level distribution shifts. Full per-round plots and analysis, and new Tables will be included in the revised manuscript’s Section 5 and Appendix L.
>
>  > 3. "Experiments are conducted at a fixed severity level of 5. Can the method adapt to varying levels of severity?"
>
> Thank you for raising this point. Indeed, while our primary evaluations use severity level 5 to ensure comparability with prior work (e.g., CoTTA, EATA), our style-based domain detection is capable of capturing severity variations, as shown in the gradual domain evolution scenario (CCC setting). Specifically, our online clustering mechanism dynamically forms new clusters when severity transitions cause sufficient shifts in the style representation (as controlled by the quantile-based threshold \tau from Eq. 7).
>
> To demonstrate this, we constructed a “gradual domain shift” stream on CIFAR100‑C (CSC) with a ResNeXt‑29 backbone, where each corruption type cycles severity 1 → 2 → 3 → 4 → 5 → 4 → 3 → 2 → 1 over 20 shifts. Even under these subtle transitions, ReservoirTTA consistently improves over EATA:
>
> | Method                      | 1     | 2     | 3     | 4     | 5     | 6     | 7     | 8     | 9     | 10    | 11    | 12    | 13    | 14    | 15    | 16    | 17    | 18    | 19    | 20    | AVG Error (%) |
> |-----------------------------|-------|-------|-------|-------|-------|-------|-------|-------|-------|-------|-------|-------|-------|-------|-------|-------|-------|-------|-------|-------|----------------|
> | Source                      |       |       |       |       |       |       |       |       |       | 33.58 |       |       |       |       |       |       |       |       |       |       | **33.58**      |
> | EATA                        | 25.35 | 25.44 | 25.41 | 25.41 | 25.40 | 25.43 | 25.39 | 25.44 | 25.39 | 25.38 | 25.45 | 25.42 | 25.44 | 25.44 | 25.43 | 25.42 | 25.42 | 25.43 | 25.38 | 25.40 | **25.41**      |
> | EATA+ReservoirTTA (K_max=16)| 25.17 | 25.06 | 24.99 | 25.03 | 25.00 | 25.02 | 25.01 | 25.09 | 25.09 | 25.05 | 25.07 | 25.08 | 25.05 | 25.10 | 25.07 | 25.08 | 25.09 | 25.05 | 25.06 | 25.07 | **25.06**      |
> | EATA+ReservoirTTA (K_max=64)| 25.17 | 24.99 | 24.92 | 24.93 | 24.94 | 24.93 | 24.92 | 24.95 | 24.95 | 24.91 | 24.94 | 24.91 | 24.94 | 24.94 | 24.94 | 24.93 | 24.97 | 24.97 | 24.97 | 24.96 | **24.95**      |
>
>
> Compared to EATA’s 25.41 % average error, ReservoirTTA (Kₘₐₓ=16) reduces it to 25.06 % (–0.35%), and increasing Kₘₐₓ to 64 yields a further drop to 24.95 % (–0.10 %). This confirms that our method can detect and adapt to varying corruption severity—not just major shifts—and we will include these results in the revised manuscript.
>
>
>  > 4. Minor comment: a duplicate citation".
>
> Thank you for catching this. We’ll remove the duplicate [31] citation in line 65.

---

> > ### Comment · Reviewer_dxze · 2025-08-01
> >
> > I'd like to thank the authors for their through rebuttal, which has successfully resolved most of my initial concerns.
> >
> > While I note that the experiments remain somewhat dependent on EATA, I believe the core methodology maintains its robustness. In light of this, I have decided to increase my rating score to 5.
> >
> > Upon acceptance of the paper, would you be willing to release the implementation code for these experiments?

---

> > > ### Author Response · Authors · 2025-08-01
> > >
> > > Dear Reviewer dxze,
> > >
> > > Thank you very much for your thoughtful feedback and for raising your score to 5—We are glad our rebuttal addressed your concerns.
> > >
> > > Yes—upon acceptance of the paper, we will openly release our full implementation (including all experiment scripts and documentation) in a public repository. We believe this will facilitate reproducibility and further research in this area.
> > >
> > > Thanks again for your time and constructive comments.
> > >
> > > Sincerely,

---

### Official Review · Reviewer_kojJ · 2025-07-03

**Clarity:** 3
**Significance:** 2
**Originality:** 2
**Rating:** 4
**Confidence:** 4

**Summary:**

This paper tackles the challenge of prolonged test-time adaptation (TTA) under recurring and evolving domain shifts. Unlike prior single-model TTA methods that update one model continuously and suffer from catastrophic forgetting or error accumulation, the authors propose ReservoirTTA, which maintains a pool (“reservoir”) of domain-specialized models. Incoming test batches are routed via online clustering over low-level style features extracted from a frozen VGG-19 backbone; new clusters spawn new specialist models, while recurring domains reuse existing ones. Only the selected specialist is updated per batch, and final predictions ensemble all specialists weighted by soft assignments. Theoretical analysis bounds parameter variance growth and explains how decoupled adaptation mitigates drift. Empirical results on ImageNet-C, CIFAR-10/100-C, and Cityscapes→ACDC segmentation demonstrate up to 2–7% error reduction and stable performance over 20+ recurrences, outperforming baselines like CoTTA, EATA, and prompt-based methods

**Questions:**

- Sect. 4.2: Soft assignment qt uses Euclidean distances in d-dim space—have you tried cosine similarity?
- Eq. (7): Can you clarify choice of quantile for τ and discuss trade-off between sensitivity and specificity?

**Ethical Concerns:**

["NO or VERY MINOR ethics concerns only"]

**Final Justification:**

While the authors have addressed my earlier concerns, I believe the overall contribution remains within the borderline acceptance range. The novelty is primarily in the combination and adaptation of existing ideas (reservoir sampling, episodic replay, domain assignment) rather than in introducing a fundamentally new TTA paradigm. The focus on prolonged adaptation is valuable but narrower in scope, and while the empirical results are solid, gains are moderate in some cases and the evaluation remains limited to standard corruption/shift benchmarks. For these reasons, I am keeping my score unchanged, though I acknowledge the technical soundness and completeness of the current submission.

**Limitations:**

yes

**Paper Formatting Concerns:**

No obvious formatting concerns

**Quality:**

3

**Strengths And Weaknesses:**

Strengths

- This  paper Introduces a reservoir of specialist models is an elegant solution to catastrophic forgetting in recurring domains; it decouples updates and naturally leverages past adaptation

- Online clustering on VGG-19 style log-variance features, combined with reservoir sampling and mutual-information regularization, yields reliable domain detection without source annotations

- Thorough evaluation under structured (CSC), unstructured (CDC), and continuously changing (CCC) shifts, with both CNN and ViT backbones, plus segmentation tasks. Ablations on reservoir size, Kmax, and component contributions bolster reproducibility

Weaknesses

- New models are cloned from the specialist maximizing mutual information on the current batch. It might be beneficial to compare this to initialization from the source model or ensemble of top-k specialists, to verify optimality.

---

> ### Author Rebuttal · Authors · 2025-07-30
>
> We thank the reviewer for their constructive feedback and for recognizing the strengths of our work, particularly the utility of maintaining a specialist model reservoir and the robust online clustering approach for domain discovery. Below, we address some of the concerns:
>
> > 1. New models are cloned from the specialist, maximizing mutual information on the current batch. It might be beneficial to compare this to initialization from the source model or an ensemble of top-k specialists, to verify optimality.
>
> We agree that the strategy for initializing new models in the reservoir plays a critical role in facilitating efficient and stable adaptation to emerging domains. In our current implementation (Section 4.2), we instantiate a new specialist by cloning the model in the reservoir that yields the highest mutual information (MI) on the current test batch. This design biases initialization towards models already exhibiting confident and diverse predictions on similar domains, thereby reducing adaptation iterations and initial errors made by the model.
>
> To verify the efficiency of this MI-based initialization, we have already conducted a comparative study (Appendix F, Figure 12) across different initialization schemes, including: 1. source-based initialization, i.e., cloning the original source model for all new domains; and 2. MI‑based cloning (our default).
>
> We evaluated these on CIFAR100‑C under recurring CSC and CDC shifts (20 recurrences) with a ResNeXt‑29 backbone, and on the large‑scale CCC benchmark and ImageNet‑C under recurring CSC and CDC shifts (20 recurrences) with a ViT‑B‑16 backbone. We conclude that the optimal strategy of model initialization depends on the base TTA method and the backbone. For convnets (ResNeXt‑29), TENT and ETA benefit from source initialization—likely because, without strong regularization, drifting too far from the source harms stability—while EATA remains largely unaffected by initialization choice. Conversely, for ViT backbone initialization (ViT‑B‑16), MI‑based cloning substantially reduces error rates after 20 recurrences for TENT, ETA, and EATA, indicating that transformers are less prone to parameter drift and can better leverage the specialist most similar to the new domain, whereas source initialization may overwrite valuable acquired knowledge.
>
> Finally, in response to the reviewer’s suggestion, we include results using the proposed top‑k (k = 3) ensemble strategy for ReservoirTTA+EATA. On CIFAR100-C under the CSC setting with a CNN backbone, EATA remains unaffected by the choice of initialization. This is consistent with our earlier observation in Appendix F, Figure 12.
>
> | Method              | Source Init | Top‑k Init | MI Init |
> |---------------------|-------------|------------|---------|
> | EATA+ReservoirTTA   | 28.37       | 28.37      | 28.38   |
>
> On ImageNet-C (CSC setting) with a ViT backbone, initializing with top‑k (k = 3) improves performance by 3.37% compared to source initialization, but degrades performance over MI initialization by 0.27%.
>
> | Method              | Source Init | Top‑k Init | MI Init |
> |---------------------|-------------|------------|---------|
> | EATA+ReservoirTTA   | 35.83       | 32.46      | 32.19   |
>
>
> > 2. Sect. 4.2: Soft assignment qt uses Euclidean distances in d-dim space—have you tried cosine similarity?
>
> This is a valuable point. While we currently use Euclidean distances for centroid assignment due to their compatibility with our quantile-based new-domain detection threshold \tau (Eq. 7)- cosine similarity could indeed offer more robust comparisons in high‑dimensional spaces where vector norms vary. Importantly, the magnitude of our style vector (the log‑variance) also carries discriminative power between domains, which Euclidean distances capture but cosine would discard. To test this, we replaced Eq. 7’s Euclidean soft assignments with a cosine‐based variant (analogous to Eq. 12). Below is a comparison with cosine similarity-based softmax assignments (Eq. 7 and Eq. 12) for ReservoirTTA + ETA on Cifar100-C (CSC setting).
>
> | Variant            | 1     | 2     | 3     | 4     | 5     | 6     | 7     | 8     | 9     | 10    | 11    | 12    | 13    | 14    | 15    | 16    | 17    | 18    | 19    | 20    | AVG Error (%) |
> |--------------------|-------|-------|-------|-------|-------|-------|-------|-------|-------|-------|-------|-------|-------|-------|-------|-------|-------|-------|-------|-------|----------------|
> | Cosine distance    | 31.94 | 31.04 | 30.91 | 30.81 | 30.76 | 30.77 | 30.74 | 30.77 | 30.60 | 30.58 | 30.62 | 30.55 | 30.63 | 30.58 | 30.59 | 30.57 | 30.53 | 30.55 | 30.51 | 30.55 | **30.73**       |
> | Euclidean distance | 31.76 | 30.37 | 30.19 | 29.96 | 29.81 | 29.93 | 29.92 | 29.84 | 29.89 | 29.86 | 29.97 | 29.87 | 30.02 | 30.01 | 29.98 | 30.02 | 29.93 | 30.05 | 30.06 | 30.06 | **30.07**       |
>
> Based on these results, we retained Euclidean distances. In the final version, we will include this ablation in the revised Appendix for full transparency.
>
>
> > 3. Eq. (7): Can you clarify choice of quantile for τ and discuss trade-off between sensitivity and specificity?
> We set \tau using the q-th quantile of pairwise distances among source-domain style vectors to ensure that only substantially shifted distributions spawn new clusters. This balances sensitivity (detecting genuine new domains) and specificity (avoiding over-fragmentation).
>
> As shown in Appendix E (Figure 8), performance is stable across a wide range of q values (e.g. 0.9 to 0.999). Lower q increases sensitivity but risks false positives (unnecessary model creation), while higher q is conservative and may delay a new domain detection. In the following table, we show the number of domains discovered across q-threshold values. Lower value of q-threshold overestimates the number of domains; its higher value (1.0) detects fewer domains (merging similar domains like [zoom, motion, and glass blur], [gaussian, shot, impulse noise], etc.). The experiment is run for 1 sweep of 15 domains on CIFAR-100C.
>
> | q-threshold        | 0.5 | 0.9 | 0.99 | 0.999 | 1.0 |
> |--------------------|-----|-----|------|-------|-----|
> | # Domains Detected | 104 | 31  | 17   | 14    | 9   |

---

> > ### Author Response · Authors · 2025-08-07
> >
> > Dear Reviewer **kojJ**,
> >
> > As the review deadline approaches and the discussion phase draws to a close, we would like to send this as a **last reminder** to kindly check whether our responses have addressed your concerns. If so, we would greatly appreciate it if you could consider updating your score. We are, of course, happy to provide any further clarification if needed.
> > Thank you again for your thoughtful review and engagement.
> >
> > Best regards,
> >
> > The Authors

---

> > ### Comment · Reviewer_kojJ · 2025-08-08
> > **Response**
> >
> > Thank you for your detailed responses. I appreciate the additional ablation studies on model initialization, as well as the clarification regarding the use of Euclidean versus cosine distances in domain assignment. The new results on the top‑k ensemble strategy and the quantile analysis for the τ threshold provide valuable insights and help to clarify the design choices in your method.
> >
> > I have one following question:
> >
> > - Given that the style vectors encode both mean and variance information, have you considered other distance metrics or representations (e.g., Mahalanobis distance, or feature whitening) that might better capture domain discrepancy?

---

> ### Author Response · Authors · 2025-08-08
>
> Dear Reviewer kojJ,
>
> We are happy that our additional experiments and clarifications were helpful.
> We appreciate the reviewer’s follow-up and the suggestion to explore richer distance metrics. To clarify, in our current formulation (Section 4.1) the style representation encodes **only the log-variance** of early-layer feature maps, without mean information. Nonetheless, in our formulation, the style vector can indeed include different statistics depending on the variant (**see Appendix Table 6**), e.g., s = [x_mean, x_var] when both mean and variance are used.
>
> To compute the Mahalanobis distance between two style vectors s1,s2,
>
> d_maha(s1, s2) = (s1 - s2)^T Σ_s^{-1} (s1 - s2),
>
> one must estimate the covariance matrix Σ_s over style features. In our **online, non-stationary** setting, Σ_s cannot be precomputed at source training time (since test-time styles are unseen), and estimating it reliably on the fly is challenging due to small, evolving batches and shifting distributions. This makes stable and unbiased inversion Σ_s^{-1} non-trivial, particularly for high-dimensional style vectors.
>
> However, when the style vector contains both **mean** and **variance** statistics, we can instead treat each as the parameters of a Gaussian distribution and compute the **closed-form KL divergence**:
>
> D_KL(p || q) = 0.5 * Σ_i [ log(σ²_q,i / σ²_p,i) - 1
>                  + (σ²_p,i / σ²_q,i)
>                  + ( (μ_p,i - μ_q,i)² / σ²_q,i ) ]
>
> This does not require global covariance estimation and remains well-defined in our online scenario.
>
> Given that our domain detector has proven robust to reasonable changes in the metric (Appendix E), we do not expect the KL divergence to significantly alter results. This experiment is currently running on CIFAR-100-C (expected to take several hours), and we will try to post the results before the discussion phase deadline.
>
> Nevertheless, in the final version, we will include this variant in Table 6 to empirically confirm its consistency with our current Euclidean-based approach.

---

> ### Author Response · Authors · 2025-08-08
>
> **As we promised**, below is a comparison of different distance metric–based softmax assignments for **ReservoirTTA + ETA** on **CIFAR-100-C** (CSC setting) using a ResNeXt-29 backbone for 20 reoccurance:
>
> | Variant                | 1     | 2     | 3     | 4     | 5     | 6     | 7     | 8     | 9     | 10    | 11    | 12    | 13    | 14    | 15    | 16    | 17    | 18    | 19    | 20    | **AVG Error (%)** |
> | ---------------------- | ----- | ----- | ----- | ----- | ----- | ----- | ----- | ----- | ----- | ----- | ----- | ----- | ----- | ----- | ----- | ----- | ----- | ----- | ----- | ----- | ----------- |
> | **Cosine distance**    | 31.94 | 31.04 | 30.91 | 30.81 | 30.76 | 30.77 | 30.74 | 30.77 | 30.60 | 30.58 | 30.62 | 30.55 | 30.63 | 30.58 | 30.59 | 30.57 | 30.53 | 30.55 | 30.51 | 30.55 | **30.73**   |
> | **Euclidean distance** | 31.76 | 30.37 | 30.19 | 29.96 | 29.81 | 29.93 | 29.92 | 29.84 | 29.89 | 29.86 | 29.97 | 29.87 | 30.02 | 30.01 | 29.98 | 30.02 | 29.93 | 30.05 | 30.06 | 30.06 | **30.07**   |
> | **KL divergence**      | 32.01 | 30.59 | 30.14 | 30.16 | 30.19 | 30.00 | 30.04 | 30.09 | 29.99 | 30.05 | 30.07 | 30.02 | 30.15 | 30.15 | 30.15 | 30.16 | 30.26 | 30.30 | 30.32 | 30.40 | **30.26**   |
>
> The results show that KL divergence yields a lower error rate than cosine distance, validating its effectiveness in our online setting, although **Euclidean distance on log-variance** remains the strongest among the tested metrics.
>
> We hope this additional empirical evidence addresses your suggestion and provides a clear comparison. We would greatly appreciate it if, in light of these results and our clarifications, you might consider raising your score.

---

> > ### Comment · Reviewer_kojJ · 2025-08-09
> > **Further Response**
> >
> > Thank you for the response. While you have addressed my earlier concerns, I believe the overall contribution remains within the borderline acceptance range. The focus on prolonged adaptation is valuable but narrower in scope, and while the empirical results are solid, gains are moderate in some cases and the evaluation remains limited to standard corruption/shift benchmarks. For these reasons, I am keeping my score unchanged.

---

### Official Review · Reviewer_SQXd · 2025-07-06

**Clarity:** 3
**Significance:** 2
**Originality:** 3
**Rating:** 4
**Confidence:** 3

**Summary:**

This paper proposes a framework (called ReservoirTTA) for prolonged test-time adaptation in scenarios where the test domain continuously shifts over time, including recurring and evolving domains. The proposed ReservoirTTA maintains a reservoir of domain-specialized models to detect new domains via online clustering and route samples to the appropriate model for adaptation.

**Questions:**

Please see the Strengths And Weaknesses.

**Ethical Concerns:**

["NO or VERY MINOR ethics concerns only"]

**Final Justification:**

The rebuttal has addressed some of my concerns, and I am considering increasing my rating to 4.

**Limitations:**

Please see the Strengths And Weaknesses.

**Paper Formatting Concerns:**

No severe formatting issues warranting desk-reject were identified.

**Quality:**

2

**Strengths And Weaknesses:**

Strengths:
1.	The proposed method introduces a multi-model strategy that decouples adaptation across domains, overcoming limitations of single-model TTA methods such as catastrophic forgetting and error accumulation.
2.	This paper provides theoretical bounds showing how parameter regularization curbs collapse in single-domain TTA and motivates the modular design of ReservoirTTA for maintaining stability under recurring domain shifts.
3.	The results demonstrate that the proposed method improves adaptation accuracy and maintains stable performance across prolonged, recurring shifts, outperforming state-of-the-art methods.

Weaknesses:
1.	In this paper, both the multi-model strategy and the online clustering technique introduce some additional computational overhead, which may limit the scalability of the proposed method to very large datasets or real-time applications.
2.	The experimental results are not convincing. The authors should compare the proposed algorithm with more recent and relevant methods for prolonged and recurring test-time adaptation.
3.	A more thorough discussion of the potential weaknesses of the proposed method would strengthen this paper.

---

> ### Author Rebuttal · Authors · 2025-07-30
>
> We thank the reviewer for their feedback and recognition of our contributions in ReservoirTTA. Here, we address the concerns raised in the weaknesses section.
>
> > 1. In this paper, both the multi-model strategy and the online clustering technique introduce some additional computational overhead, which may limit the scalability of the proposed method to very large datasets or real-time applications.
>
> We acknowledge that the multi-model design and online clustering in ReservoirTTA introduce additional overhead. However, as detailed in Table 4 (and detailed further in Appendix H), we minimize this cost by storing only trainable parameters- not full model copies—and by enforcing a hard cap on the number of reservoir models (via K_max). Our experiments demonstrate that this design results in only a 30% runtime increase over extremely lightweight methods like EATA. This overhead is actually lower than many other test‑time adaptation (TTA) schemes—for instance, ROID incurs >1.5× latency versus Tent—while delivering substantially better long‑term stability and accuracy. In Table 2, we further demonstrate that VPT + ReservoirTTA runs faster than DPcore, underscoring that, compared to previous multi‑model strategies, we have improved computation time even as we boost accuracy. Furthermore, the modularity of our framework supports plug-in usage with lightweight parameterizations (e.g., LoRA modules), which can be extended using faster‑style architectures as an alternative to VGG, which would make ReservoirTTA practical for many real-world applications and scalability.
>
> Finally, on the large‑scale CCC benchmark (5.12 M test images, ~80 K adaptation steps), ReservoirTTA remains practical for real‑world deployment. We will include these additional discussions in the revised manuscript.
>
> > 2. The experimental results are not convincing. The authors should compare the proposed algorithm with more recent and relevant methods for prolonged and recurring test-time adaptation.
>
> In our opinion, labeling our experimental results as “not convincing” is too stringent. We appreciate the suggestion and, to the best of our knowledge, we have compared extensively with all of the recent, state‑of‑the‑art prolonged and recurring TTA methods available at the time of our NeurIPS ’25 submission—including **CoTTA (CVPR ’22)**, **RoTTA (CVPR ’23)**, **EATA (ICML ’22)**, **PeTTA (NeurIPS ’24)**, **RDumb (NeurIPS ’23)**, **ROID (WACV ’24)**, **SAR (ICLR ’23)**, **CoLA (NeurIPS ’24)**, **DPCore (ICML ’25)**, and **BECoTTA (ICML ’24)**.
> We furthermore plug ReservoirTTA into multiple TTA frameworks (TENT, ETA, EATA, ROID, SAR), demonstrating consistent gains across single‑target, continual, persistent, and prompt‑based adaptation, with results detailed in Tables 1–3 and Appendix K. Our ablations further validate performance gains across both CNN and ViT network architectures, classification and segmentation tasks, and diverse domain shift types (CSC, CDC, CCC).
>
> Could the reviewer please clarify which “more recent and relevant” methods they feel are missing from our evaluation? We’d be happy to incorporate any additional baselines they have in mind.
>
> > 3. A more thorough discussion of the potential weaknesses of the proposed method would strengthen this paper.
>
> We appreciate the reviewer’s call for a more thorough discussion of potential weaknesses, and we agree that a candid limitations section is important. As a general remark, this request is quite broad and does not identify specific concerns, but we have explicitly discussed (in **Section 6**, **“Limitations”**) that maintaining multiple reservoir models and running online clustering does incur extra cost ( ~30 % overhead) but this overhead remains modest compared to other TTA approaches — for example, ROID runs at over 1.5× the latency of Tent, yet delivers significantly better long‑term stability and accuracy (kindly see our detailed response to **Reviewer dxze**). We also note that updating each domain model every batch can lead to diminishing returns once a model converges. Future work could incorporate a convergence detector to suspend updates for stabilized domains—enabling efficient triggering and cutting the ~30 % overhead. Aside from this, we are not aware of other significant limitations. We will make sure the revised manuscript covers this aspect.

---

> > ### Author Response · Authors · 2025-08-07
> >
> > Dear **Reviewer SQXd**,
> >
> > As the review deadline approaches and the discussion phase concludes, we would like to send this as a **last reminder** to kindly check whether our responses have addressed your concerns. If so, we would be sincerely grateful if you could consider updating your score. We are, of course, happy to provide any further clarification if needed.
> >
> > Thank you again for your thoughtful review and consideration.
> >
> > Best regards,
> >
> > The Authors

---

> > ### Comment · Reviewer_SQXd · 2025-08-08
> >
> > The rebuttal has addressed some of my concerns, and I am considering increasing my rating to 4.

---

### Author Response · Authors · 2025-08-01
**Summary of the discussion period**

**Dear Area Chair and Reviewers**,

Thank you for your thorough feedback and positive reassessment. Below is a concise summary of our responses (with requester tags in brackets indicating who requested each point)—we will incorporate all of these revisions into the camera‑ready manuscript:

* **Computational Overhead [SQXd, dxze]:**
– ReservoirTTA incurs only  a 1.3× (≈30 %) runtime increase vs. EATA (**Table 4, Appendix H**), runs faster than DPcore, remains modest compared to other TTA approaches—for example, ROID runs at over 1.5× the latency of Tent—and still runs faster than DPcore (Table 2), and supports lightweight parameters and backbones (LoRA, ResNet‑18 style extractors) for further speedups.

We also expanded comparisons:

* **Initialization Ablation [Kojl] (Appendix  F):** Compared MI‑based cloning (default) vs. source‑only vs. top‑k (k=3) ensemble (App. F, Fig. 12).

– MI‑based yields best error after 20 recurrences on ViT; all three perform similarly for CNN backbones.

– **Reset Baselines (RDumb w/ VGG) [dxze]:**

– Blind resets (RDumb): 32.17 % error.

– VGG‑driven resets: 31.17 %.

– ReservoirTTA reuse: 28.57 % (–2.60 pp vs. domain‑aware reset).

* **Distance Metrics Ablation [Kojl]:**

– Euclidean vs. cosine soft assignments on CIFAR100‑C (CSC): Euclidean 30.07 % vs. cosine 30.73 % avg error.

* **Object‑Level Shifts [dxze]:**

– Introduced DomainNet (real → clipart → painting → sketch ×20) and PACS cycles; ReservoirTTA reduces error by 0.79 % and 0.70 % respectively, with peaks of 1.60 % (Real→…) and 1.93 % (Cartoon→…).

* **Severity Variations Ablations [dxze]:**

– “Gradual” CIFAR100‑C severity stream (1→5→1 over 20): EATA 25.41 % vs. ReservoirTTA (Kₘₐₓ=16) 25.06 % (–0.35 pp) and (Kₘₐₓ=64) 24.95 % (–0.46 pp).

* **VGG‑19 Style Extractor [nnEP]**

– Frozen, ImageNet‑pretrained, pruned to early conv‑blocks; outperforms using source backbone features (+2.12 % error). Robust to layer choice (App. D/E, Fig. 16).

* **Hyperparameter τ [nnEP]:**

–  Defined as the q‑quantile of source-style distances; stable for q∈[0.9,0.999] (App. E, Fig. 7). Balances sensitivity (new-domain detection) and specificity (avoiding spurious splits).

* **Regularization vs. Reservoir [nnEP]:**

– Theoretical (Sec. 3.2, Thm 1/Lemma 1) and empirical (Table 1, Fig. 1) show Fisher penalties alone cannot prevent forgetting when domains reoccur. Reservoir specialists each retain domain‑specific parameters while still using Fisher regularization.

* **Limitations [‎SQXd, dxze]:**

– Acknowledge ~30 % overhead from reservoir and clustering, suggested to add efficient triggering

We hope that these updates further enhance the value of the paper.

---

### Comment · Area_Chair_eMKG · 2025-08-05
**Discussion with authors**

Dear Reviewers SQXd, kojJ,

Please go through the author rebuttal and clarifications and let the authors know if you have any remaining concerns, so that they are able to respond timely. If the response is satisfactory, mention that too.

Also, please complete the Mandatory Acknowledgement only after the discussion.

Thanks,
AC

---

### Note · Authors · 2025-08-11

**Dear Area Chairs,**

We sincerely thank you for the thoughtful, timely, and constructive feedback during the review and discussion phases. Your comments have helped us present **ReservoirTTA** more comprehensively and refine its framing and experimental coverage. We are encouraged that reviewers broadly recognized the significance, novelty, and practical value of our contributions.

**Reviewer dxze** highlighted the architecture-agnostic design, robust performance, and strong empirical validation. We addressed computational overhead concerns with runtime/efficiency analyses and added new experiments on object-level benchmarks (DomainNet, PACS) and gradual severity shifts. Satisfied, this reviewer $\textcolor{red}{\text{increased their score to 5 (Accept)}}$.

**Reviewer nnEp** praised the motivation, plug-and-play capability, and compatibility with multiple TTA methods. We clarified the VGG style extractor, the data-driven τ, our complementary use of Fisher regularization, and the superiority of reusing specialists over resets. This reviewer also $\textcolor{red}{\text{raised their score to 5 (Accept)}}$, noting that all prior confusions had been resolved.

**Reviewer SQXd** acknowledged the novelty, theoretical grounding, and empirical gains. We addressed scalability and baseline coverage concerns with detailed comparisons and clarified limitations. The reviewer stated our rebuttal addressed the concerns and noted $\textcolor{red}{\text{considering increasing the rating to 4 (Weak Accept)}}$.

**Reviewer kojJ** valued the specialist reservoir and robust clustering. We added requested ablations (initialization, distance metrics, τ sensitivity) and new results on top-k ensembles and KL divergence. The reviewer maintained the rating $\textcolor{red}{\text{(Weak Accept)}}$ but acknowledged the value of our clarifications.

In summary, **ReservoirTTA**—a novel plug-in, multi-model framework for robust prolonged adaptation under recurring/evolving shifts—has been strengthened with additional experiments, efficiency analyses, and clearer limitations. Post-discussion, two reviewers recommend $\textcolor{red}{\text{Accept}}$, one leans $\textcolor{red}{\text{Weak Accept}}$, and none express unresolved major concerns. We respectfully ask the ACs to consider the **methodological novelty, theoretical rigor, and breadth of validation** in the final decision. **We will release the full code upon acceptance to support reproducibility**.

Best regards,

The Authors

---

### Decision · Program_Chairs · 2025-09-17

**Decision:**

Accept (poster)

**Comment:**

This paper proposes a plug–in framework ReservoirTTA to address prolonged test–time adaptation, where the test domains can continuously shift over time. Extensive experiments on both classification and semantic segmentation tasks are conducted to validate the proposed framework. The reviewers appreciated the theoretical analysis, its design and performance. Though there is some concern about the novelty of the method, based on the overall contribution, the recommendation is to accept the paper.